# Equivariant Light Field Convolution and Transformer in Ray Space

**Yinshuang Xu**
University of Pennsylvania
xuyin@seas.upenn.edu

**Jiahui Lei**
University of Pennsylvania
leijh@seas.upenn.edu

**Kostas Daniilidis**
University of Pennsylvania
and Archimedes, Athena RC
kostas@cis.upenn.edu

## Abstract

3D reconstruction and novel view rendering can greatly benefit from geometric priors when the input views are not sufficient in terms of coverage and inter-view baselines. Deep learning of geometric priors from 2D images requires each image to be represented in a $2D$ canonical frame and the prior to be learned in a given or learned $3D$ canonical frame. In this paper, given only the relative poses of the cameras, we show how to learn priors from multiple views equivariant to coordinate frame transformations by proposing an $SE(3)$-equivariant convolution and transformer in the space of rays in 3D. We model the ray space as a homogeneous space of $SE(3)$ and introduce the $SE(3)$-equivariant convolution in ray space. Depending on the output domain of the convolution, we present convolution-based $SE(3)$-equivariant maps from ray space to ray space and to $\mathbb{R}^3$. Our mathematical framework allows us to go beyond convolution to $SE(3)$-equivariant attention in the ray space. We showcase how to tailor and adapt the equivariant convolution and transformer in the tasks of equivariant $3D$ reconstruction and equivariant neural rendering from multiple views. We demonstrate $SE(3)$-equivariance by obtaining robust results in roto-translated datasets without performing transformation augmentation.

## 1 Introduction

Recent years have seen significant advances in learning-based techniques [67, 68, 60, 69, 61, 73, 11, 53] harnessing the power of deep learning for extraction of geometric priors from multiple images and associated ground-truth shapes. Such approaches extract features from each view and aggregate these features into a geometric prior. However, these approaches are not $SE(3)$-equivariant to transformations of the frame where the priors and images are defined. While view pooling or calculating variance [69, 72, 46, 73, 11] can be used to aggregate features and tackle equivariance, view pooling discards the rich geometric information contained in a multiple view setup.

In this paper, we address the problem of learning geometric priors that are $SE(3)$-equivariant with respect to transformations of the reference coordinate frame. We argue that all information needed for tasks like novel view rendering or 3D reconstruction is contained in the light field [6, 37]. Our input is a light field, a function defined on oriented rays in 3D whose values can be the radiance or features extracted from pixel values. We will use the term light field, and we will be specific when it is a radiance field or a feature field. Images are discrete samples of this field: the camera position determines which rays are sampled, while the camera orientation leaves the sample of the light field unchanged up to pixel discretization. We model the light field as a field over a homogeneous space of $SE(3)$, the ray space $\mathcal{R}$ parameterized by the Plücker coordinates. The ray space $\mathcal{R}$ is the space of oriented light rays, for any ray $x \in \mathcal{R}$, the Plücker coordinate is $x = (\boldsymbol{d}, \boldsymbol{m})$, where $\boldsymbol{d} \in \mathbb{S}^2$ is the direction of the ray, and $\boldsymbol{m} = \boldsymbol{x} \times \boldsymbol{d}$ where $\boldsymbol{x}$ is any point on the ray.

37th Conference on Neural Information Processing Systems (NeurIPS 2023).

We define a convolution in the continuous ray space as an equivariant convolution on a homogeneous space [18]. Since our features are not limited to scalar values, we will draw upon the tools of tensor field networks and representation theory, discussed in detail in the Appendix. In Sec. 3.1 we study the group action of $SE(3)$ on $\mathcal{R}$, the stabilizer group for $\mathcal{R}$, and how $SE(3)$ transforms the feature field over $\mathcal{R}$. In Sec. 3.4, we focus on developing the equivariant convolution in $\mathcal{R}$, providing analytical solutions for the kernels with the derived constraints in convolution from $\mathcal{R}$ to $\mathcal{R}$ and from $\mathcal{R}$ to $\mathbb{R}^3$, respectively. Meanwhile, we make the kernel locally supported without breaking the equivariance. By varying the output domain of the convolution, we introduce equivariant convolutions from the ray space to the ray space and from the ray space to the $3D$ Euclidean space.

The constraint of the kernel limits the expressiveness of equivariant convolution when used without a deep structure. In Sec 3.5, we introduce an equivariant transformer in $\mathcal{R}$. The equivariant transformer generates the equivariant key, query, and value by leveraging the kernel derived in the convolution, resulting, thus, in invariant attention weights and, hence, equivariant outputs. We provide a detailed derivation of two cases of cross-attention: the equivariant transformer from $\mathcal{R}$ to $\mathcal{R}$ and the equivariant transformer from $\mathcal{R}$ to $\mathbb{R}^3$. In the first case, the features that generate the key and value are attached to source rays, while the feature generating the query is attached to the target ray. In the second case, the feature generating the query is attached to the target point.

We demonstrate the composition of equivariant convolution and transformer modules in the tasks of $3D$ reconstruction from multi-views and novel view synthesis given the multi-view features. The inputs consist of finite sampled radiance fields or finite feature fields, while our proposed equivariant convolution and transformer are designed for continuous light fields. If an object or a scene undergoes a rigid transformation and is resampled by the same multiple cameras, the $SE(3)$ group action is not transitive in the light field sample. This lack of transitivity can significantly impact the computation of equivariant features, mainly because the views are sparse, unlike densely sampled point clouds. Object motion introduces new content, resulting in previously non-existing rays in the light field sampling. Hence, our equivariance is an exact equivariance with respect to the choice of coordinate frame. In the 3D reconstruction task, we experimentally show that equivariance is effective for small camera motions or arbitrary object rotations and generally provides more expressive representations. In the $3D$ object reconstruction application, we first apply an equivariant convolutional network in ray space to obtain the equivariant features attached to rays. We then apply equivariant convolution and equivariant transformer from $\mathcal{R}$ to $\mathbb{R}^3$ to obtain equivariant features attached to the query point, which are used to calculate the signed distance function (SDF) values and ultimately reconstruct the object. In the generalized rendering task, our model queries a target ray and obtains neighboring rays from source views. Our composition of equivariant modules is based on IBRNet [61], which consists of view feature aggregation and ray transformer. We replace the view feature aggregation in [61] with the equivariant convolution and transformer over rays and the ray transformer part with the equivariant transformerover the points along the ray to get the density and color of the point, see Sec. 3.3. We summarize here our main contributions:

(1) We model the ray space as a homogeneous space with $SE(3)$ as the acting group, and we propose the $SE(3)$-equivariant generalized convolution as the fundamental operation on a light field whose values may be radiance or features. We derive two $SE(3)$-equivariant convolutions, both taking input ray features and producing output ray features and point features, respectively.

(2) To enhance the feature expressiveness, we extend the equivariant convolution to an equivariant transformer in $\mathcal{R}$, in particular, a transformer from $\mathcal{R}$ to $\mathcal{R}$ and a transformer from $\mathcal{R}$ to $\mathbb{R}^3$.

(3) We adapt and compose the equivariant convolution and transformer module for $3D$ reconstruction from multiple views and generalized rendering from multi-view features. The experiments demonstrate the equivariance of our models.

## 2   Related Work

**Equivariant Networks**   Group equivariant networks [15, 65, 62, 55, 63, 12, 19, 17, 22, 24, 23] provide deep learning pipelines that are equivariant by design with respect to group transformations of the input. While inputs like point clouds, 2D and 3D images, and spherical images have been studied extensively, our work is the first, as far as we know, to study equivariant convolution and cross-attention on light fields. The convolutional structure on homogeneous spaces or groups is sufficient and necessary for equivariance with respect to compact group actions as proved in [18, 1, 36]. Recently, Cesa et al. [8], Xu et al. [70] provided a uniform way to design the steerable kernel in

an equivariant convolutional neural network on a homogeneous space using Fourier analysis of the stabilizer group and the acting group, respectively, while Finzi et al. [26] proposed a numerical algorithm to compute a kernel by solving the linear equivariant map constraint. For arbitrary Lie groups, Finzi et al. [25], MacDonald et al. [39], Bekkers [4] designed the uniform group convolutional neural network. The fact that any $O(n)$ equivariant function can be expressed in terms of a collection of scalars is shown in [59]. For general manifolds, Cohen et al. [16], Weiler et al. [64] derived the general steerable kernel from a differential geometry perspective, where the group convolution on homogeneous space is a special case. The equivalent derivation for the light field is in the Appendix. Recently, equivariant transformers drew increasing attention, in particular for 3D point cloud analysis and reconstruction [27, 48, 10, 7]. A general equivariant self-attention mechanism for arbitrary groups was proposed in [44, 43], while an equivariant transformer model for Lie groups was introduced in Hutchinson et al. [32]. We are the first to propose an equivariant attention model in the $3D$ ray space.

**Light Field and Neural Rendering from Multiple Views**   The plenoptic function introduced in perception [6] and later in graphics [37] brought a new light into the scene representation problem and was directly applicable to the rendering problem. Instead of reconstructing and then rendering, light fields enabled rendering just by sampling the right rays. Recently, learning-based light field reconstruction [41, 34, 5, 66, 51, 2] became increasingly popular for novel view synthesis, while [50, 54, 53] proposed non-equivariant networks in the ray space. Due to the smaller dimension of the ray space, the networks in the ray space are more efficient compared to neural radiance fields [42], which leverages volumetric rendering. Several studies [73, 61, 53, 50, 11, 38, 13, 31, 58, 20] concentrate on generalizable rendering. These works are similar to ours in that they obtain the $3D$ prior from the $2D$ images, but they are not equivariant since they explicitly use the coordinates of the points or the rays in the network.

The most related equivariant rendering approaches to us are [21, 47, 45]. The equivariance in the paper [21] is not the equivariance we mean in this work: [21] enforces the geometric consistency via a loss function. [47] is not strictly equivariant and it depends on the assumption of upright views and the camera ID embeddings, which results in its non-equivariance to camera permutation. It achieves data-driven equivariance by randomly choosing the first camera frame as the canonical frame. While [45] addresses the frame problem through relative pose, it is not theoretically equivariant in cases of individual camera rotations around their axes or minor individual rotations accompanied by small content changes. We want to emphasize that our central contribution is to propose an equivariant convolution and transformer on ray space, which can be integrated into a wide range of 3D learning models.

**Reconstruction from Multiple Views**   Dense reconstruction from multiple views is a well-established field of computer vision with advanced results even before the introduction of deep learning [28]. Such approaches cannot take advantage of shape priors and need a lot of views to provide a dense reconstruction. Deep learning enabled semantic reconstruction, i.e., the reconstruction from single or multiple views by providing the ground-truth 3D shape during training [14, 67, 68, 40]. These approaches decode the object from a global code without using absolute or relative camera poses. Regression of absolute or relative poses applied in [35, 71, 56, 69, 72, 57, 3, 46, 33, 20] is non-equivariant.

## 3   Method

In this section, we will first introduce the (feature) field on the ray space and $3D$ Euclidean space, respectively, and the corresponding $SE(3)$ group actions on the values of the fields in Sec. 3.1. To offer readers a holistic grasp—from a broad overview down to the intricate specifics and from foundational concepts to advanced techniques— we will present the reconstruction and the generalized rendering with the neural components of their architectures (convolutional and attentional) and their inputs and outputs in Sec. 3.2 and Sec. 3.3. Following that, we expose our central contribution: equivariant convolution and attention in ray space in Sec. 3.4 and Sec. 3.5.

### 3.1   Feature field on Ray Space and $3D$ Euclidean Space

#### 3.1.1   Ray Space

The ray space is the space of oriented light rays. As introduced in the introduction and App. Ex. 1, we use Plücker coordinates to parameterize the ray space $\mathcal{R}$: for any ray $x \in \mathcal{R}$, $x$ can be denoted as $(\boldsymbol{d}, \boldsymbol{m})$, where $\boldsymbol{d} \in \mathbb{S}^2$ is the direction of the ray, and $\boldsymbol{m} = \boldsymbol{x} \times \boldsymbol{d}$ is the moment of the ray with $\boldsymbol{x}$

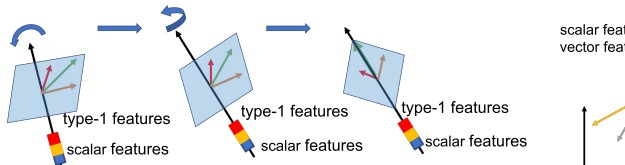
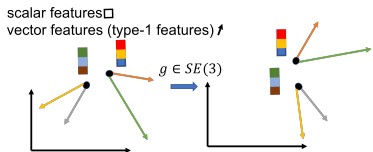

Figure 1: Feature attached to rays: we show the scalar feature and type-1 feature. When $\rho_2$ is the trivial representation, tensor features can be viewed in the plane orthogonal to the ray (the blue plane). When rotations act on the feature field, the scalar feature only changes position as attached to the rays: $(\mathcal{L}_g f)(x) = f(g^{-1}x)$; while the type-1 feature changes position and is itself rotated: $(\mathcal{L}_g f)(x) = \rho(\text{h}(g^{-1}, x)^{-1})f(g^{-1}x)$, where $\rho(\gamma, t) = e^{i\gamma}$.

Figure 2: Features attached to points: we show scalars and vectors (type-1 features). The black dot in the figure is the point, and the square and the vectors are the scalar and type-1 features attached to the point. When $g \in SE(3)$ acts on the feature field, we will see that the scalars are kept the same while the attached position is rotated, and the vector features change their position and alter their direction.

being a point on the ray. Then any $g = (R, \boldsymbol{t}) \in SE(3)$ acts on the the ray space as:

$$gx = g(\boldsymbol{d}, \boldsymbol{m}) = (R\boldsymbol{d}, R\boldsymbol{m} + \boldsymbol{t} \times (R\boldsymbol{d})). \tag{1}$$

The ray space $\mathcal{R}$ is a homogeneous space with a transitive group action by $SE(3)$. Given the origin in the homogeneous space as $\eta = ([0, 0, 1]^T, [0, 0, 0]^T)$ (the line representing $z$-axis), the stabilizer group $H$ that leaves $\eta$ unchanged is $SO(2) \times \mathbb{R}$ (the rotation around and translation along the ray). The ray space is, thus, isomorphic to the quotient space $\mathcal{R} \cong SE(3)/(SO(2) \times \mathbb{R})$. We parameterize the stabilizer group $H$ as $H = \{(\gamma, t) | \gamma \in [0, 2\pi), t \in \mathbb{R}\}$.

We follow the generalized convolution derivation for other homogeneous spaces in [18], which requires the use of principal bundles, section maps, and twists [29] explained in the appendix section A.2 and onwards. $SE(3)$ can be viewed as the principal $SO(2) \times \mathbb{R}$-bundle, where we have the projection $p : SE(3) \to \mathcal{R}$, for any $g \in SE(3)$, $p(g) = g\eta$; a section map $s : \mathcal{R} \to SE(3)$ can be defined such that $p \circ s = id_\mathcal{R}$. In App. Example 6, we elaborate on how we define the section map from the ray space to $SE(3)$ in our model. Generally, the action of $SE(3)$ induces a twist as $gs(x) \neq s(gx)$. The twist can be characterized by the twist function h $: SE(3) \times \mathcal{R} \to SO(2) \times \mathbb{R}$, $gs(x) = s(gx)\text{h}(g, x)$, we provide the twist function in our model and its visualization in App. Example 6.

### 3.1.2 Light Field

The light field can be modeled as a function from the ray space to a vector space, $f : \mathcal{R} \to V$. We also need to define the $SE(3)$ group action on the values of that field. Since the group action will be on a vector space $V$, we will use the corresponding group representation of the stabilizer group $\rho : SO(2) \times \mathbb{R} \to GL(V)$, see details in App. A.3. For example, a light field can be a radiance field $f$ that maps the ray space of oriented rays to their observed radiance (RGB) $f : \mathcal{R} \to \mathbb{R}^3$ which is a concatenation of three scalar fields over $\mathcal{R}$. The group representation $\rho$ in this case is the identity and $g \in SE(3)$ acts on the radiance field $f$ as $(\mathcal{L}_g f)(x) = f(g^{-1}x)$, shown as the scalar features in Fig. 1. Given that the stabilizer $H = SO(2) \times \mathbb{R}$ is a product group, the stabilizer representation can be written as the product $\rho(\gamma, t) = \rho_1(\gamma) \otimes \rho_2(t)$, where $\rho_1$ is the group representation of $SO(2)$ and $\rho_2$ is the group representation of $\mathbb{R}$. If the light field is a feature field (Fig. 1) with $\rho_2$ being the identity representation and $\rho_1$ corresponding to a type-1 field, $\rho_1(\gamma) = e^{i\gamma}$, then type-1 features change position and orientation when $g \in SE(3)$ acts on it. Having explained the examples of scalar (type-0) and type-1 fields, we introduce the action on any feature field $f$ as [18]:

$$(\mathcal{L}_g f)(x) = \rho(\text{h}(g^{-1}, x)^{-1})f(g^{-1}x), \tag{2}$$

where $\rho$ is the group representation of $SO(2) \times \mathbb{R}$ corresponding to the space $V$, determined by the field type of $f$, and h is the twist function introduced by $SE(3)$ as shown in App. Example 6.

### 3.1.3 Feature Field on $\mathbb{R}^3$

$\mathbb{R}^3$ is also a homogeneous space of $SE(3)$ like the ray space $\mathcal{R}$, with the stabilizer group as $SO(3)$, as stated in App. Example 2. For any $g = (R, \boldsymbol{t}) \in SE(3)$, it acts on the field $f$ over $\mathbb{R}^3$ also follows

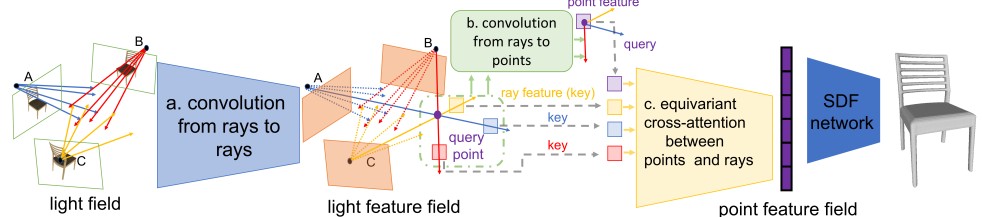

Figure 3: The pipeline of equivariant $3D$ reconstruction: Firstly, we obtain the feature field over the ray space. Secondly, we perform an equivariant convolution from ray space to point space. Thirdly, we apply a $SE(3)$ equivariant cross-attention module to obtain an equivariant feature for a query.

[18]:

$$(\mathcal{L}_g f)(x) = \rho(R)f(R^{-1}(x - \boldsymbol{t}))$$

,where $\rho$ is the group representation of $SO(3)$, since the twist function can be independent of the $3D$ position due to the fact that $SE(3) = \mathbb{R}^3 \rtimes SO(3)$ is a semidirect product group as stated in App. Example 4. The feature field over $\mathbb{R}^3$ and the corresponding group action is also used in [55, 63]. Fig. 2 visualizes the scalar feature ($l_{out} = 0$) and vector feature ($l_{out} = 1$) attached to one point, offering an intuitive understanding of the feature field over $\mathbb{R}^3$.

Given the feature field on the ray space and $3D$ Euclidean space and the corresponding group actions of $SE(3)$. We will show two 3D multi-view applications of the equivariant convolution and transformer: $3D$ reconstruction and generalized neural rendering. In each application, we start with the specific definition of equivariance and then outline the corresponding pipeline.

### 3.2 Equivariant 3D Reconstruction

The radiance field serves as the input for the $3D$ reconstruction, which ultimately generates a signed distance field (SDF) denoted by the function $e : \mathbb{R}^3 \to \mathbb{R}$. As aforementioned, the radiance field is the multi-channel scalar field over $\mathcal{R}$, while SDF is the scalar field over $\mathbb{R}^3$. A $3D$ reconstruction $\Phi : \mathcal{F} \to \mathcal{E}$, where $\mathcal{F}$ denotes the space of radiance fields and $\mathcal{E}$ denotes the space of signed distance fields, is equivariant when for any $g \in SE(3)$, any $x \in \mathbb{R}^3$, and any $f \in \mathcal{F}$, $\boldsymbol{\Phi}(\mathcal{L}_g \boldsymbol{f})(\boldsymbol{x}) = \mathcal{L}'_g(\boldsymbol{\Phi}(\boldsymbol{f}))(\boldsymbol{x})$, where $\mathcal{L}_g$ and $\mathcal{L}'_g$ are group actions on the light field and the SDF, respectively. Specifically, as $f$ and $e$ are scalar fields, $(\mathcal{L}_g f)(x) = f(g^{-1}x)$ for any $x \in \mathcal{R}$, and $(\mathcal{L}'_g e)(x) = e(g^{-1}x)$ for any $x \in \mathbb{R}^3$.

In practice, we have a finite sampling of the radiance field corresponding to the pixels of multiple views $V = \{f(x)|x \in L_V\}$, where $L_V$ denotes the ray set of multi-views and $f \in \mathcal{F}$ is the radiance field induced by multi views sample from. The $3D$ reconstruction $\Phi$ is equivariant when for any $g \in SE(3)$ and any $x \in \mathbb{R}^3$: $\boldsymbol{\Phi}(\boldsymbol{g} \cdot \boldsymbol{V})(\boldsymbol{x}) = \boldsymbol{\Phi}(\boldsymbol{V})(\boldsymbol{g}^{-1}\boldsymbol{x})$. If we denote $V$ as $(L_V, f)$, $g \cdot V = (g \cdot L_V, \mathcal{L}_g f)$, where $g \cdot L_V$ is $g$ acting on the rays defined Eq. 1.

We achieve equivariance using three steps as illustrated in Fig. 3: (1) the transition from pixel colors to a feature-valued light field (equi-CNN over rays), (2) the computation of features in $\mathbb{R}^3$ from features on the ray space by equivariant convolution from $\mathcal{R}$ to $\mathbb{R}^3$, and (3) the equivariant transformer with the query generated by the feature on the point we want to compute SDF and key/value generated by features on rays. Note that we need (3) following (2) because the output feature of a single convolution layer is not expressive enough due to the constrained kernel.For the detailed practical adaption of the convolution and transformer in $3D$ reconstruction, please see the App. B, where we approximate the intra-view with $SE(2)$ equivariant convolution.

### 3.3 Generalized Neural Rendering

The light feature field $f_{in} : \mathcal{R} \to V$ serves as the input for neural rendering, which ultimately generates the light field $f : \mathcal{R} \to \mathbb{R}^3$, a multi-channel scalar field over $\mathcal{R}$. A neural rendering $\Psi : \mathcal{I} \to \mathcal{F}$, where $\mathcal{I}$ denotes the space of the light feature fields and $\mathcal{E}$ denotes the space of the light field, is equivariant when for any $g \in SE(3)$, any $x \in \mathcal{R}$, and any $f_{in} \in \mathcal{I}$, $\boldsymbol{\Psi}(\mathcal{L}_g \boldsymbol{f_{in}})(\boldsymbol{x}) = \boldsymbol{\Psi}(\boldsymbol{f_{in}})(\boldsymbol{g}^{-1}\boldsymbol{x})$,where $\mathcal{L}_g$ is the group operator on the light feature field $f_{in}$, as shown in Eq. 2 depending on the feature type. In the experiment of this paper, the input light feature field is

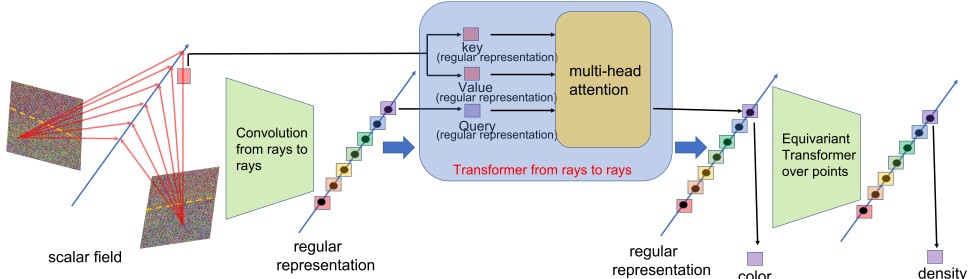

Figure 4: The pipeline of equivariant neural rendering. Firstly, we obtain the features of the points along the target ray through convolution over rays. Secondly, we apply the equivariant cross-attention module to obtain features for generating the color of the points. Finally, we use equivariant self-attention over the points along the ray to obtain features for generating the density of points.

scalar, i.e., $\mathcal{L}_g f_{in}(x) = f_{in}(g^{-1}x)$. Similar to reconstruction, in practice, the neural rendering $\Psi$ is equivariant when for any $g \in SE(3)$ and any $x \in \mathcal{R}$: $\boldsymbol{\Psi(g \cdot V)(x) = \Psi(V)(g^{-1}x)}$, where $V = \{f_{in}(x)|x \in L_V\}$, and if we denote $V$ as $(L_V, f_{in})$, then $g \cdot V = (g \cdot L_V, \mathcal{L}_g f_{in})$,

By restricting the field type of the output field over rays to have a group representation of $SO(2) \times \mathbb{R}$ as $\rho(\gamma, t) = \rho_1(\gamma) \otimes \rho_2(t)$, where $\rho_2$ is the regular representation, we can obtain the feature of points along the ray by convolution or transformer from $\mathcal{R}$ to $\mathcal{R}$. See App. Example 9 for more explanation of the regular representation. Alternatively, we can obtain the desired feature by applying convolution or transformer from $\mathcal{R}$ to $\mathcal{R}$, with output features attached to the target ray corresponding to different irreducible representations of the stabilizer group. These features can be interpreted as Fourier coefficients of the function of the points along the ray. The Inverse Fourier Transform yields features for the points along the ray. More details are in the App. I.1.

The feature of the points along the ray can be used to generate density and color for volumetric rendering [61, 73], or fed into attention and pooling for the final ray feature [58]. In this paper, we opt to generate the density and color and utilize volumetric rendering, which can be viewed as a specialized equivariant convolution from points to the ray. Method details are in App. I.

We achieve the equivariant rendering through three steps as shown in Fig. 4: (1) we apply equivariant convolution from rays to rays to get the equivariant feature for points along the rays, which is a specific field type over $\mathcal{R}$; 2) to enhance the feature expressivity, we apply an equivariant transformer from rays to rays to get the color for each point; (3) we apply the equivariant self-attention over the points along the ray to reason over the points on the same ray; the output feature of the points will be fed to multiple perceptron layers to get the density of the points.

### 3.4 Convolution in Ray Space

### 3.4.1 Convolution from Rays to Rays

The convolution, as stated in App. A.4 and [18] is then defined as

$$f^{l_{out}}(x) = \int_{\mathcal{R}} \kappa(s(x)^{-1}y)\rho_{in}(\mathrm{h}(s(x)^{-1}s(y)))f^{l_{in}}(y)dy, \tag{3}$$

where $\mathrm{h}(g)$ is the simplified form of the twist $\mathrm{h}(g, \eta)$. Eq.3 is equivariant to $SE(3)$ if and only if the convolution kernel $\kappa$ satisfies that $\kappa(hx) = \rho_{out}(h)\kappa(x)\rho_{in}(\mathrm{h}^{-1}(h, x))$, where $\rho_{in}$ and $\rho_{out}$ are the group representations of $SO(2) \times \mathbb{R}$ corresponding to the input feature type $l_{in}$ and output feature type $l_{out}$, respectively. We derive the solutions of the kernel in the App. Example 9.

**Local kernel support** The equivariance stands even if we constrain the kernel to be local. When $x = (\boldsymbol{d}_x, \boldsymbol{m}_x)$ meets the condition that $\angle(\boldsymbol{d}_x, [0, 0, 1]^T) \le \beta_0$ and $d(x, \eta) \le d_0$, $\kappa(x) \ne 0$, this local support will not violate the constraint that $\kappa(hx) = \rho_{out}(h)\kappa(x)\rho_{in}(\mathrm{h}^{-1}(h, x))$.

Then, convolution in Eq. 3 is accomplished over the neighbors only as visualized in Fig. 5. In Fig. 5, any ray $y = (\boldsymbol{d}_y, \boldsymbol{m}_y)$ (denoted in blue) in the neighborhood of a ray $x = (\boldsymbol{d}_x, \boldsymbol{m}_x)$ will go through the cylinder with $x$ as the axis and $d_0$ as the radius since $d(x, y) \leq d_0$.

Moreover, for any $y$, $\angle(\boldsymbol{d}_y, \boldsymbol{d}_x) \leq \beta_0$. Any ray $y \in \mathcal{N}(x)$ is on one tangent plane of a cylinder with $x$ as the axis and $d(x, y)$ as the radius when $d(x, y) > 0$.

### 3.4.2 Convolution from Rays to Points

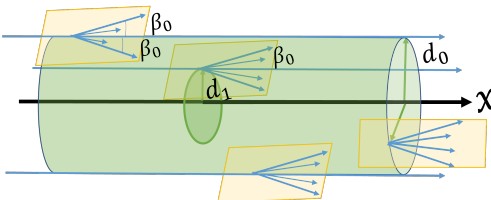

Figure 5: Neighborhood of a ray $x$ in the convolution.

In applications such as $3D$ reconstruction, key point detection, and $3D$ segmentation, we expect the output to be the field over $\mathbb{R}^3$. Using a convolution, we will define an equivariant map from light fields (fields on $\mathcal{R}$) to fields on $\mathbb{R}^3$. We denote with $H_1$ and $H_2$ the stabilizer groups for the input and output homogeneous spaces, respectively, i.e., $SO(2) \times \mathbb{R}$ and $SO(3)$ in this case. As shown in the App. Example 4, we can choose the section map $s_2 : \mathbb{R}^3 \to SE(3)$: $s_2(\boldsymbol{x}) = (I, \boldsymbol{x})$ for any $\boldsymbol{x} \in \mathbb{R}^3$ and I is the identity matrix. Following [18], the convolution from rays to points becomes:

$$f_2^{l_{out}}(x) = \int_{\mathcal{R}} \kappa(s_2(x)^{-1} y) \rho_{in}(\mathrm{h}_1(s_2(x)^{-1} s_1(y))) f_1^{l_{in}}(y) dy,$$

where $\mathrm{h}_1$ is the twist function corresponding to section $s_1 : \mathcal{R} \to SE(3)$ defined aforementioned, $\rho_{in}$ is the group representation of $H_1$ ($SO(2) \times \mathbb{R}$) corresponding to the feature type $l_{in}$. The subscripts 1 and 2 denote the homogeneous spaces the features are defined on. The convolution is equivariant if and only if the kernel $\kappa$ satisfies that $\kappa(h_2 x) = \rho_{out}(h_2) \kappa(x) \rho_{in}(\mathrm{h}_1^{-1}(h_2, x))$ for any $h_2 \in H_2$, where $\rho_{out}$ is the group representation of $H_2$ ($SO(3)$) corresponding to the feature type $l_{out}$.

In 3D reconstruction, $f^{l_{in}}$ is the scalar field over $\mathcal{R}$, i.e., $\rho_{in} = 1$. The convolution is simplified to $f_2^{l_{out}}(x) = \int_{G/H_1} \kappa(s_2(x)^{-1} y) f_1^{l_{in}}(y) dy$ and the corresponding constraint becomes $\kappa(h_2 x) = \rho_{out}(h_2) \kappa(x)$. App. Example 10 provides analytical kernel solutions.

### 3.5 Equivariant Transformer over Rays

We can extend the equivariant convolution to the equivariant transformer model. In general, the equivariant transformer can be formulated as:

$$f_2^{out}(x) = \sum_{y \in \mathcal{N}(x)} \frac{exp(\langle f_q(x, f_2^{in}), f_k(x, y, f_1^{in}) \rangle)}{\sum_{y \in \mathcal{N}(x)} exp(\langle f_q(x, f_2^{in}) f_k(x, y, f_1^{in}) \rangle)} f_v(x, y, f_1^{in}), \tag{4}$$

where the subscript 1 denotes the homogeneous space $M_1 \cong G/H_1$ of the feature field $f_1^{in}$ that generates the key and value in the transformer; the subscript 2 denotes the homogeneous space $M_2 \cong G/H_2$ of the feature field $f_2^{in}$ that generates query in the transformer, which is also the homogeneous space of the output feature $f_2^{out}$; $x$ and $y$ represent elements in the homogeneous spaces $M_2$ and $M_1$, respectively, where $y \in \mathcal{N}(x)$ indicates that the attention model is applied over $y$, the neighbor of $x$ based on a defined metric. $f_k$, $f_q$, and $f_v$ are constructed equivariant keys, queries, and values in the transformer. $f_k$ and $f_v$ are constructed by equivariant kernel $\kappa_k$ and $\kappa_v$ while $f_q$ is constructed through an equivariant linear map, see App. F for detailed construction.

When the transformer is a self-attention model, homogeneous space $M_1$ and $M_2$ are the same since $f_2^{in} = f_1^{in}$. The above equivariant transformer could be applied to the other homogeneous space other than $\mathcal{R}$, $\mathbb{R}^3$, and acting group other than $SE(3)$. This paper presents the equivariant cross-attention model over rays, i.e., $M_1$ is $\mathcal{R}$. When the transformer is the cross-attention from rays to rays, $M_2$ is also $\mathcal{R}$, the equivariant kernel $\kappa_k$ and $\kappa_v$ is the convolution kernel we derived in convolution from rays to rays in Sec. 3.4.1. When the transformer is the cross-attention from rays to points, $M_2$ is $\mathbb{R}^3$, the equivariant kernel $\kappa_k$ and $\kappa_v$ is the convolution kernel we derived in convolution from rays to points in Sec. 3.4.2. With the construction in App. F, we claim that the transformer from rays to rays or from rays to points, as shown in the equation 4, is equivariant. The proof is in App. G.

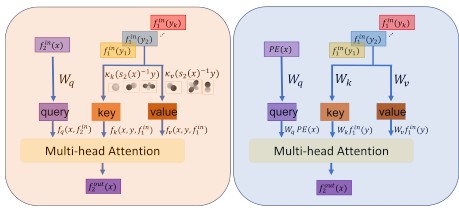 

Figure 6: In the equivariant transformer (L), positional encoding is not directly used due to its lack of equivariance. Instead, the relative position within the kernel is utilized. To generate the query $f_q$, we multiply the feature $f_2^{in}(x)$ (pre-existing or yielded by convolution) attached to $x$ (in $\mathcal{R}$ or $\mathbb{R}^3$, depending on the task) by the designed equivariant linear matrix $W_q$ (see App. F). The key $f_k$ and value $f_v$ are constructed using designed equivariant kernels $\kappa_k$ and $\kappa_v$. The transformer is equivariant due to equivariant $f_k$, $f_q$, and $f_v$. As $f_k$, $f_q$ and $f_v$ are equivariant, the entire transformer is equivariant. The conventional transformer (R) uses point position encoding for the query feature and obtains the query, key, and value through nonequivariant conventional linear mappings.

Figure 7: In the equivariant transformer (U), the query, key, and value are equivariant and can be composed of different types of features; they can be scalars, vectors, or higher-order tensors. The inner product, determined by the feature type, should apply to the same type of features. In contrast, the feature in a conventional transformer (D) is not equivariant, it does not contain vectors and tensors, and the inner product is conventional.

To better understand the equivariant transformer, we visualize the comparison of the equivariant cross-attention transformer and conventional transformer shown in Fig. 6. Meanwhile, as stated in App. F, key, query, and value are generally composed of different types of features and are multi-channel, allowing for the multi-head attention mechanism. In Fig. 7, we visualize the comparison of the equivariant multi-head attention module from rays to points with conventional multi-head attention module. The attention module from rays to rays follows a similar concept but with variations in the feature types due to the differing group representations of $SO(2) \times \mathbb{R}$ and $SO(3)$.

## 4 Experiment

### 4.1 3D Object Reconstruction from Multiple Views

**Datasets and Implementation** We use the same train/val/test split of the Shapenet Dataset [9] and render ourselves for the equivariance test. To render the views for each camera, we fix eight cameras to one cube's eight corners. The cameras all point in the same direction toward the object's center. We use the following notation to denote the variety of transformations in training and testing: $I$ (no transformation), $Z$ (optical axis rotation), $R$ (bounded 3-dof camera rotation), $Y$ (vertical axis object rotation), $SO(3)$ (full object rotation). The details of the five settings are in App. J.1.

As described in App. B, we use $SE(2)$ equivariant CNNs to approximate the equivariant convolution over the rays. For the fusion from the ray space to the point space model, we use one layer of convolution and three combined blocks of updating ray features and $SE(3)$ transformers. For more details, please see the App. J.2.

**Results** We evaluate our model in seven experiment settings, $I/I$, $I/Z$, $I/R$, $R/R$, $Y/SO(3)$, $SO(3)/SO(3)$. The setting A/B indicates training the model on the A setup of the dataset and evaluating it on the B setup. Following the previous works, we use *IoU* and *Chamfer-L1 Distance* as the evaluation metric. Quantitative results are reported in table 1, and qualitative results are in Fig. 8.

We compare with two other approaches [69], which follows a classic paradigm that queries 3D positions that are then back-projected to obtain image features for aggregation, and [72], which was state of the art in 3D object reconstruction from multi-views. Note that both baselines originally estimate the object poses, but we directly provide ground truth poses to them. See App. J.4 for more qualitative results.

In table 1, our model outperforms the [72] and [69] by a large margin on $I/Z$, $I/R$, and $Y/SO(3)$ settings. Although theoretically, our model is not equivariant to the arbitrary rotation of the object, $Y/SO(3)$ shows the robustness of our model to the object rotation and the generalization ability to some extent. Our model outperforms other models for the chair and car categories in $R/R$ and

| Method | chair | | | | | | |
|---|---|---|---|---|---|---|---|
| | I/I | I/Z | I/R | R/R | Y/Y | Y/SO(3) | SO(3)/SO(3) |
| Fvor w/ gt pose[72] | 0.691/0.099 | 0.409/0.253 | 0.398/0.257 | 0.669/0.113 | 0.687/0.103 | 0.518/0.194 | 0.664/0.114 |
| DISN w/ gt pose[69] | 0.725/0.094 | 0.335/0.396 | 0.322/0.405 | 0.500/0.201 | 0.659/0.120 | 0.419/0.303 | 0.549/0.174 |
| Ours | **0.731/0.090** | **0.631/0.130** | **0.592/0.137** | **0.689/0.105** | **0.698/0.102** | **0.589/0.142** | **0.674/0.113** |

| Method | airplane | | | | | | |
|---|---|---|---|---|---|---|---|
| | I/I | I/Z | I/R | R/R | Y/Y | Y/SO(3) | SO(3)/SO(3) |
| Fvor w/ gt pose[72] | 0.770/0.051 | 0.534/0.168 | 0.533/0.174 | **0.766**/0.053 | **0.760/0.052** | 0.579/0.147 | **0.746/0.056** |
| DISN w/ gt pose[69] | 0.752/0.058 | 0.465/0.173 | 0.462/0.171 | 0.611/0.104 | 0.706/0.069 | 0.530/0.151 | 0.631/0.103 |
| Ours | **0.773/0.050** | **0.600/0.092** | **0.579/0.100** | 0.759/**0.051** | 0.734/**0.052** | **0.597/0.101** | 0.722/**0.056** |

| Method | car | | | | | | |
|---|---|---|---|---|---|---|---|
| | I/I | I/Z | I/R | R/R | Y/Y | Y/SO(3) | SO(3)/SO(3) |
| Fvor w/ gt pose[72] | 0.837/0.090 | 0.466/0.254 | 0.484/0.258 | 0.816/0.107 | **0.830**/0.094 | 0.496/0.240 | 0.798/0.111 |
| DISN w/ gt pose[69] | 0.822/0.089 | 0.610/0.232 | 0.567/0.236 | 0.772/0.135 | 0.802/0.098 | 0.614/0.205 | 0.769/0.123 |
| Ours | **0.844/0.081** | **0.739/0.142** | **0.741/0.150** | **0.836/0.089** | **0.830/0.089** | **0.744/0.137** | **0.813/0.097** |

Table 1: The results for the seven experiments of 8-view $3D$ reconstruction for the ShapeNet dataset. The metrics in the cell are *IoU↑* and *Chamfer-L1 Distance↓*. We implement [72] and [69] ourselves on our equivariant dataset. For the performance of [69], we follow their work to conduct the multi-view reconstruction by pooling over the feature of every view. The value of *Chamfer-L1 Distance* is $\times 10$.

$SO(3)/SO(3)$ settings while it is slightly inferior to [72] in the airplane category. Notably, our model only requires relative camera poses, while [72] and [69] utilize camera poses relative to the object frame, leveraging explicit positional encoding of the query point in the object frame, which is concatenated to the point feature. In addition, our model performs better in several experiments in $I/I$ and $Y/Y$ settings. This superiority can be attributed to the $SE(3)$ equivariant attention model, which considers scalar features and ray directions. For a detailed discussion of the results, please see appendix Sec. J.3

We provide an ablation study of the effectiveness of $SE(2)$ CNNs, equivariant convolution, transformer, and type-1 feature (vector feature) in our model. Meanwhile, we compare our method with the model that explicitly encodes the direction of rays. Please see the App. J.5 for the details of the ablation study.

## 4.2 Neural Rendering

**Datasets and Implementation** We use the same training and test dataset as in [61], which consists of both synthetic and real data. Two experiment settings illustrate our model's equivariance: $I/I$ and $I/SO(3)$. $I/I$ is the canonical setting, where we train and test the model in the same canonical frame defined in the dataset. In the $I/SO(3)$ setting, we test the model trained in the conical frame under arbitrarily rotated coordinate frames while preserving relative camera poses and the relative poses between the camera and the scene, thereby preserving the content of the multiple views. Each individual view itself is not transformed. Note that this experiment's $SO(3)$ setup differs from the $R$ and $SO(3)$ setups used in the reconstruction. Further details and discussions on this difference can be found in App. K.1.

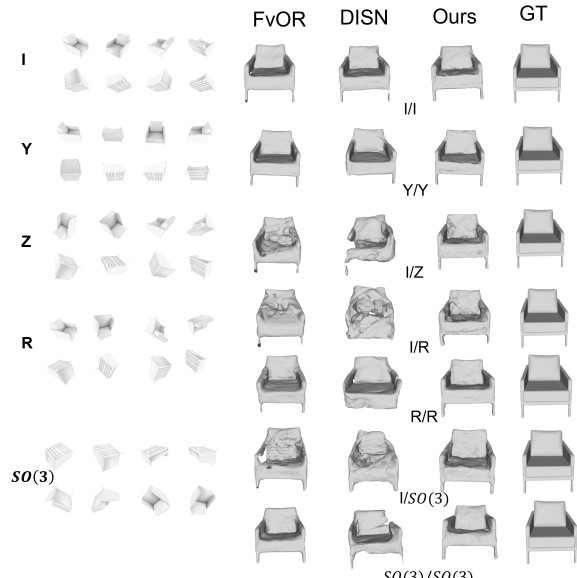

Figure 8: Qualitative results for equivariant reconstruction. Left: input views; Right: reconstruction meshes of different models and ground truth meshes show how the model is trained and tested, explained in the text.

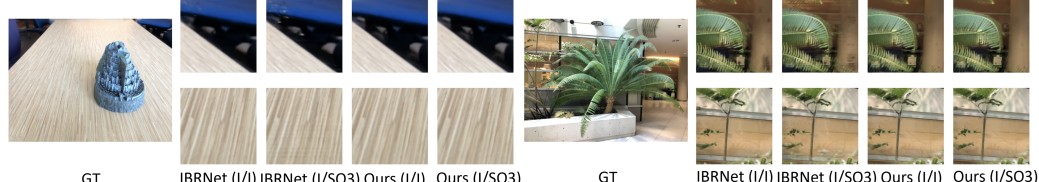

GT IBRNet (I/I) IBRNet (I/SO3) Ours (I/I) Ours (I/SO3) GT IBRNet (I/I) IBRNet (I/SO3) Ours (I/I) Ours (I/SO3)

Figure 9: Qualitative results for Generalized Rendering. We observe a performance drop for IBRNet from $I$ to $SO(3)$, while ours are robust to rotations in the testset.

| Dataset | Method | I/I | | | I/SO(3) | | | |
|---|---|---|---|---|---|---|---|---|
| | | PSNR↑ | SSIM↑ | LPIPS↓ | PSNR↑ | SSIM↑ | LPIPS↓ | Pix- Var↓ |
| Realistic Synthetic | IBRNet[61] | **26.91** | 0.928 | **0.084** | 26.77 | 0.923 | 0.091 | 66.58 |
| 360°[42] | Ours | 26.90 | **0.929** | 0.086 | **26.90** | **0.929** | **0.086** | **0.00** |
| Real | IBRNet[61] | **25.13** | **0.817** | **0.205** | 24.60 | 0.797 | 0.223 | 52.66 |
| Forward-Facing [41] | Ours | 24.93 | 0.808 | 0.212 | **24.93** | **0.808** | **0.212** | **0.00** |
| Diffuse Synthetic | IBRNet [61] | **37.21** | **0.989** | **0.019** | 37.07 | **0.988** | **0.019** | 34.51 |
| 360°[49] | Ours | 37.11 | 0.987 | **0.019** | **37.11** | 0.987 | **0.019** | **0.00** |

Table 2: The results for the experiments of generalized rendering without per-scene tuning. The metrics in the cell are PSNR↑, SSIM↑, and Pixel Variance↓ (denoted as Pix-Var). The evaluation of IBRNet[61] is performed by testing the released model on both canonical and rotated test datasets.

Our model architecture is based on IBRNet[61], with view feature aggregation and ray transformer components modifications. Specifically, we replace the view feature aggregation in [61] with the equivariant convolution and transformer over rays and the ray transformer part with the equivariant self-attention over the points along the ray. For more information on the implementation details, please refer to App. K.2.

**Results** We compare with IBRNet on $I/I$ and $I/SO(3)$ settings to show that our proposed models can be embedded in the existing rendering framework and achieve equivariance. Following previous works on novel view synthesis, our evaluation metrics are PSNR, SSIM, and LPIPS [74]. In the $I/SO(3)$ test period, we randomly rotate each data six times and report the average metrics. Meanwhile, we record the max pixel variance and report the average value. We show a qualitative result in Fig. 9, where IBRNet presents several blurred transverse lines in the $I/SO(3)$ setting while ours are robust to the rotation. In table 2, our model performs comparably with IBRNet [61] in $I/I$ setting without performance drop in $I/SO(3)$ setting. The slight decrease in PSNR/SSIM/LPIPS for IBRNet from $I/I$ to $I/SO(3)$ can be attributed to the training process involving multiple datasets with different canonical frames, which includes transformation augmentation and makes the model more robust to coordinate frame changes. Additionally, conventional metrics like PSNR/SSIM may not directly capture image variations. Therefore, we introduce an additional metric, pixel variance, to illustrate the changes better. We observe that IBRNet [61] exhibits pixel variance for different rotations, whereas our approach remains robust to rotation. Our method performs comparably with IBRNet in the $I/SO(3)$ setting in DeepVoxels [49] because the synthetic data consists of Lambertian objects with simple geometry, where the ray directions do not significantly affect the radiance. For more qualitative results, see App. K.3.

## 5 Conclusion and Broader Impacts

To learn equivariant geometric priors from multiple views, we modeled the convolution on the light field as a generalized convolution on the homogeneous space of rays with $SE(3)$ as the acting group. To obtain expressive point features, we extended convolution to equivariant attention over rays. The main limitation of the approach is the finite sampling of the light field. The sampling of the light field by sparse views cannot account for large object motions with drastic aspect change, leading to a breakdown of equivariance. This novel general equivariant representation framework for light fields can inspire further work on 3D vision and graphics tasks. We do not see any direct negative impact of our work, but it could have negative societal consequences if misused without authorization, for example, when using images violating privacy.

# 6 Acknowledgement

The authors gratefully acknowledge support by the support by the following grants: NSF FRR 2220868, NSF IIS-RI 2212433, NSF TRIPODS 1934960, NSF CPS 2038873.

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

# Appendix

The introduction of convolution and attention to the space of rays in 3D required additional geometric representations for which there was no space in the main paper to elaborate. We will introduce here all the necessary notations and definitions. We have accompanied this presentation with examples of specific groups to elucidate the abstract concepts needed in the definitions.

## A    Preliminary

### A.1    Group Actions and Homogeneous Spaces

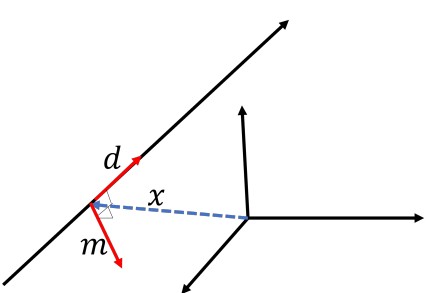

Figure 10: The visualization of Plücker coordinates: A ray $x$ can be denoted as $(\boldsymbol{d}, \boldsymbol{m})$ where $\boldsymbol{x}$ is any point on the ray $x$, and $\boldsymbol{d}$ is the direction of the ray $x$. $\boldsymbol{m}$ is defined as $\boldsymbol{x} \times \boldsymbol{d}$.

Given the action of the group $G$ on a homogeneous space $X$, and given $x_0$ as the origin of $X$, the stabilizer group $H$ of $x_0$ in $G$ is the group that leaves $x_0$ intact, i.e., $H = \{h \in G | hx_0 = x_0\}$. The group, $G$, can be partitioned into the quotient space (the set of left cosets) $G/H$ and $X$ is isomorphic to $G/H$ since all group elements in the same coset transform $x_0$ to the same element in $X$, that is, for any element $g' \in gH$ we have $g'x_0 = gx_0$.

***Example*** 1. $SE(3)$ acting on the ray space $\mathcal{R}$: Take $SE(3)$ as the acting group and the ray space $\mathcal{R}$ as its homogeneous space. We use Plücker coordinates to parameterize the ray space $\mathcal{R}$: any $x \in \mathcal{R}$ can be denoted as $(\boldsymbol{d}, \boldsymbol{m})$, where $\boldsymbol{d} \in \mathbb{S}^2$ is the direction of the ray, and $\boldsymbol{m} = \boldsymbol{x} \times \boldsymbol{d}$ where $\boldsymbol{x}$ is any point on the ray, as shown in figure 10. A group element $g = (R, \boldsymbol{t}) \in SE(3)$ acts on the the ray space as:

$$gx = g(\boldsymbol{d}, \boldsymbol{m}) = (R\boldsymbol{d}, R\boldsymbol{m} + \boldsymbol{t} \times (R\boldsymbol{d})). \tag{5}$$

We can choose the fixed origin of the homogeneous space to be $\eta = ([0, 0, 1]^T, [0, 0, 0]^T)$, the line identical with the $z$-axis of the coordinate system. Then, the stabilizer group $H$ (the rotation around and translation along the ray) can be parameterized as $H = \{(R_Z(\gamma), t[0, 0, 1]^T) | \gamma \in [0, 2\pi), t \in \mathbb{R}\}$, i.e., $H \simeq SO(2) \times \mathbb{R}$. We can simplify $H$ as $H = \{(\gamma, t) | \gamma \in [0, 2\pi), t \in \mathbb{R}\}$. $\mathcal{R}$ is the quotient space $SE(3)/(SO(2) \times \mathbb{R})$ up to isomorphism.

***Example*** 2. $SE(3)$ acting on the 3D Euclidean space $\mathbb{R}^3$: $\mathbb{R}^3$ is isomorphic to $SE(3)/SO(3)$. Consider another case when $SE(3)$ acts on the homogeneous space $\mathbb{R}^3$; for any $g = (R, \boldsymbol{t}) \in SE(3)$ and $\boldsymbol{x} \in \mathbb{R}^3$, $g\boldsymbol{x} = R\boldsymbol{x} + \boldsymbol{t}$. If the fixed origin is $[0, 0, 0]^T$, the stabilizer subgroup is $H = SO(3)$ since any rotation $g = (R, \boldsymbol{0})$ leaves $[0, 0, 0]^T$ unchanged.

***Example*** 3. $SO(3)$ acting on the sphere $\mathbb{S}^2$: $\mathbb{S}^2$ is isomorphic to $SO(3)/SO(2)$. The last example is $SO(3)$ acting on the homogeneous space sphere $\mathbb{S}^2$. Given the fixed origin point as $[0, 0, 1]^T$, the stabilizer group is $SO(2)$.

### A.2    Principal Bundle

As stated in [29, 18], the partition of the group $G$ into cosets allows us to treat the group $G$ as the principal bundle where the **total space** is $G$, the **base space** is the homogeneous space $G/H$[1], the canonical **fiber** is the stabilizer group $H$, the **projection map** $p : G \to G/H$ reads $p(g) = gH = gx_0 = x$. The **section** $s : G/H \to G$ of $p$ should satisfy that $p \circ s = id_{G/H}$, where $id_{G/H}$ is the identity map on $G/H$. Note that non-trivial principal bundles do not have a continuous global section, but we can define a continuous section locally on the open set $U \subseteq G/H$. The action of $G$ causes a twist of the fiber, i.e., $gs(x)$ might not be equal to $s(gx)$ though they are in the same coset. We use the **twist function** $\mathrm{h} : G \times G/H \to H$ to denote the twist: $gs(x) = s(gx)\mathrm{h}(g, x)$. Same as [18], we simplify $\mathrm{h}(g, eH)$ to be $\mathrm{h}(g)$, where $e$ is the identity element in $G$ and $eH = x_0$.

---

[1]We use $G/H$ to denote the homogeneous space since the homogeneous space $X$ can be identified with $G/H$ up to an isomorphism, i.e., $X \simeq G/H$.

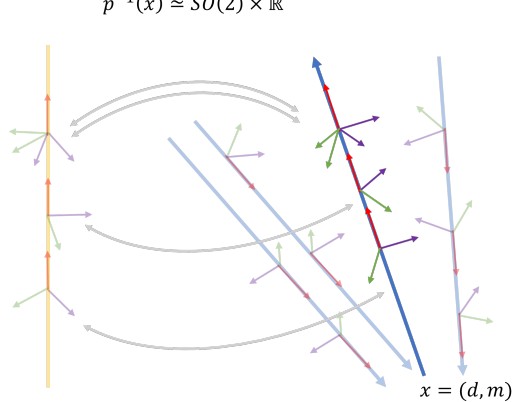

$$p^{-1}(x) \simeq SO(2) \times \mathbb{R}$$

$$x = (d, m)$$

$$H = SO(2) \times \mathbb{R}$$

Figure 11: We can view $SE(3)$ as an $SO(2) \times \mathbb{R}$-principal bundle, where the projection map $p : SE(3) \to \mathcal{R}$ is $p(R, \boldsymbol{t}) = (R[0,0,1]^T, \boldsymbol{t} \times (R[0,0,1]^T))$, and the inverse of $p$ is $p^{-1}(x) = \{(R, \boldsymbol{t}) | (R, \boldsymbol{t})\eta = x\}$. We use the coordinate frames (red axis denotes $Z$-axis, green axis denotes $X$-axis, and purple axis denotes $Y$-axis) to denote the element in $SE(3)$ because we can use the position of the coordinate origin to denote the translation $\boldsymbol{t}$ and use $X$-axis, $Y$-axis, and $Z$-axis to represent the first, second and third columns in rotation $R$. When we say next "the coordinate frame on the line/ray" we will mean that its origin is on the line/ray. By this convention, the coordinate frames representing the element in $H = SO(2) \times \mathbb{R}$ are the frames whose $Z$-axis aligns with $[0,0,1]^T$ and whose origin is $[0,0,t]^T$ for any $t \in \mathbb{R}$, i.e., frames on the yellow line in the left of the figure. For one ray $x = (\boldsymbol{d}, \boldsymbol{m})$ (illustrated as the chosen blue ray), the coordinate frames on the ray $x$ whose $Z$-axis aligns with the ray $\boldsymbol{d}_x$ are in $p^{-1}(x)$. As shown in the figure, there exists a bijection (gray double arrow line ) between $p^{-1}(x)$ and $H = SO(2) \times \mathbb{R}$. $p^{-1}(x)$ is isomorphic to $H = SO(2) \times \mathbb{R}$.

**Example** 4. Projection, section map and twist function for $\mathbb{R}^3$ and $SE(3)$: According to Ex. 2, we can consider a bundle with total space as $SE(3)$, base space as $\mathbb{R}^3$, and the fiber as $SO(3)$. For any $g = (R, \boldsymbol{t}) \in SE(3)$, the projection map $p : SE(3) \to \mathbb{R}^3$ projects $g$ as $p(R, \boldsymbol{t}) = \boldsymbol{t}$. For any $\boldsymbol{x} \in \mathbb{R}^3$, we can define the section map $s : \mathbb{R}^3 \to SE(3)$ as $s(\boldsymbol{x}) = (I, \boldsymbol{x})$. The twist function h $: SE(3) \times \mathbb{R}^3 \to SO(3)$ is that $\text{h}(g, \boldsymbol{x}) = s(g\boldsymbol{x})^{-1}gs(\boldsymbol{x}) = R$ for any $\boldsymbol{x} \in \mathbb{R}^3$ and any $g = (R, \boldsymbol{t}) \in SE(3)$. This twist function is independent of $\boldsymbol{x}$ due to the fact that $SE(3) = \mathbb{R}^3 \rtimes SO(3)$ is a semidirect product group as stated in [18].

**Example** 5. Projection, section map, and twist function for $\mathbb{S}^2$ and $SO(3)$: As shown in Ex. 3, $SO(3)$ can be viewed as a principal bundle with the base space as $\mathbb{S}^2$ and the fiber as $SO(2)$. With the rotation $R \in SO(3)$ parameterized as $R = R_Z(\alpha)R_Y(\beta)R_Z(\gamma)$, the projection $p : G \to G/H$ maps $R$ as follows:

$$p(R) = R_Z(\alpha)R_Y(\beta)R_Z(\gamma)[0,0,1]^T$$
$$= R_Z(\alpha)R_Y(\beta)[0,0,1]^T$$
$$= [sin(\beta)cos(\alpha), sin(\beta)sin(\alpha), cos(\beta)]^T.$$

For any $\boldsymbol{d} \in \mathbb{S}^2$, the section map $s : \mathbb{S}^2 \to SO(3)$ of $p$ should satisfy that $p \circ s = id_{\mathbb{S}^2}$ as mentioned above, i.e., $s(\boldsymbol{d})[0,0,1]^T = \boldsymbol{d}$. For instance, we could define the section map $s$ as:

$$s(\boldsymbol{d}) = R_Z(\alpha_{\boldsymbol{d}})R_Y(\beta_{\boldsymbol{d}}),$$

where $\alpha_{\boldsymbol{d}}$ and $\beta_{\boldsymbol{d}}$ satisfies that

$$\boldsymbol{d} = [sin(\beta_{\boldsymbol{d}})cos(\alpha_{\boldsymbol{d}}), sin(\beta_{\boldsymbol{d}})sin(\alpha_{\boldsymbol{d}}), cos(\beta_{\boldsymbol{d}})]^T.$$

Specifically, when $\boldsymbol{d} = [0,0,1]^T$, $\alpha_{\boldsymbol{d}} = 0$ and $\beta_{\boldsymbol{d}} = 0$; when $\boldsymbol{d} = -[0,0,1]^T$, $\alpha_{\boldsymbol{d}} = 0$ and $\beta_{\boldsymbol{d}} = \pi$.

As defined, the twist function h $: SO(3) \times \mathbb{S}^2 \to SO(2)$ is that $\text{h}(R, \boldsymbol{d}) = s(R\boldsymbol{d})^{-1}Rs(\boldsymbol{d})$.

**Example** 6. Projection, section map, and twist function for $\mathcal{R}$ and $SE(3)$: The final example is $SE(3)$ with $\mathcal{R}$ as the base space and $SO(2) \times \mathbb{R}$ as the fiber, which is the focus of this work, as shown

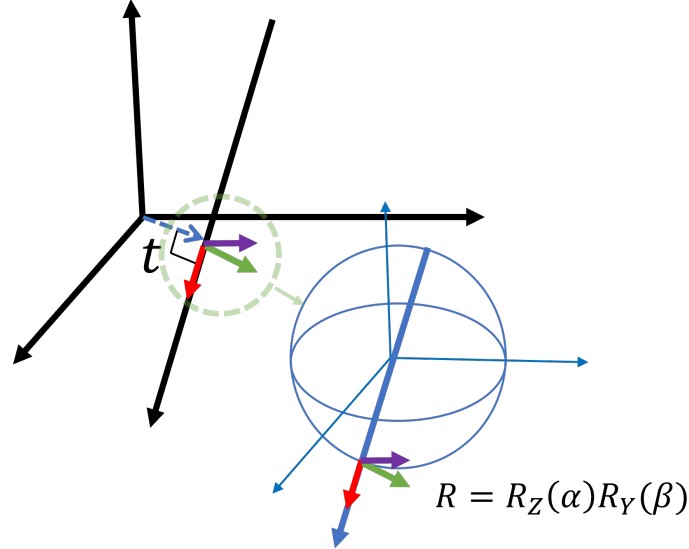

$$R = R_Z(\alpha)R_Y(\beta)$$

$$d = [\sin(\beta)\cos(\alpha), \sin(\beta)\sin(\alpha), \cos(\beta)]^T$$

Figure 12: For a ray $x = (\boldsymbol{d}, \boldsymbol{m})$, we need to choose an element $(R, \boldsymbol{t}) \in SE(3)$ as the representative element $s(x)$ such that $s(x)([0,0,1]^T, [0,0,0]^T) = x$. This figure shows one example of the section map $s$ from ray space to $SE(3)$. This map also serves as the section provided in this paper. The axes of the coordinate frame in the figure represent $R = s_a(\boldsymbol{d}) = R_Z(\alpha_{\boldsymbol{d}})R_Y(\beta_{\boldsymbol{d}})$, where the green axis, purple axis, and red axis represent 1st, 2nd and 3rd column in the rotation matrix $R$, respectively. The origin of the frame, $\boldsymbol{t} = s_b(\boldsymbol{d}, \boldsymbol{m}) = \boldsymbol{d} \times \boldsymbol{m}$, denotes the translation.

in figure 11. According to the group action defined in Eq. 5, the projection map $p : SE(3) \to \mathcal{R}$ is:

$$p((R, \boldsymbol{t})) = (R, \boldsymbol{t})\eta = (R[0,0,1]^T, \boldsymbol{t} \times (R[0,0,1]^T)).$$

This represents a ray direction $\boldsymbol{d}$ with the 3rd column of a rotation matrix and the moment $\boldsymbol{m}$ with the cross product of the translation and the ray direction. We can construct a section $s : G/H \to G$ using the Plücker coordinate:

$$s((\boldsymbol{d}, \boldsymbol{m})) = (s_a(\boldsymbol{d}), s_b(\boldsymbol{d}, \boldsymbol{m})),$$

where $s_a(\boldsymbol{d}) \in SO(3)$ is a rotation that $s_a(\boldsymbol{d})[0,0,1]^T = \boldsymbol{d}$, i.e., $s_a$ is a section map from $\mathbb{S}^2$ to $SO(3)$ as shown in Ex. 5; and $s_b(\boldsymbol{d}, \boldsymbol{m}) \in \mathbb{R}^3$ is a point on the ray $(\boldsymbol{d}, \boldsymbol{m})$. In this paper, we define the section map as $s((\boldsymbol{d}, \boldsymbol{m})) = (R_Z(\alpha_{\boldsymbol{d}})R_Y(\beta_{\boldsymbol{d}}), \boldsymbol{d} \times \boldsymbol{m})$, where $\alpha_{\boldsymbol{d}}$ and $\beta_{\boldsymbol{d}}$ satisfy that $\boldsymbol{d} = R_Z(\alpha_{\boldsymbol{d}})R_Y(\beta_{\boldsymbol{d}})[0,0,1]^T$, which is the same as Ex. 5. Figure 12 displays the visualization of the section map.

Given the section map, for any $g = (R_g, \boldsymbol{t}_g) \in SE(3)$ and $x = (\boldsymbol{d}_x, \boldsymbol{m}_x) \in \mathcal{R}$, we have the twist function $\mathrm{h} : SE(3) \times \mathcal{R} \to SO(2) \times \mathbb{R}$ is $\mathrm{h}(g, x) = s^{-1}(gx)gs(x) = (\mathrm{h}_a(R_g, \boldsymbol{d}_x), \mathrm{h}_b(g, x))$, where $\mathrm{h}_a : SO(3) \times \mathbb{S}^2 \to SO(2)$ is the twist function corresponding to $s_a$, as shown in Ex. 5, and $\mathrm{h}_b(g, x) = \langle R_g s_b(x) + \boldsymbol{t}_g - s_b(gx), R_g \boldsymbol{d}_x \rangle$. With the above section $s$ defined in this paper, the twist function $\mathrm{h} : SE(3) \times \mathcal{R} \to SO(2) \times \mathbb{R}$ is

$$\mathrm{h}(g, x) = s^{-1}(gx)gs(x) = (R_Z(R_g, \boldsymbol{d}_x), \langle \boldsymbol{t}_g, (R_g \boldsymbol{d}_x) \rangle),$$

where $R_Z(R_g, \boldsymbol{d}_x) = R_Y^{-1}(\beta_{R_g \boldsymbol{d}_x})R_Z^{-1}(\alpha_{R_g \boldsymbol{d}_x})R_g R_Z(\alpha_{\boldsymbol{d}_x})R_Y(\beta_{\boldsymbol{d}_x})$.

To understand the twist function clearly, we visualize a twist induced by a translation in $SE(3)$ in figure 13,

## A.3  Associated Vector Bundle

Given the principal bundle $G$, we can construct the associated vector bundle by replacing the fiber $H$ with the vector space $V$, where $V \simeq \mathbb{R}^n$ and $H$ acts on $V$ through a group representation

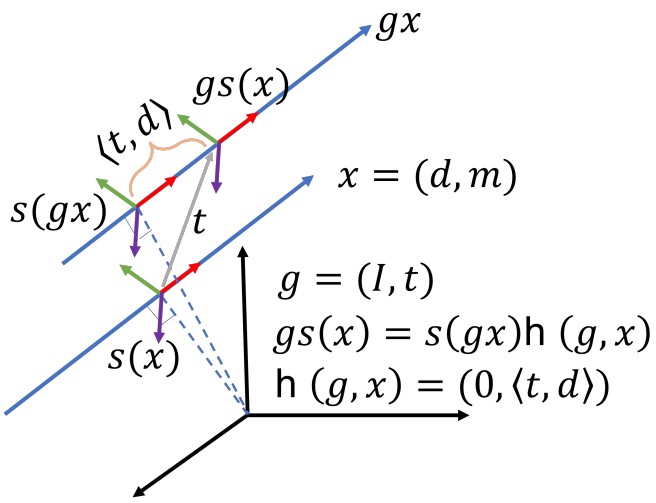

Figure 13: When we translate a ray $x = (\boldsymbol{d}, \boldsymbol{m})$ with $g = (I, \boldsymbol{t}) \in SE(3)$, we will find that $gs(x)$ does not agree with $s(gx)$. As defined in figure 12, we have $s_b(x) \perp \boldsymbol{d}$ and $s_b(gx) \perp \boldsymbol{d}$. Following the geometry of the figure, we obtain that $\mathsf{h}_{\mathsf{b}}(g, x) = \langle t, \boldsymbol{d} \rangle [0, 0, 1]^T$, i.e., $\mathsf{h}(g, x) = s(gx)^{-1}gs(x) = (I, \langle t, \boldsymbol{d} \rangle [0, 0, 1]^T) = (0, \langle t, \boldsymbol{d} \rangle)$.

$\rho : H \rightarrow GL(V)$. The group representation corresponds to the type of geometric quantity in the vector space $V$, for example, the scalar, the vector, or the higher-order tensor.

The quotient space $E = G \times_\rho V/H$ is defined through the right action of $H$ on $G \times V$: $(g, v)h = (gh, \rho(h)^{-1}v)$ for any $h \in H$, $g \in G$ and $v \in V$. With the defined projection map $p : G \times_\rho V \rightarrow G/H$: $p([g, v]) = gH$, where $[g, v] = \{(gh, \rho(h)^{-1}v) | h \in H\}$, the element in $G \times_\rho V$, we obtain the fiber bundle $E = G \times_\rho V$ associated to the principal bundle $G$. For more background and details of the associated vector bundle, we recommend referring to the following sources: [52] and [18].

The feature function $f : U \subseteq G/H \rightarrow V$ can encode the local section of the associated vector bundle $s_v : U \subseteq G/H \rightarrow G \times_\rho V$: $s_v(x) = [s(x), f(x)]$, where $s$ is the section map of the principal bundle as defined in Sec. A.2. The group $G$ acting on the field $f$ as shown in [18]:

$$(\mathcal{L}_g f)(x) = \rho(\mathsf{h}(g^{-1}, x))^{-1} f(g^{-1}x), \tag{6}$$

where $\mathsf{h} : G \times G/H \rightarrow H$ is the twist function as defined in Sec. A.2.

### A.4 Equivariant Convolution Over Homogeneous Space

The generalized equivariant convolution over homogeneous space, as stated in [18], that maps a feature field $f^{l_{in}}$ over homogeneous space $G/H_1$ to a feature $f^{l'_{out}}$ over homogeneous space $G/H_2$ by convolving with a kernel $\kappa$ is defined as:

$$f^{l'_{out}}(x) = \int_{G/H_1} \kappa(s_2(x)^{-1}y)\rho_{in}(\mathsf{h}_1(s_2(x)^{-1}s_1(y)))f^{l_{in}}(y)dy, \tag{7}$$

where $l_{in}$ and $l'_{out}$ [2] denote the input and output feature types, respectively. $\rho_{in}$ is the group representation of $H_1$ corresponding to the feature type $l_{in}$, $s_1$ is the section map from $G/H_1$ to $G$ (see Sec. A.2), $s_2$ is the section map from $G/H_2$ to $G$ (see Sec. A.2), $\mathsf{h}_1$ is the twist function corresponding to $s_1$ (see Sec. A.2).

The convolution is equivariant with respect to $G$, that is

$$\mathcal{L}_g^{out} f^{l'_{out}} = \kappa * \mathcal{L}_g^{in} f^{l_{in}},$$

---

[2] In this context, the feature type indicates the specific geometric quantity in vector spaces $V_{in}$ and $V_{out}$. $V_{in}$ corresponds to the stabilizer $H_1$ and $V_{out}$ corresponds to the stabilizer $H_2$. $H_1$ and $H_2$ can be distinct; therefore, to differentiate the types of features corresponding to different stabilizers, we utilize $l$ and $l'$ as notations for the feature types.

if and only if $\kappa(h_2 x) = \rho_{out}(h_2)\kappa(x)\rho_{in}(\mathrm{h}_1^{-1}(h_2, x))$ for any $h_2 \in H_2$, where $\rho_{out}$ is the group representation of $H_2$ corresponding to the feature type $l'_{out}$.

In the following examples, we will illustrate three instances where the input and output homogeneous spaces, denoted as $G/H_1$ and $G/H_2$, respectively, are identical, meaning that $H_1 = H_2$. These examples involve convolutions from $\mathbb{R}^3$ to $\mathbb{R}^3$, from $\mathbb{S}^2$ to $\mathbb{S}^2$, and from $\mathcal{R}$ to $\mathcal{R}$. Furthermore, we will show an example where $H_1$ and $H_2$ differ, explicitly focusing on the convolution from $\mathcal{R}$ to $\mathbb{R}^3$.

***Example*** 7. $SE(3)$ equivariant convolution from $\mathbb{R}^3$ to $\mathbb{R}^3$: If we use the section map as stated in Ex. 4, we will find that $\mathrm{h}(s(x)^{-1}s(y)) = I$, therefore convolution 7 becomes:

$$f^{l_{out}}(x) = \int_{\mathbb{R}^3} \kappa(s(x)^{-1}y)f^{l_{in}}(y)dy$$

$$= \int_{\mathbb{R}^3} \kappa(y - x)f^{l_{in}}(y)dy$$

and $\kappa$ should satisfy

$$\kappa(Rx) = \rho_{out}(R)\kappa(x)\rho_{in}(\mathrm{h}^{-1}(R, x))$$
$$= \rho_{out}(R)\kappa(x)\rho_{in}(\mathrm{h}^{-1}(R))$$
$$= \rho_{out}(R)\kappa(x)\rho_{in}^{-1}(R)$$

for any $R \in SO(3)$. When the feature type $l_{in}$ and $l_{out}$ corresponds to the irreducible representation, we have

$$\kappa(Rx) = D_{l_{out}}(R)\kappa(x)D_{l_{in}}(R)^{-1}$$

where $D^{l_{in}}$ and $D^{l_{out}}$ are the Wigner-D matrices, i.e. irreducible representations corresponding to the feature types $l_{in}$ and $l_{out}$, which is the same as the analytical result in [63].

***Example*** 8. $SO(3)$ equivariant spherical convolution from $\mathbb{S}^2$ to $\mathbb{S}^2$: For spherical convolution, when we substitute the section in Eq. 7 with the section we defined in Ex. 5, the convolution integral takes the following form:

$$f^{l_{out}}(\alpha, \beta)$$
$$= \int_{\alpha' \in [0, 2\pi), \beta' \in [0, \pi)} \kappa(R_Y^{-1}(\beta)R_Z^{-1}(\alpha)R_Z(\alpha')R_Y(\beta')[0, 0, 1]^T)$$
$$\rho_{in}(\mathrm{h}(R_Y^{-1}(\beta)R_Z^{-1}(\alpha)R_Z(\alpha')R_Y(\beta')))f^{l_{in}}(\alpha', \beta')d\alpha' \sin(\beta')d\beta'$$

where $[0, 0, 1]^T$ is the fixed original point as stated in Ex. 3, $\rho_{in}$ is the group representation of $SO(2)$ corresponding to the feature type $l_{in}$. When $\rho_{in}$ and $\rho_{out}$ are the irreducible representations of $SO(2)$, $\rho_{in}$ and $\rho_{out}$ can be denoted as $\rho_{in}(\theta) = e^{-il_{in}\theta}$ and $\rho_{out}(\theta) = e^{-il_{out}\theta}$.

To simplify the notation, we utilize $R(\theta)$ to represent $R_Z(\theta) \in SO(2)$, where $\theta \in [0, 2\pi)$. When considering the cases where $x = [0, 0, 1]^T$, $h(R(\theta)x) = R(\theta)$; when $x = -[0, 0, 1]^T$, $h(R(\theta)x) = R(-\theta)$; and when $x \in \mathbb{S}^2 - \{[0, 0, 1]^T, -[0, 0, 1]^T\}$, $h(R(\theta)x) = R(-\theta) = I$. Therefore, the kernel $\kappa$ should satisfy the following conditions: $\kappa(R(\theta)x) = e^{-il_{out}\theta}\kappa(x)$ for any $R(\theta) \in SO(2)$ and any $x \in \mathbb{S}^2 - \{[0, 0, 1]^T, -[0, 0, 1]^T\}$; $\kappa(x) = e^{-i(l_{out}-l_{in})\theta}\kappa(x)$ for $x = [0, 0, 1]^T$; and $\kappa(x) = e^{-i(l_{out}+l_{in})\theta}\kappa(x)$ for $x = -[0, 0, 1]^T$.

Specifically, when the input and output are scalar feature fields over the sphere, convolution reads

$$f^{out}(\alpha, \beta)$$
$$= \int_{\alpha' \in [0, 2\pi), \beta' \in [0, \pi)} \kappa(R_Y^{-1}(\beta)R_Z^{-1}(\alpha)R_Z(\alpha')R_Y(\beta')\eta)$$
$$f^{in}(\alpha', \beta')d\alpha' \sin(\beta')d\beta'$$

$\kappa$ has such constraint:

$$\kappa(R(\theta)x) = \kappa(x)$$

for any $R(\theta) \in SO(2)$, which is consistent with the isotropic kernel of the convolution in [22].

***Example* 9.** $SE(3)$ equivariant convolution from $\mathcal{R}$ to $\mathcal{R}$: In our case, the equivariant convolution from ray space to ray space is also based on the generalized equivariant convolution over a homogeneous space. See Sec. 3.4.1 for the details. We solve the constraint of the kernel here:

$$\kappa(hx) = \rho_{out}(h)\kappa(x)\rho_{in}(\mathrm{h}^{-1}(h,x)), \tag{8}$$

for any $h \in SO(2) \times \mathbb{R}$.

The irreducible group representation $\rho_{in}$ for the corresponding feature type $l_{in} = (\omega_{in}^1, \omega_{in}^2)$, where $\omega_{in}^1 \in \mathbb{N}$ and $\omega_{in}^2 \in \mathbb{R}$, can be written as $\rho_{in}(\gamma, t) = e^{-i(\omega_{in}^1 \gamma + \omega_{in}^2 t)}$ for any $h = (\gamma, t) \in SO(2) \times \mathbb{R}$; and the irreducible group representation $\rho_{out}(\gamma, t) = e^{-i(\omega_{out}^1 \gamma + \omega_{out}^2 t)}$ for the feature type $l_{out} = (\omega_{out}^1, \omega_{out}^2)$, where $\omega_{out}^1 \in \mathbb{N}$ and $\omega_{out}^2 \in \mathbb{R}$, for any $h = (\gamma, t) \in SO(2) \times \mathbb{R}$.

To simplify the notation, we utilize $R(\gamma)$ to represent $R_Z(\gamma) \in SO(2)$, where $\gamma \in [0, 2\pi)$. For any $h = (\gamma, t) \in SO(2) \times \mathbb{R}$ and any $x = (\boldsymbol{d}_x, \boldsymbol{m}_x) \in \mathcal{R}$, we have $\mathrm{h}(h, x) = s(hx)^{-1}hs(x) = (R_Z(R(\gamma), \boldsymbol{d}_x), \langle t[0,0,1]^T, \boldsymbol{d}_x \rangle)$ according to Ex. 6. Since $SO(2) \times \mathbb{R}$ is a product group, we can have $\kappa(x) = \kappa_1(x)\kappa_2(x)$, where

$$\kappa_1((\gamma, t)x) = \rho_{out}((\gamma, 0))\kappa_1(x)\rho_{in}^{-1}((R_Z(R(\gamma), \boldsymbol{d}_x), 0)) \tag{9}$$

$$\kappa_2((\gamma, t)x) = \rho_{out}((0, t))\kappa_2(x)\rho_{in}^{-1}((0, \langle t[0,0,1]^T, \boldsymbol{d}_x \rangle)) \tag{10}$$

Now we solve the constraint for the kernel $\kappa_1$:

One can check that for any $\boldsymbol{d}_x \in \mathbb{S}^2 - \{[0,0,1]^T, -[0,0,1]^T\}$, $R_Z(R(\gamma), \boldsymbol{d}_x) = I$; when $\boldsymbol{d}_x = [0,0,1]^T$, $R_Z(R(\gamma), \boldsymbol{d}_x) = R(\gamma)$; and when $\boldsymbol{d}_x = -[0,0,1]^T$, $R_Z(R(\gamma), \boldsymbol{d}_x) = R(-\gamma)$.

Therefore, we obtain the constraint that

$$\kappa_1((\gamma, t)x) = e^{-i\omega_{out}^1 \gamma}\kappa_1(x) \tag{11}$$

when $\boldsymbol{d}_x \in \mathbb{S}^2 - \{[0,0,1]^T, -[0,0,1]^T\}$;

$$\kappa_1((\gamma, t)x) = e^{-i(\omega_{out}^1 - \omega_{in}^1)\gamma}\kappa_1(x) \tag{12}$$

when $\boldsymbol{d}_x = [0,0,1]^T$;

$$\kappa_1((\gamma, t)x) = e^{-i(\omega_{out}^1 + \omega_{in}^1)\gamma}\kappa_1(x) \tag{13}$$

when $\boldsymbol{d}_x = -[0,0,1]^T$;

The solution for Eq. 11 is that $\kappa_1(x) = f(d(\eta, x), \angle([0,0,1]^T, \boldsymbol{d}_x))e^{-i\omega_{out}^1 atan2([0,1,0]\boldsymbol{d}_x, [1,0,0]\boldsymbol{d}_x)}$, where $atan2$ is the 2-argument arctangent function, and $f$ is an arbitrary function that maps $(d(\eta, x), \angle([0,0,1]^T, \boldsymbol{d}_x))$ to the complex domain.

The solution for Eq. 12 is that when $\omega_{out}^1 = \omega_{in}^1$, $\kappa_1(x) = C$, where $C$ is any constant value; when $\omega_{out}^1 \neq \omega_{in}^1$ and $x = \eta$, $\kappa_1(x) = 0$; when $\omega_{out}^1 \neq \omega_{in}^1$ and $x \neq \eta$, $\kappa_1(x) = f(d(\eta, x))e^{-i(\omega_{out}^1 - \omega_{in}^1)atan2([0,1,0]\boldsymbol{m}_x, [1,0,0]\boldsymbol{m}_x)}$, where $f$ is an arbitrary function that maps $d(x, \eta)$ to the complex domain.

The solution for Eq. 13 is that when $\omega_{out}^1 = -\omega_{in}^1$, $\kappa_1(x) = C$, where $C$ is any constant value; when $\omega_{out}^1 \neq -\omega_{in}^1$ and $x = -\eta$, $\kappa_1(x) = 0$; when $\omega_{out}^1 \neq -\omega_{in}^1$ and $x \neq -\eta$, $\kappa_1(x) = f(d(\eta, x))e^{-i(\omega_{out}^1 + \omega_{in}^1)atan2([0,1,0]^T\boldsymbol{m}_x, [1,0,0]^T\boldsymbol{m}_x)}$, where $f$ is an arbitrary function that maps $d(x, \eta)$ to the complex domain.

Next, we will solve the constraint for the kernel $\kappa_2$, which is that $\kappa_2((\gamma, t)x) = e^{-i(\omega_{out}^2 - \omega_{in}^2 \langle [0,0,1]^T, \boldsymbol{d}_x \rangle)t}\kappa_2(x)$.

When $\boldsymbol{d}_x = [0,0,1]^T$, and $\omega_{out}^2 \neq \omega_{in}^2$, $\kappa_2(x) = 0$; When $\boldsymbol{d}_x = -[0,0,1]^T$ and $\omega_{out}^2 \neq -\omega_{in}^2$, $\kappa_2(x) = 0$; When $\boldsymbol{d}_x = [0,0,1]^T$, and $\omega_{out}^2 = \omega_{in}^2$, $\kappa_2(x) = f(d(x, \eta))$, where $f$ is an arbitrary

function that maps $d(x, \eta)$ to the complex domain; When $\boldsymbol{d}_x = -[0, 0, 1]^T$, and $\omega_{out}^2 = -\omega_{in}^2$, $\kappa_2(x) = f(d(x, \eta))$, where $f$ is an arbitrary function that maps $d(x, \eta)$ to the complex domain; when $\boldsymbol{d}_x \in \mathbb{S}^2 - \{[0, 0, 1]^T, -[0, 0, 1]^T\}$,

$$\kappa_2(x) = f(d(\eta, x), \angle([0, 0, 1]^T, \boldsymbol{d}_x))e^{-i(\omega_{out}^2 - \omega_{in}^2\langle[0,0,1]^T, \boldsymbol{d}_x\rangle)g(x)}, \tag{14}$$

where $f$ is an arbitrary function that maps $(d(\eta, x), \angle([0, 0, 1]^T, \boldsymbol{d}_x))$ to the complex domain; $g(x) = [0, 0, 1](\boldsymbol{x}_Q - [0, 0, 0]^T)$, where $\boldsymbol{x}_Q$ represents the 3D coordinates of a point $Q$. This point $Q$ can be defined as the intersection of $x$ and $\eta$ if $x$ and $\eta$ intersect. Alternatively, if $x$ and $\eta$ do not intersect, $Q$ is determined as the intersection of $\eta$ and the ray $y$, which is perpendicular to both $x$ and $\eta$, and intersects with both $x$ and $\eta$. Refer to Figure 16 for a visual representation. One can easily check that $g((\gamma, t)x) = t + g(x)$, as shown in figure 16, which makes the solution valid.

If $x$ and $\eta$ are intersected, i.e., $[0, 0, 1]\boldsymbol{m}_x = 0$,

$$g(x) = [0, 0, 1](\boldsymbol{d}_x \times \boldsymbol{m}_x - \frac{[1, 0, 0](\boldsymbol{d}_x \times \boldsymbol{m}_x)}{[1, 0, 0]\boldsymbol{d}_x}\boldsymbol{d}_x)$$

when $[1, 0, 0]\boldsymbol{d}_x \neq 0$;

$$g(x) = [0, 0, 1](\boldsymbol{d}_x \times \boldsymbol{m}_x - \frac{[0, 1, 0](\boldsymbol{d}_x \times \boldsymbol{m}_x)}{[0, 1, 0]\boldsymbol{d}_x}\boldsymbol{d}_x)$$

when $[1, 0, 0]\boldsymbol{d}_x = 0$;

When $x$ and $\eta$ are not intersected,

$$g(x) = [0, 0, 1](\boldsymbol{d}_x \times \boldsymbol{m}_x - \frac{[1, 0, 0](\boldsymbol{d}_x \times \boldsymbol{m}_x)[1, 0, 0]\boldsymbol{d}_x + [0, 1, 0](\boldsymbol{d}_x \times \boldsymbol{m}_x)[0, 1, 0]\boldsymbol{d}_x}{([1, 0, 0]\boldsymbol{d}_x)^2 + ([0, 1, 0]\boldsymbol{d}_x)^2}\boldsymbol{d}_x).$$

**Regular Representation**    Here, we delve into the case where the output field type corresponds to the group representation of $SO(2) \times \mathbb{R}$ that $\rho(\gamma, t) = \rho_1(\gamma) \otimes \rho_2(t)$ for any $(\gamma, t) \in SO(2) \times \mathbb{R}$, where $\rho_2$ is the regular representation. The regular representation of a group G is a linear representation that arises from the group action of G on itself by translation, that is when $\rho_2 : \mathbb{R} \to GL(V)$ is the regular representation, for any $v \in V$, for any $t, t' \in \mathbb{R}$, we have $(\rho_2(t')v)_t = v_{t-t'}$, in other words, $v \in V$ can be viewed as a function defined on $\mathbb{R}$ or an infinite dimensional vector. Then according to Ex. 6, the group $SE(3)$ acting on the the field $f$ would be:

$$\begin{aligned}
(\mathcal{L}_g f)(x)_t &= (\rho(\text{h}(g^{-1}, x))^{-1}f(g^{-1}x))_t \\
&= \rho_1(\text{h}_a(R_{g^{-1}}, \boldsymbol{d}_x))^{-1}f(g^{-1}x)_{t+\text{h}_b(g^{-1}, x)} \\
&= \rho_1(R_Z(R_{g^{-1}}, \boldsymbol{d}_x))^{-1}f(g^{-1}x)_{t+\langle\boldsymbol{t}_{g^{-1}}, (R_{g^{-1}}\boldsymbol{d}_x)\rangle}
\end{aligned}$$

for any $t \in \mathbb{R}$, $x \in \mathcal{R}$ and $g \in SE(3)$.

The points $\boldsymbol{x}$ on the ray $x = (\boldsymbol{d}_x, \boldsymbol{m}_x)$ can be uniquely expressed as $\boldsymbol{x} = s_b(x) + t_{\boldsymbol{x}}\boldsymbol{d}_x = \boldsymbol{d}_x \times \boldsymbol{m}_x + t_{\boldsymbol{x}}\boldsymbol{d}_x$, therefore for any $x \in \mathcal{R}$, any $t \in \mathbb{R}$, $f(x)_t$ can be expressed as a feature attached to the point $s_b(x) + t\boldsymbol{d}_x$ along the ray $x$,i.e., $f(x)_t = f'(s_b(x) + t\boldsymbol{d}_x, \boldsymbol{d}_x)$ as shown in figure 14.

Therefore, we have $f'(\boldsymbol{x}, \boldsymbol{d}) = f((\boldsymbol{d}, \boldsymbol{x} \times \boldsymbol{d}))_{\langle\boldsymbol{x} - \boldsymbol{d}\times(\boldsymbol{x}\times\boldsymbol{d}), \boldsymbol{d}\rangle}$, one can easily check:

$$(\mathcal{L}_g f')(\boldsymbol{x}, \boldsymbol{d}) = \rho_1(\text{h}_a(R_{g^{-1}}, \boldsymbol{d}))^{-1}f'(R_{g^{-1}}\boldsymbol{x} + \boldsymbol{t}_{g^{-1}}, R_{g^{-1}}\boldsymbol{d}) = \rho_1(R_Z(R_{g^{-1}}, \boldsymbol{d}))^{-1}f'(g^{-1}\boldsymbol{x}, R_{g^{-1}}\boldsymbol{d}) \tag{15}$$

We should note the difference of the point $\boldsymbol{x}$ along the ray and the independent point $\boldsymbol{x}$, as shown in the above equation, the point $\boldsymbol{x}$ along the ray $x = (\boldsymbol{d}, \boldsymbol{x} \times \boldsymbol{d})$ is denoted as $(\boldsymbol{x}, \boldsymbol{d})$ instead of $\boldsymbol{x}$. Actually, it can be viewed as a homogeneous space of $SE(3)$ larger than $\mathbb{R}^3$, whose elements are in $\mathbb{R}^3 \times \mathbb{S}^2$, as shown in figure 15.

To summarize, the features attached to the ray, whose type corresponds to the regular representation of translation, can be considered as the features attached to the points along the ray. The action of $SE(3)$ on features attached to these points can be expressed as shown in Eq. 15.

The solution $\kappa$ also can be expressed as

$$\kappa(x)_t = \kappa_1(x)\kappa_2(x)_t \tag{16}$$

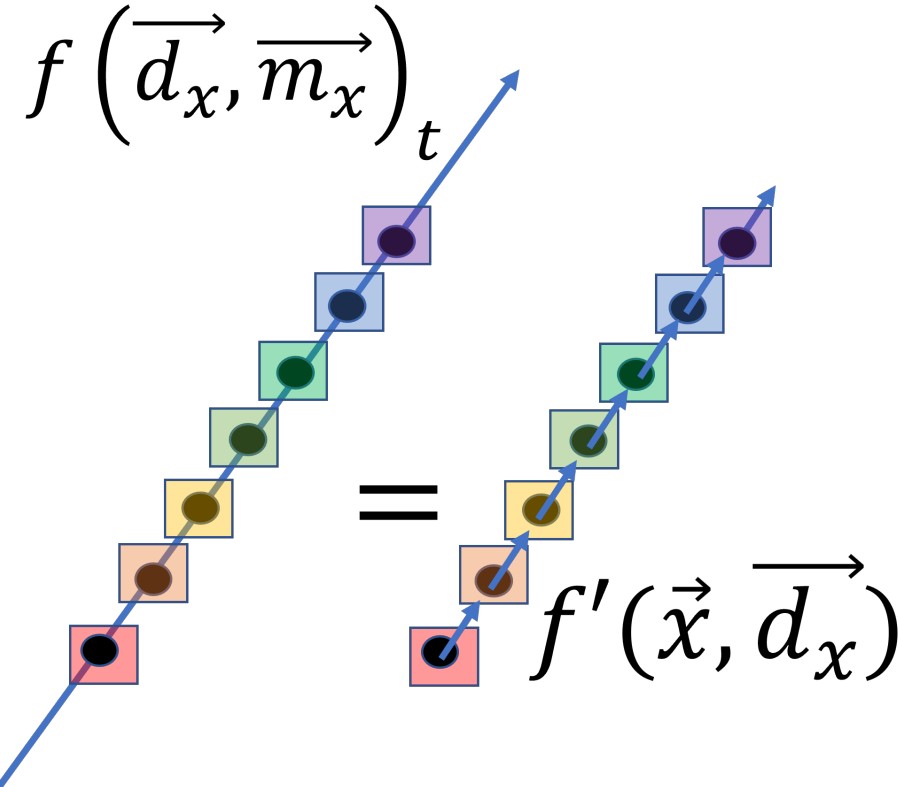

Figure 14: The feature attached to the ray, which corresponds to the regular representation of translation, can also be treated as the features attached to the points along the ray.

for any $t \in \mathbb{R}$, and their constraint is also the same as Eq. 9 and Eq. 10. As a result, the solution for $\kappa_1$ should be the same. We only need to solve $\kappa_2$:

$$\kappa_2((\gamma, t')x)_t = e^{i\omega_{in}^2 \langle [0,0,1]^T, \boldsymbol{d}_x \rangle t'} \kappa_2(x)_{t-t'} \tag{17}$$

for any $(\gamma, t') \in SO(2) \times \mathbb{R}$.

When $\boldsymbol{d}_x \in \mathbb{S}^2 - \left\{ [0,0,1]^T, -[0,0,1]^T \right\}$,

$$\kappa_2(x)_t = f(d(\eta, x), \angle([0,0,1]^T, \boldsymbol{d}_x)) e^{i\omega_{in}^2 \langle [0,0,1]^T, \boldsymbol{d}_x \rangle g(x)} \delta(t - g(x)), \tag{18}$$

where $f$ and $g$ are the same function as defined in 14, and $\delta(t) = 1$ only when $t = 0$.

when $\boldsymbol{d}_x \in \left\{ [0,0,1]^T, -[0,0,1]^T \right\}$, $\kappa_2(x)_t = 0$ for any $t \in \mathbb{R}$.

***Example*** 10. $SE(3)$ equivariant convolution from $\mathcal{R}$ to $\mathbb{R}^3$: Following [18], the convolution from rays to points becomes:

$$f_2^{l_{out}}(x) = \int_{\mathcal{R}} \kappa(s_2(x)^{-1}y)\rho_{in}(\mathrm{h}_1(s_2(x)^{-1}s_1(y)))f_1^{l_{in}}(y)dy, \tag{19}$$

where $\mathrm{h}_1$ is the twist function corresponding to section $s_1 : \mathcal{R} \to SE(3)$ defined aforementioned, $\rho_{in}$ is the group representation of $SO(2) \times \mathbb{R}$, corresponding to the feature type $l_{in}$, $s_2 : \mathbb{R}^3 \to SE(3)$ is the section map defined in paper as $s_2(\boldsymbol{x}) = (I, \boldsymbol{x})$.

In this paper, we give the analysis and solutions for the kernel where the input is the scalar field over the ray space, i.e.,$\rho_{in} = 1$, the trivial group representation, which is also the case of our application in reconstruction.

The convolution is equivariant if and only if

$$\kappa(h_2 x) = \rho_{out}(h_2)\kappa(x),$$

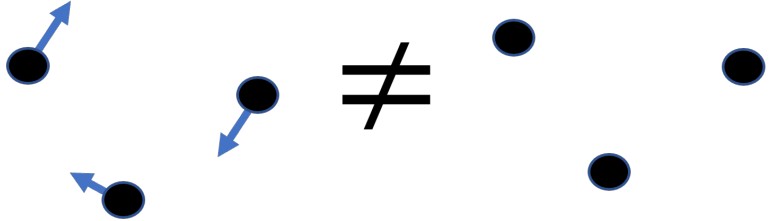

vector features(type-1)  vector features(type-1)

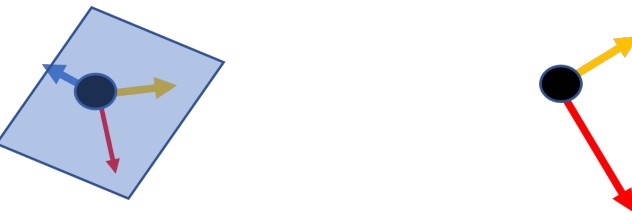

Figure 15: As shown in the figure, the point along the ray is distinct from the independent point. Moreover, we can observe that the type-1 feature of the point along the ray differs from that of the independent point. Specifically, the type-1 feature for the point along the ray can be interpreted as a vector on the plane orthogonal to the ray direction. In contrast, the type-1 feature for the independent point can be interpreted as a three-dimensional vector.

for any $h_2 \in SO(3)$, where $\rho_{out}$ is the group representation of $SO(3)$ corresponding to the feature type $l_{out}$.

We can derive $\kappa(h_2 x) = \rho_{out}(h_2)\kappa(x)$ analytically. For irreducible representation $\rho_{out}$ and any $x = (\boldsymbol{d}_x, \boldsymbol{m}_x) \in \mathcal{R}$, if $\|\boldsymbol{m}_x\| = 0$, $\kappa(x) = cY^{l_{out}}(\boldsymbol{d}_x)$, where $c$ is an arbitrary constant and $Y^{l_{out}}$ is the spherical harmonics and $l_{out}$ is the order (type) of output tensor corresponding to the representation $\rho_{out}$; With $\|\boldsymbol{m}_x\| \neq 0$, $\kappa(x)$ becomes $\rho_{out}(\hat{x})f(\|\boldsymbol{m}\|_x)$, where $\hat{x}$ denotes the element $(\boldsymbol{d}_x, \frac{\boldsymbol{m}_x}{\|\boldsymbol{m}_x\|}, \boldsymbol{d}_x \times \frac{\boldsymbol{m}_x}{\|\boldsymbol{m}_x\|})$ in $SO(3)$ and $f : \mathbb{R} \to \mathbb{R}^{(2l_{out}+1)\times 1}$.

Similar to the convolution from rays to rays, we also can have the local support of the kernel. We set $\kappa(x) \neq 0$ when $\|\boldsymbol{m}_x\| \leq d_0$, otherwise $\kappa(x) = 0$. One can easily check that it doesn't break the equivariant constraint for the kernel.

Specifically, when we set $d_0 = 0$, the neighborhood of the target points in the convolution only includes the rays from all views going through the point. Hence, we can simplify the convolution to $f_2^{l_{out}}(x) = \int_{d(y,x)=0} Y^{l_{out}}(\boldsymbol{d}_{s_2(x)^{-1}y})f_1^{in}(y)dy$. This equation shows that for every point $x$, we can treat the ray $y$ going through $x$ with feature $f_1^{in}$ as a point $y'$, where $y' - x = \boldsymbol{d}_{s_2(x)^{-1}y}$, as shown in figure 17.

## B  Equivariant 3D Reconstruction

### B.1  Approximation of the Equivariant Convolution from Rays to Rays

In practical $3D$ reconstruction, we have multiple views instead of the whole light field. Although the convolution above is defined on the continuous ray space, the equivariance still strictly holds when the ray sampling (pixels from camera views) is the same up to coordinate change. In this case, we will show how we adjust the equivariant convolution from rays to rays and approximate it by an intra-view $SE(2)$-convolution.

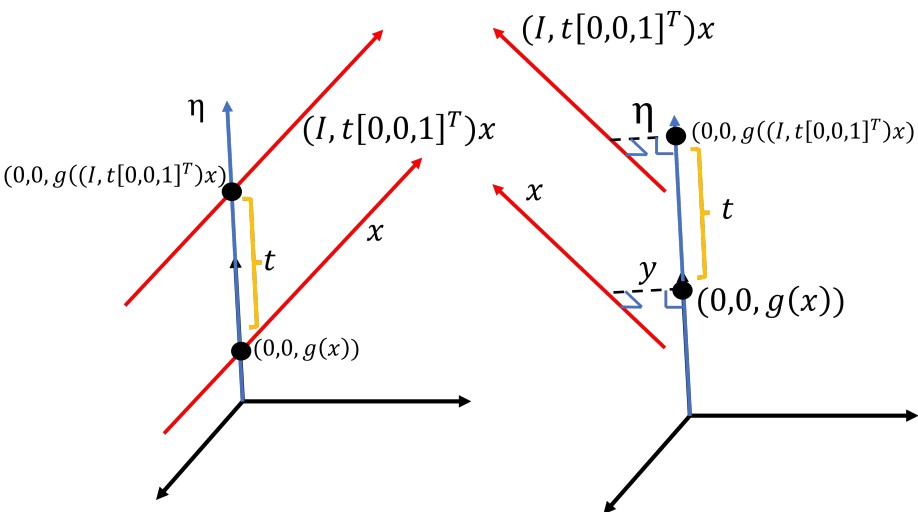

Figure 16: Visualization of $g(x)$. The left is the case that the ray $x$ and the ray $\eta$ are intersected, and the right is the case that the ray $x$ and the ray $\eta$ are not intersected. For the left, the point $Q$ is the intersection of $x$ and $\eta$, and $Q = (0, 0, g(x))$; for the right, the point $Q$ is the intersection of the line $y$ and the ray $\eta$, where $y$ is perpendicular to both $\eta$ and $x$, and intersects with both $\eta$ and $x$. From the figure, in both cases, we can see that for any $t \in \mathbb{R}$, $g((0, t)x) = t + g(x)$. In general, we actually have for any $(\gamma, t) \in SO(2) \times \mathbb{R}$, $g((\gamma, t)x) = t + g(x)$.

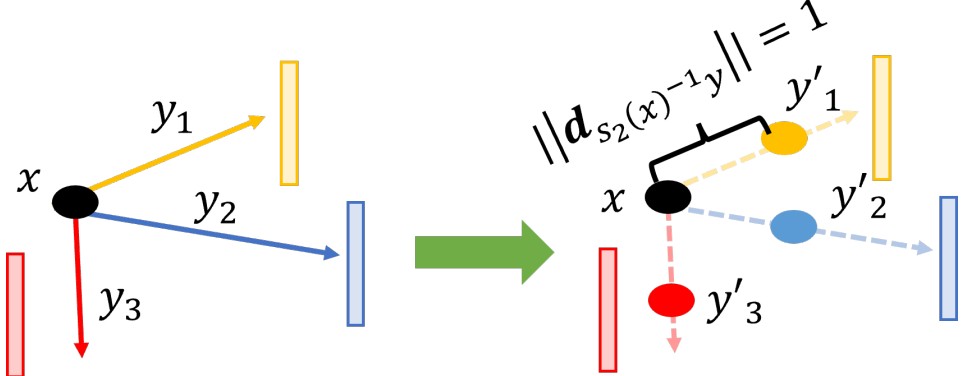

Figure 17: Interpreting rays $y_i$ as points $y_i'$

### B.1.1 From Light Field to Intra-view Convolution

Following Fig. 18, neighboring rays are composed of two parts: a set of rays from the same view and another set of rays from different views. For one ray $x$ in view $A$, the neighboring rays from view $B$ are in the neighborhood of the epipolar line of $x$ in view $B$. When the two views are close, the neighborhood in the view $B$ would be very large.

The kernel solution in Ex. 9 suggests that $\kappa(x)$ is related to $\angle(\boldsymbol{d}_x, [0, 0, 1]^T)$ and $d((x, \eta))$, where $\eta = ([0, 0, 1]^T, [0, 0, 0]^T)$ as mentioned before. It would be memory- and time-consuming to memorize the two metrics beforehand or to compute the angles and distances on the fly. Practically, the light field is only sampled from a few sparse viewpoints, which causes the relative angles of the rays in different views to be large and allows them to be excluded from the kernel neighborhood; therefore, in our implementation, the ray neighborhood is composed of only rays in the same view.

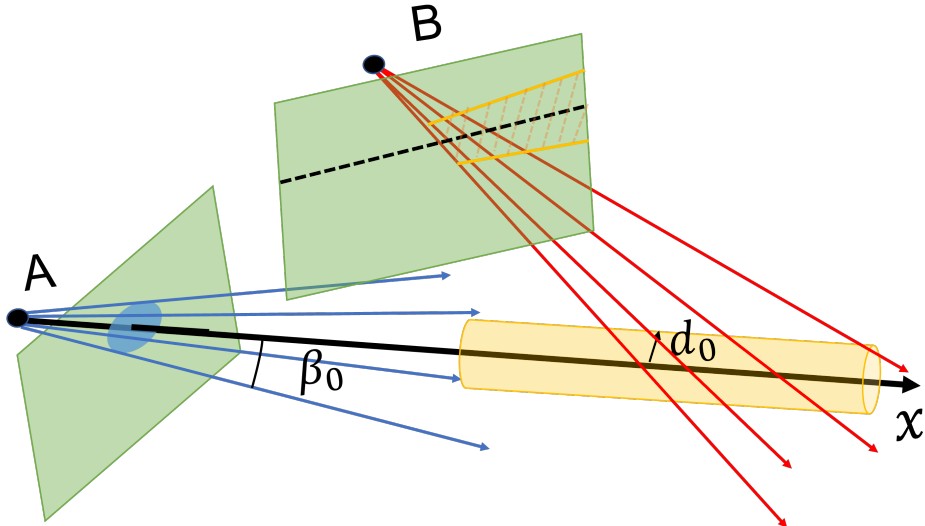

Figure 18: For simplification, we show a situation of two views. For a ray $x$ from view A, one part of the neighboring rays is from view A (the blue rays in the figure), $\mathcal{N}_A(x)$. For any ray $y \in \mathcal{N}_A(x)$, we have $d(y, x) = 0$, and we require $\angle(\boldsymbol{d}_y, \boldsymbol{d}_x) \leq \beta_0$. The other part is from the other view B (the red rays in the figure). As illustrated in figure 5, the neighboring rays always cross a cylinder around $x$; therefore, the neighboring rays from view B are the projection of the cylinder with radius $r = d_0$ in view B, that is, $\mathcal{N}_B$ is composed of the neighboring pixels of the epipolar line (the black dotted dash) corresponding to $x$ in view B. For any ray $y$ in the projection of the cylinder, we have $d(y, x) \neq d_0$. Since we require that $\angle(\boldsymbol{d}_y, \boldsymbol{d}_x) \leq \beta_0$ for any ray $y \in \mathcal{N}_B(x)$, $\mathcal{N}_B(x)$ is part of the projection of the cylinder, denoted as the shaded yellow part in view B.

### B.1.2 From Intra-view Light Field to Spherical Convolution

After showing that a small kernel support in the case of sparse views affects only intra-view rays, we can prove that an intra-view light-field convolution is equivalent to a spherical convolution when we constrain the feature field types over $\mathcal{R}$.

We exploit the desired property that a feature defined on a ray is constant along the ray. This means that the translation part of the stabilizer group (translation along the ray) leaves the feature as is. In math terms, the irreducible representation for the translation $\mathbb{R}$ is the identity, which means that the field function is a scalar field for the translation group, with the formula $(\mathcal{L}_t f)(x) = f(t^{-1}x)$. We prove that, in this case, the intra-view convolution over rays is equivalent to the spherical convolution; please see Sec. C.

### B.1.3 From SO(3)- to SE(2)-convolution

While there is an established framework for spherical convolution using a Fourier transform [17, 22, 24] it is not applicable in our case because the boundaries of the constrained field of view cause an explosion in the high frequencies of the spherical harmonics. We will make a compromise here and approximate the SO(3) convolution with an SE(2) convolution on the image plane by making the assumption that the field of view is small. One can see the rationale behind this approximation by keeping only the first order terms in the optical flow equation: the rotational term is only due to $\Omega_z$ while the translational term is $(-T_x - \Omega_y, -T_y + \Omega_x)$ with $(\Omega_x, \Omega_y, \Omega_z)$ as the angular velocity. We provide a justification using the formalism of the previous paragraphs in appendix Sec. E.

### B.2 Ray Fusion: Equivariant Convolution and Transformer

To reconstruct a 3D object, we use an implicit function known as the signed distance function (SDF) defined on $\mathbb{R}^3$. As a result, we require an equivariant model that can transform features from rays to

points to obtain the SDF. This can be achieved using the equivariant convolution in Sec. 3.4.2 and transformer in Sec. 3.2 in the paper 3.5, which allows us to transform features from the ray space to points in 3D space while maintaining equivariance.

### B.2.1 Equivariant Convolution from Rays to Points

In this paper, we obtain the scalar feature field over rays after the SE(2)-equivariant CNNs. As illustrated in figure 3 , we utilize the equivariant convolution (discussed in Sec. 3.4.2 ) to compute features for a query point by convolving over neighboring rays. Our experiments have shown that convolving only over rays that go through the point achieves the best results, and the equivariant kernel used for this convolution is provided in Ex.10. Moreover, in the implementation, we can concatenate the input feature $f_1^{in}$ with the depth embedding of the query point $x$. While this theoretically breaks the ideal equivariance for continuous light fields, it does not affect the practical equivariance, as it is rare for two cameras to share the same ray.

### B.2.2 Equivariant Transformer from Rays to Points

For the third step, we introduce an equivariant transformer to alleviate the loss of expressivity due to the constrained kernel $\kappa$ in Eq. 19. Again, the attention key and values are generated from the feature attached to rays, while the query is generated from the feature attached to points.

In the implementation, we apply a transformer over the rays going through the query point. We can continue to use the interpretation that treats any ray $y$ passing through the point $x$ as a point $y'$ such that $y' - x = \boldsymbol{d}_{s_2(x)^{-1}y}$, as shown in figure 17. Since $y$ becomes point $y'$, the ray feature $f_1^{in}$ becomes the feature over $\mathbb{R}^3$ attached to "points" $y'$. We can update the neighboring ray feature by directly concatenating the equivariant feature of the point to every ray feature before through a $SO(3)$ equivariant MLP. The transformer in Eq. 4 would be converted to the transformer in [27] over $\mathbb{R}^3$. See appendix Sec. H for details. The composition of the ray updating block and transformer block are shown in figure 22.

## C Proof of Equivalence of Intra-view Light Field Convolution and Spherical Convolution

The property that a feature defined on a ray is constant along the ray means that the translation part of the stabilizer group (translation along the ray) leaves the feature as is. In math terms, the irreducible representation for the translation $\mathbb{R}$ is the identity, which means that the field function is a scalar field for the translation group, with the formula $(\mathcal{L}_t f)(x) = f(t^{-1}x)$. The equivariant condition on the kernel can then be simplified as

$$\kappa((h,t)x) = \rho_{out}(h)\kappa(x)\rho_{in}(\mathrm{h}_a^{-1}(h, \boldsymbol{d}_x)),$$

where $h \in SO(2)$ and $t \in \mathbb{R}$, $\rho_{in}$ and $\rho_{out}$ are irreducible representations for $SO(2)$, and $\mathrm{h}_a$ is the twist function as shown in Ex. 6 that $\mathrm{h}(g,x) = (\mathrm{h}_a(R_g, \boldsymbol{d}_x), \mathrm{h}_b(g,x))$,i.e., the twist of the fiber introduced by action of $SO(3)$ corresponding to the section map $s_a$ of $SO(3)$ in Ex. 5 and Ex. 6. Now we describe the relationship between the intra-view light-field convolution and the spherical convolution:

**Proposition C.1.** *When the translation group acts on feature $f : \mathcal{R} \to V$ as $(\mathcal{L}_t f)(x) = f(t^{-1}x)$ for any $x \in \mathcal{R}$, the equivariant intra-view light-field convolution:*

$$f^{l_{out}}(x) = \int_{y \in \mathcal{N}(x)} \kappa(s(x)^{-1}y)\rho_{in}(h(s(x)^{-1}s(y)))f^{l_{in}}(y)dy$$

*becomes a spherical convolution:*

$$f^{l_{out}}(x) = \int_{\boldsymbol{d}_y \in \mathbb{S}^2} \kappa'(s_a(\boldsymbol{d}_x)^{-1}\boldsymbol{d}_y)\rho_{in}(h_a(s_a(\boldsymbol{d}_x)^{-1}s_a(\boldsymbol{d}_y)))$$
$$f'^{l_{in}}(\boldsymbol{d}_y)d\boldsymbol{d}_y, \tag{20}$$

*where $f'^{l_{in}}(\boldsymbol{d}_y) = f^{l_{in}}(\boldsymbol{d}_y, \boldsymbol{c}_x \times \boldsymbol{d}_y)$, $\boldsymbol{c}_x$ denotes the camera center that $x$ goes through, $s_a$ is the section map of $SO(3)$ as defined in appendix Ex. 5, and $\kappa'(s_a(\boldsymbol{d}_x)^{-1}\boldsymbol{d}_y) = \kappa(s_a(\boldsymbol{d}_x)^{-1}\boldsymbol{d}_y, (s(x)^{-1}\boldsymbol{x}_c) \times (s_a(\boldsymbol{d}_x)^{-1}\boldsymbol{d}_y))$.*

*Proof.* The $SE(3)$ equivariant convolution over rays transforms into intra-view convolution when the neighboring lights are in the same view. Moreover, the simplified kernel constraint derived in the paper is that for any $(h, t) \in SO(2) \times \mathbb{R}$ and $x = (\boldsymbol{d}_x, \boldsymbol{m}_x) \in \mathcal{R}$:

$$\kappa((h, t)x) = \rho_{out}(h)\kappa(x)\rho_{in}(\text{h}_a^{-1}(h, \boldsymbol{d}_x)),$$

where $\text{h}_a : SO(3) \times \mathbb{S}^2 \to SO(2)$ is the twist function: $\text{h}_a(g, \boldsymbol{d}) = s_a(g\boldsymbol{d})^{-1}gs_a(\boldsymbol{d})$ for any $g \in SO(3)$ and $\boldsymbol{d} \in \mathbb{S}^2$.

With the simplified kernel constraint, we can prove that intra-view light field convolution is equivalent to spherical convolution:

$$
\begin{aligned}
&f^{l_{out}}(x) \\
&= \int_{d(y, \boldsymbol{c}_x)=0} \kappa(s(x)^{-1}y)\rho_{in}(\text{h}(s(x)^{-1}s(y)))f^{l_{in}}(y)dy && (21) \\
&= \int_{d(y, \boldsymbol{c}_x)=0} \kappa(s(x)^{-1}y)\rho_{in}(\text{h}_a(s_a(\boldsymbol{d}_x)^{-1}s_a(\boldsymbol{d}_y)))f^{l_{in}}(y)dy && (22) \\
&= \int_{\boldsymbol{d}_y \in \mathbb{S}^2} \kappa(s_a(\boldsymbol{d}_x)^{-1}\boldsymbol{d}_y, s(x)^{-1}\boldsymbol{x}_c \times (s_a(\boldsymbol{d}_x)^{-1}\boldsymbol{d}_y)) \\
&\qquad \rho_{in}(\text{h}_a(s_a(\boldsymbol{d}_x)^{-1}s_a(\boldsymbol{d}_y)))f^{l_{in}}(\boldsymbol{d}_y, \boldsymbol{c}_x \times \boldsymbol{d}_y)d\boldsymbol{d}_y && (23) \\
&= \int_{\boldsymbol{d}_y \in \mathbb{S}^2} \kappa'(s_a(\boldsymbol{d}(x))^{-1}\boldsymbol{d}_y)\rho_{in}(\text{h}_a(s_a(\boldsymbol{d}_x)^{-1}s_a(\boldsymbol{d}_y))) \\
&\qquad f'^{l_{in}}(\boldsymbol{d}_y)d\boldsymbol{d}_y. && (24)
\end{aligned}
$$

In line 21, $\boldsymbol{c}_x$ is the camera center that $x$ goes through.

The line 21 is equal to the line 22 because we assume that the irreducible representation for the translation $\mathbb{R}$ is the identity as mentioned in the paper.

From line 22 to line 23, We can replace $s(x)^{-1}y$ with

$$(s_a(\boldsymbol{d}_x)^{-1}\boldsymbol{d}_y, (s(x)^{-1}\boldsymbol{x}_c) \times (s_a(\boldsymbol{d}_x)^{-1}\boldsymbol{d}_y))$$

due to the facts that $s_a(\boldsymbol{d}_x)^{-1}\boldsymbol{d}_y = \boldsymbol{d}_{s(x)^{-1}y}$ and point $s(x)^{-1}\boldsymbol{x}_c$ is on the ray $s(x)^{-1}y$. Since $y$ goes through $\boldsymbol{c}_x$, we can replace $y$ with $(\boldsymbol{d}_y, \boldsymbol{c}_x \times \boldsymbol{d}_y)$.

From line 23 to 24, we have $f'^{l_{in}}(\boldsymbol{d}_y) = f^{l_{in}}(\boldsymbol{d}_y, \boldsymbol{c}_x \times \boldsymbol{d}_y)$ because $\boldsymbol{c}_x$ is fixed for any view. Additionally, from line 23 to 24 we replace

$$\kappa(s_a(\boldsymbol{d}_x)^{-1}\boldsymbol{d}_y, (s(x)^{-1}\boldsymbol{x}_c) \times (s_a(\boldsymbol{d}_x)^{-1}\boldsymbol{d}_y))$$

with $\kappa'(s_a(\boldsymbol{d}_x)^{-1}\boldsymbol{d}_y)$. It is because according to

$$\kappa((h, t)x) = \rho_{out}(h)\kappa(x)\rho_{in}(\text{h}_a^{-1}(h, \boldsymbol{d}_x)),$$

we have $\kappa((e, t)x) = \kappa(x)$ for any $t \in \mathbb{R}$, where $e$ is the identity element in $SO(2)$; thus when $t = ((-s(x)^{-1}\boldsymbol{x}_c))^T[0, 0, 1]^T$, we have

$$
\begin{aligned}
&\kappa(s_a(x)^{-1}\boldsymbol{d}_y, s(x)^{-1}\boldsymbol{x}_c \times (s_a(x)^{-1}\boldsymbol{d}_y)) \\
&= \kappa(s_a(x)^{-1}\boldsymbol{d}_y, (s(x)^{-1}\boldsymbol{x}_c + t[0, 0, 1]^T) \times (s_a(x)^{-1}\boldsymbol{d}_y)) && (25) \\
&= \kappa((s_a(x)^{-1}\boldsymbol{d}_y, [0, 0, 0]^T) && (26) \\
&= \kappa'((s_a(x)^{-1}\boldsymbol{d}_y).
\end{aligned}
$$

Line 25 is equal to 26 because $s(x)^{-1}\boldsymbol{x}_c$ is always on the $z$ axis, and thus $s(x)^{-1}\boldsymbol{x}_c + t[0, 0, 1]^T = [0, 0, 0]^T$. $\square$

## D  Spherical Convolution Expressed in Gauge Equivariant Convolution Format

Group convolution is a particular case of gauge equivariant convolution [64], where gauge equivariant means the equivariance with respect to the transformation of the section map (transformation of the

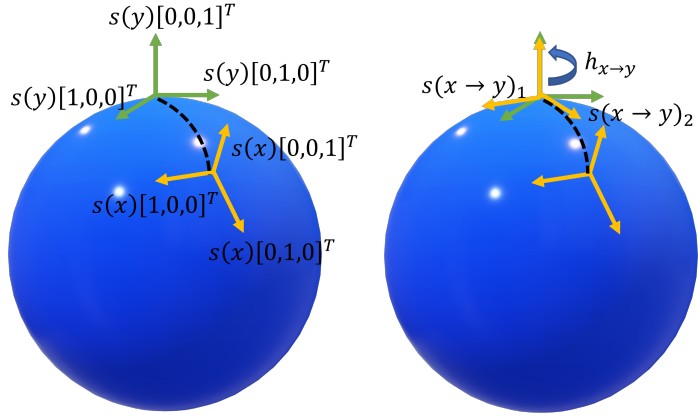

Figure 19: Illustration of $h_{x \to y}$. $s(x)[1,0,0]^T$ and $s(x)[0,1,0]^T$ (yellow) attached to $x$ are tangent vectors on $x$. We parallel transport $s(x)[1,0,0]^T$ and $s(x)[0,1,0]^T$ along the geodesic (black dashed line) between $x$ and $y$. The transported tangent vectors need to undergo a transformation $h_{x \to y}$ in $SO(2)$ to align with the vectors $s(y)[1,0,0]^T$ and $s(y)[0,1,0]^T$ (green) attached to $y$.

tangent frame). In the following paragraph we give the elaborated definition of gauge equivariance for the sphere.

Suppose $f : \mathbb{S}^2 \to V$ is the field function corresponding to the section choice $s_a : \mathbb{S}^2 \to SO(3)$, we use $\mathcal{L}_{s_a \to s_a'}$ acting on $f$ to denote the change of section map from $s_a$ to $s_a'$: $(\mathcal{L}_{s_a \to s_a'} f)(x) = \rho(s_a(x)^{-1} s_a'(x))^{-1} f(x)$, where $\rho$ is the irreducible representation of $SO(2)$ corresponding to the field type of $f$. The convolution $\Phi$ is gauge equivariant when $\Phi(\mathcal{L}_{s_a \to s_a'} f) = \mathcal{L}_{s_a \to s_a'}(\Phi(f))$.

In this section, we show that the spherical convolution can be expressed in terms of the gauge equivariant convolution [16], which provides the convenience for us to verify the approximation of spherical convolution through the $SE(2)$ convolution:

$$f^{l_{out}}(x) = \int_{y \in \mathcal{N}(x)} \kappa'(s(x)^{-1} y) \rho_{in}(h_{y \to x})^{-1} f^{l_{in}}(y) dy,$$

where $\kappa'(hx) = \rho_{out}(h) \kappa'(x) \rho_{in}^{-1}(h)$ for any $h \in SO(2)$.

Since the focus of this section's discussion is spherical convolution, here we use $s(x)$ to denote $s_a(x)$ for any $x \in \mathbb{S}^2$.

For any $x, y \in \mathbb{S}^2$, $s(x)[1,0,0]^T$, $s(x)[0,1,0]^T$ attached to $x$ are tangent vectors on $x$, we parallel transport $s(x)[1,0,0]^T$ and $s(x)[0,1,0]^T$ along the geodesic between $x$ and $y$ and get two tangent vectors on $y$, denoted as $s(x \to y)_1$ and $s(x \to y)_2$ as shown in the figure 19, where the parallel transport along a smooth curve is a way to translate a vector "parallelly" based on the affine connection, that is, for a smooth curve $\gamma : [0,1] \to \mathbb{S}^2$, the parallel transport $X : \text{Im}(\gamma) \to \mathcal{T}\mathbb{S}^2$ along the curve $\gamma$ satisfies that $\nabla_{\dot{\gamma}(t)} X = 0$, where $\text{Im}(\gamma) = \{\gamma(t) | t \in [0,1]\}$ and $\nabla$ is the affine connection.

$s(x \to y)_1$ and $s(x \to y)_2$ need to undergo a transformation in $SO(2)$ to align with $s(y)[1,0,0]^T$ and $s(y)[0,1,0]^T$ on y as shown in the figure 19. We denote the transformation as $h_{x \to y}$.

With the above notation, the spherical convolution can be expressed as:

$$f^{l_{out}}(x) = \int_{y \in \mathcal{N}(x)} \kappa(s(x)^{-1}y)\rho_{in}(\mathrm{h}(s(x)^{-1}s(y)))f^{l_{in}}(y)dy$$

$$= \int_{y \in \mathcal{N}(x)} \kappa(s(x)^{-1}y)\rho_{in}(h_{s(x)^{-1}y \to \eta})$$

$$\rho_{in}(h_{s(x)^{-1}y \to \eta})^{-1}\rho_{in}(\mathrm{h}(s(x)^{-1}s(y))f^{l_{in}}(y)dy$$

$$= \int_{y \in \mathcal{N}(x)} \kappa(s(x)^{-1}y)\rho_{in}(h_{s(x)^{-1}y \to \eta})$$

$$\rho_{in}(h_{y \to x})^{-1}f^{l_{in}}(y)dy$$

$$= \int_{y \in \mathcal{N}(x)} \kappa'(s(x)^{-1}y)\rho_{in}(h_{y \to x})^{-1}f^{l_{in}}(y)dy,$$

where $\eta = [0,0,1]^T$, the fixed origin point in $\mathbb{S}^2$, and $\kappa'(x) = \kappa(x)\rho_{in}(h_{x \to \eta})^{-1}$ for any $x \in \mathcal{N}(\eta)$.
We can derive the equivariant condition that $\kappa'$ should satisfy:

$$\kappa'(hx) = \kappa(hx)\rho_{in}(h_{hx \to \eta})^{-1}$$

$$= \rho_{out}(h)\kappa(x)\rho_{in}(\mathrm{h}(h,x))^{-1}\rho_{in}(h_{hx \to \eta})$$

$$= \rho_{out}(h)\kappa(x)\rho_{in}(h_{x \to \eta})^{-1}\rho_{in}(h^{-1})$$

$$= \rho_{out}(h)\kappa'(x)\rho_{in}^{-1}(h).$$

Therefore, the spherical convolution can be expressed as the gauge equivariant convolution format:

$$f^{l_{out}}(x) = \int_{y \in \mathcal{N}(x)} \kappa'(s(x)^{-1}y)\rho_{in}(h_{y \to x})^{-1}f^{l_{in}}(y)dy,$$

where $\kappa'(hx) = \rho_{out}(h)\kappa'(x)\rho_{in}^{-1}(h)$ for any $h \in SO(2)$.

# E Converting Spherical Convolution to $SE(2)$ Equivariant Convolution

As stated in Sec. D, spherical convolution is gauge equivariant with respect to the choice of section map $s_a$, and the spherical convolution can be written as gauge equivariant convolution. In this section, we use the gauge equivariant convolution to analyze the $SE(2)$ equivariant convolution's approximation of spherical convolution.

Since each view performs spherical convolution on its own, we only analyze the convolution for one view for the sake of simplicity. We use $V$ to denote the space of the rays in the same view, where $V \subset \mathbb{S}^2$. For any $x \in V$, we can choose the section map $s_a$ such that $h_{x \to o} = e$, where $o \in \mathbb{S}^2$ that $o$ aligns with the optical axis as shown in the figure 20. Again, we use $s(x)$ to denote $s_a(x)$ for any $x \in \mathbb{S}^2$ in this section.

When $FOV$ is small, for any $x, y \in V$, we can have such approximation: $h_{x \to y} = e$. Then the above gauge equivariant convolution in Sec. D can be approximated as

$$f^{l_{out}}(x) = \int_{y \in \mathcal{N}(x)} \kappa'(s(x)^{-1}y)f^{l_{in}}(y)dy$$

$$\xupdownarrow{t=s(x)^{-1}y} \int_{t \in \mathcal{N}(\eta)} \kappa'(t)f^{l_{in}}(s(x)t)dt,$$

where $\eta = [0,0,1]^T$, the fixed origin in $\mathbb{S}^2$, and $\kappa'(hx) = \rho_{out}(h)\kappa'(x)\rho_{in}^{-1}(h)$ for any $h \in SO(2)$.

Additionally, as illustrated in figure 21, we have a map from $V$ to the projection points on the picture plane represented as $\omega : V \to \mathbb{R}^2$, where $\omega(o)$ is defined as $[0,0]^T$. When $FOV$ is small, we have such approximation that for any $h \in SO(2)$, $t \in \mathcal{N}(\eta)$, and $x \in V$,

$$\omega(s(x)t) \approx \omega(x) + \omega(s(o)t).$$

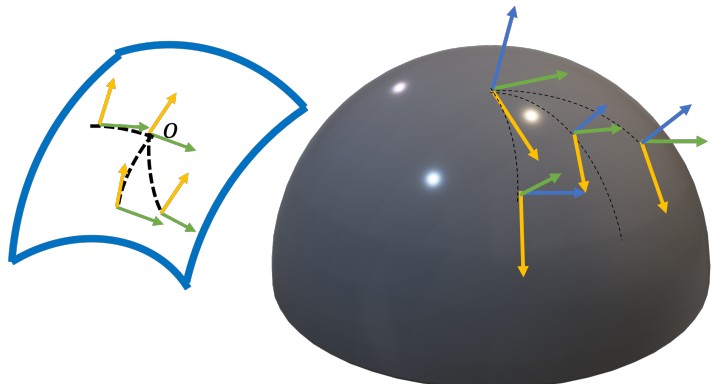

Figure 20: Section choice for every view

It is because

$$\omega(s(x)t) = \omega(x) + \omega(s(o)t)$$
$$+ r(\frac{sin\beta_t}{cos\beta_t} - \frac{sin\beta_t}{cos\beta_x cos(\beta_x + \beta_t)}),$$

and we have

$$lim_{t\to\eta} r(\frac{sin\beta_t}{cos\beta_t} - \frac{sin\beta_t}{cos\beta_x cos(\beta_x + \beta_t)})$$
$$= r(tan\beta_x)^2 \beta_t + o(\beta_t^2),$$

when $\beta_x$ is small (FOV is small), the approximation stands.

Then $f^{l_{out}}(x) = \kappa'(t) f^{l_{in}}(s(x)t)dt$ can be approximately conducted in the image plane:

$$f'^{l_{out}}(\omega(x))$$
$$= \int_{\omega(s(o)t)\in\mathcal{N}([0,0]^T)} \kappa''(\omega(s(o)t)) f'^{l_{in}}(\omega(x) + \omega(s(o)t))$$
$$d(\omega(s(o)t)), \tag{27}$$

where for any $x \in \mathbb{S}^2$, $f'(\omega(x)) = f(x)$, and for any $t \in \mathcal{N}(\eta)$, $\kappa''(\omega(s(o)t)) = \kappa'(t)$.

Since for any $h \in SO(2)$ and any $t \in \mathcal{N}(\eta)$, $\omega(s(o)ht) = h\omega(s(o)t)$, we have for any $h \in SO(2)$ and any $t \in \mathcal{N}(\eta)$,

$$\kappa''(h\omega(s(o)t)) = \kappa''(\omega(s(o)ht)) = \kappa'(ht)$$
$$= \rho_{out}(h)\kappa'(t)\rho_{in}^{-1}(h) = \rho_{out}(h)\kappa''(s(o)t)\rho_{in}^{-1}(h)$$
$$\xrightarrow{p=\omega(s(o)t)\in\mathbb{R}^2} k''(hp)$$
$$= \rho_{out}(h)\kappa''(p)\rho_{in}^{-1}(h).$$

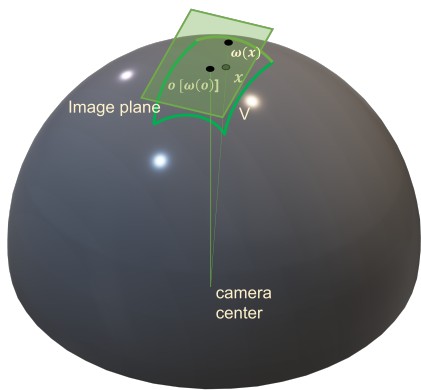

Figure 21: Illustration of projection map $\omega$

Therefore, convolution 27 is exactly $SE(2)$ equivariant convolution and it can be used to approximate the spherical convolution.

In other words, we can intuitively approximate the equivariant convolution over the partial sphere using the $SE(2)$ equivariant network when the distortion of the sphere and the tangent plane of the optical axis is modest.

## F   Construction of Features in Equivariant Light Field Transformer

Noted that $f_2^{out}$, $f_2^{in}$ and $f_1^{in}$ are features that are composed of fields of different types, denoted as $f_2^{out} = \oplus_i f_2^{l_{out_i}}$, $f_2^{in} = \oplus_i f_2^{l_{in_i}}$, and $f_1^{in} = \oplus_i f_1^{l'_{in_i}}$[3]. $f_k$, $f_q$, and $f_v$ are constructed equivariant key features, query features, and value features, respectively, which are composed of fields of different types as well.

We use $f_k = \oplus_i f_k^{l_{k_i}}$, $f_q = \oplus_i f_q^{l_{k_i}}$, and $f_v = \oplus_i f_v^{l_{v_i}}$ to denote $f_k$, $f_q$ and $f_v$, respectively. We construct the features $f_k$, $f_q$ and $f_v$ through the equivariant kernels $\kappa_k = \oplus_{j,i} \kappa_k^{l_{k_j}, l'_{in_i}}$, $\kappa_v = \oplus_{j,i} \kappa_v^{l_{v_j}, l'_{in_i}}$ and equivariant matrix $W_q = \oplus_{j,i} W_q^{l_{k_j}, l_{in_i}}$:

$$
\begin{aligned}
& f_k^{l_{k_j}}(x, y, f_1^{in}) \\
& = \sum_i \kappa_k^{l_{k_j}, l'_{in_i}}(s_2(x)^{-1}y)\rho_1^{l'_{in_i}}(h_1(s_2(x)^{-1}s_1(y)))f_1^{l'_{in_i}}(y);
\end{aligned} \tag{28}
$$

$$
\begin{aligned}
& f_v^{l_{v_j}}(x, y, f_1^{in}) \\
& = \sum_i \kappa_v^{l_{v_j}, l'_{in_i}}(s_2(x)^{-1}y)\rho_1^{l'_{in_i}}(h_1(s_2(x)^{-1}s_1(y)))f_1^{l'_{in_i}}(y);
\end{aligned} \tag{29}
$$

$$
f_q^{l_{k_j}}(x, f_2^{in}) = \sum_i W_q^{l_{k_j}, l_{in_i}} f_2^{l_{in_i}}(x), \tag{30}
$$

where for any $i, j$, any $h_2 \in SO(3)$, and any $x \in \mathcal{R}$ $\kappa_k^{l_{k_j}, l'_{in_i}}$ and $\kappa_v^{l_{v_j}, l'_{in_i}}$ should satisfy that:

$$
\kappa_k^{l_{k_j}, l'_{in_i}}(h_2 x) = \rho_2^{l_{k_j}}(h_2)\kappa_k^{l_{k_j}, l'_{in_i}}(x)\rho_1^{l'_{in_i}}(h_1^{-1}(h_2, x));
$$

$$
\kappa_v^{l_{v_j}, l'_{in_i}}(h_2 x) = \rho_2^{l_{v_j}}(h_2)\kappa_v^{l_{v_j}, l'_{in_i}}(x)\rho_1^{l'_{in_i}}(h_1^{-1}(h_2, x)),
$$

---

[3]Since here the homogeneous spaces of input and output might be different, so as the stabilizer groups, we use $l$ and $l'$ to denote the representations of different stabilizer groups.

where $\mathrm{h}_1(h_2, x) = s_1(h_2 x)^{-1} h_2 s_1(x)$ is the twist function, and for any $i, j$ and any $h_2 \in SO(3)$, $W_q^{l_{k_j}, l_{in_i}}$ satisfies that:

$$\rho_2^{l_{k_j}}(h_2) W_q^{l_{k_j}, l_{in_i}} = W_q^{l_{k_j}, l_{in_i}} \rho_1^{l_{in_i}}(h_2). \tag{31}$$

When the group representation is irreducible representation, due to Schur's Lemma, we have $W_q^{l_{k_j}, l_{in_i}} = cI$ when $l_{k_j} = l_{in_i}$, where $c$ is an arbitrary real number, otherwise $W_q^{l_{k_j}, l_{in_i}} = \mathbf{0}$.

## G  Proof for Equivariance of Light Field Transformer

The equivariant light field transformer defined in the paper reads:

$$
\begin{aligned}
& f_2^{out}(x) \\
&= \sum_{y \in \mathcal{N}(x)} \frac{exp(\langle f_q(x, f_2^{in}), f_k(x, y, f_1^{in}) \rangle)}{\sum_{y \in \mathcal{N}(x)} exp(\langle f_q(x, f_2^{in}) f_k(x, y, f_1^{in}) \rangle} \\
& f_v(x, y, f_1^{in}))
\end{aligned} \tag{32}
$$

is in a general form.

According to [18], one can prove that $f_q$, $f_k$ and $f_v$ are equivariant, that is, for any $g \in SE(3)$, $x \in \mathbb{R}^3$ and $y \in \mathcal{R}$,

$$f_q^{l_{k_j}}(g \cdot x, \mathcal{L}_g^{in}(f_2^{in})) = \rho_2^{l_{k_j}}(\mathrm{h}_2(g^{-1}, g \cdot x)^{-1}) f_q^{l_{k_j}}(x, f_2^{in});$$

$$f_k^{l_{k_j}}(g \cdot x, g \cdot y, \mathcal{L}_g'^{in}(f_1^{in})) = \rho_2^{l_{k_j}}(\mathrm{h}_2(g^{-1}, g \cdot x)^{-1}) f_k^{l_{k_j}}(x, y, f_1^{in});$$

$$f_v^{l_{v_j}}(g \cdot x, g \cdot y, \mathcal{L}_g'^{in}(f_1^{in})) = \rho_2^{l_{v_j}}(\mathrm{h}_2(g^{-1}, g \cdot x)^{-1}) f_v^{l_{v_j}}(x, y, f_1^{in}),$$

where $\mathcal{L}^{in}$ and $\mathcal{L}'^{in}$ are group action of $SE(3)$ on $f_2^{in}$ and $f_1^{in}$, respectively.

The inner product $\langle f_q, f_k \rangle = \sum_i (\overline{f_q^{l_{k_i}}})^T f_k^{l_{k_i}}$ is invariant due to the property of unitary representation, which results in the equivariance of the transformer.

## H  From $SE(3)$ Equivariant Transformer in Ray Space to $SE(3)$ Equivariant Transformer in Euclidean Space

In our implementation for the reconstruction task, the attention model is always only applied over the rays going through the points. We can continue to use the interpretation in the convolution from ray space to $\mathbb{R}^3$ in Ex. 10 that treats any ray $y$ passing through the point $x$ as a point $y'$ such that $y' - x = \mathbf{d}_{s_2(x)^{-1} y}$ as shown in the figure 17.

After we get the initial feature of query points through equivariant convolution from $\mathcal{R}$ to $\mathbb{R}^3$, we update the neighboring ray feature by directly concatenating the query point feature to every ray feature before through a $SO(3)$ equivariant MLP as shown in the figure 22. $SO(3)$ equivariant MLP is composed of an equivariant nonlinear layer and self-interaction layer as in the tensor field networks [55].

Since $y$ becomes point $y'$, and $f_1^{in}$ is the feature over $R^3$ attached to "points" $y'$, it becomes $\oplus_i f_1^{l_{in_i}4}$. Then transformer 32 would be converted to the transformer in [27] over $\mathbb{R}^3$:

---

[4]Since here $f_1^{in}$ is the fields over $\mathbb{R}^3$, we use $l$ instead of $l'$ as the denotation

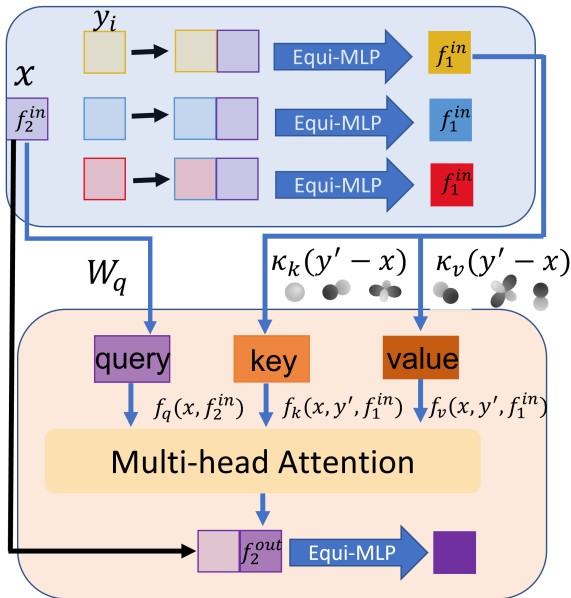

Figure 22: The structure of ray updating and $SE(3)$ transformer. We treat any ray $y$ going through point $x$ as a point $y' \in \mathbb{R}^3$ such that $y' - x = \boldsymbol{d}_{s_2(x)^{-1}}y$. The blue block indicates the ray feature update, and the pink block is the equivariant attention model. For the ray feature updating, the point feature (lavender) is concatenated to every ray feature (light yellow, light blue, and light red) and goes through an equivariant MLP. For the transformer, we get the equivariant query, key, and value feature through the designed linear matrix $W_q$, designed kernels $\kappa_k$ and $\kappa_v$, then apply multi-head attention to obtain the output point feature, which can subsequently be fed into the next ray feature updating and $SE(3)$ transformer block.

$$
\begin{aligned}
&f_2^{out}(x) \\
&= \sum_{y' \in \mathcal{N}(x)} \frac{exp(\langle f_q(x, f_2^{in}), f_k(x, y', f_1^{in}) \rangle)}{\sum_{y' \in \mathcal{N}(x)} exp(\langle f_q(x, f_2^{in}) f_k(x, y', f_1^{in}) \rangle} \\
&\quad f_v(x, y', f_1^{in})),
\end{aligned} \tag{33}
$$

where the subscript denotes the points to which the feature is attached, i.e., $x$ and $y'$.

The features $f_k, f_v$ are constructed by the equivariant kernels $\kappa_k = \oplus_{j,i} \kappa_k^{l_{k_j}, l_{in_i}}$, $\kappa_v = \oplus_{j,i} \kappa_v^{l_{v_j}, l_{in_i}}$:

$$
f_k^{l_{k_j}}(x, y, f_1^{in}) = \sum_i \kappa_k^{l_{k_j}, l_{in_i}}(y' - x) f_1^{l_{in_i}}(y);
$$

$$
f_v^{l_{k_j}}(x, f_2^{in}) = \sum_i \kappa_v^{l_{v_j}, l_{in_i}}(y' - x) f_2^{l_{in_i}}(y),
$$

where for any $i, j$, any $h_2 \in SO(3)$, and any $x \in \mathbb{R}^3$ $\kappa_k^{l_{k_j}, l_{in_i}}$ and $\kappa_v^{l_{v_j}, l_{in_i}}$ should satisfy that:

$$
\kappa_k^{l_{k_j}, l_{in_i}}(h_2 x) = \rho_2^{l_{k_j}}(h_2) \kappa_k^{l_{k_j}, l_{in_i}}(x) \rho_2^{l_{in_i}}(h_2^{-1});
$$

$$
\kappa_v^{l_{v_j}, l_{in_i}}(h_2 x) = \rho_2^{l_{v_j}}(h_2) \kappa_v^{l_{v_j}, l_{in_i}}(x) \rho_1^{l_{in_i}}(h_2^{-1})
$$

as stated in [27].

The feature $f_q$ is constructed in the same way as Equation 30.

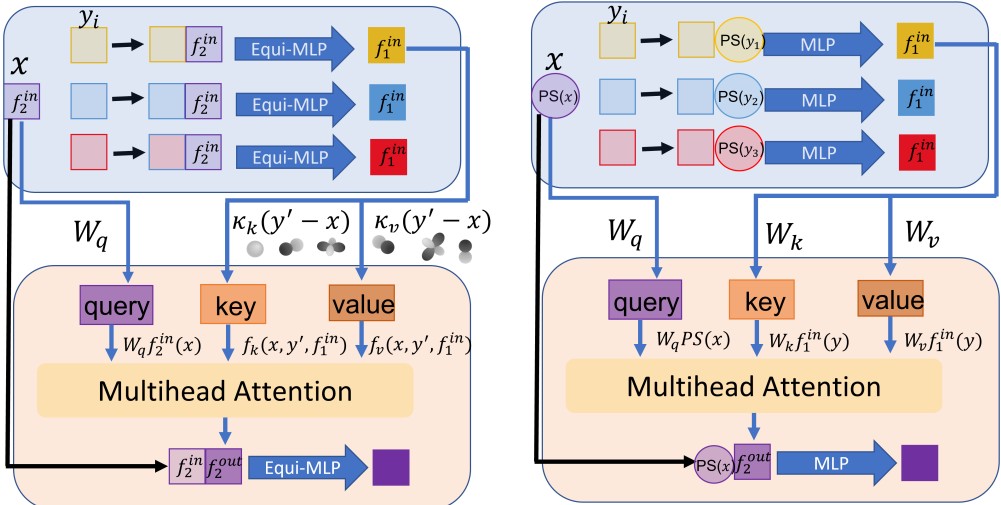

Figure 23: The comparison of the equivariant light field transformer and the conventional transformer. The left is the equivariant light field transformer, and the right is the conventional transformer. In our light field transformer, the position encoding is not directly concatenated to the features because this is not equivariant. We first obtain the equivariant feature attached to the point by equivariant convolution over the rays. We then construct features $f_k$, and $f_v$ with derived designed kernels $\kappa_k$ and $\kappa_v$ to keep them equivariant; we construct $f_q$ by the designed equivariant linear layer $W_q$. Since $f_k$, $f_q$, and $f_v$ are all equivariant, the inner product of $f_k$ and $f_q$ is invariant, which results in invariant attention weight. Therefore, the whole transformer is equivariant. In contrast, the conventional transformer concatenates the ray position encoding with the feature attached to the ray, uses the point position encoding for the query feature for the point, and applies multi-head attention using $f_k$, $f_q$, and $f_v$, which are obtained by the Linear layer. We should note that $W_q$ in the light field transformer is designed to be equivariant, satisfying equation 31, which differs from the conventional linear map $W_q$ in the conventional transformer. For the attention blocks after the first block, the query features of the point in our model and the conventional model are both the output of the last attention block. The difference is that our query feature keeps equivariant while the feature in the conventional transformer is not.

Figure 22 shows the structures of ray feature update and $SE(3)$ equivariant transformer.

In figure 23, we compare the $SE(3)$ equivariant transformer and the conventional transformer to illustrate how the equivariance is guaranteed in the equivariant transformer. In figure 24, we present the types of futures in $SE(3)$ equivariant attention head and conventional attention head, respectively. It indicates that geometric information is aggregated equivariantly in multi-head attention in the equivariant transformer.

# I    Equivariant Neural Rendering

Equivariant rendering relates to equivariant $3D$ reconstruction, where we focus on multiple views instead of the entire light field. The equivariance property is maintained when the ray sampling is invariant up to a coordinate change.

## I.1    Convolution from Rays to Rays

For neural rendering tasks, we query one ray and apply the convolution over the neighboring rays to obtain the feature attached to the target query ray. Similar to the reconstruction, we utilize a kernel with local support. However, there is a distinction in that for neural rendering, the kernel $\kappa$ is constrained to be nonzero only when $d(x, \eta) = 0$, while there are no constraints on $\angle(\boldsymbol{d}_x, [0, 0, 1]^T)$.

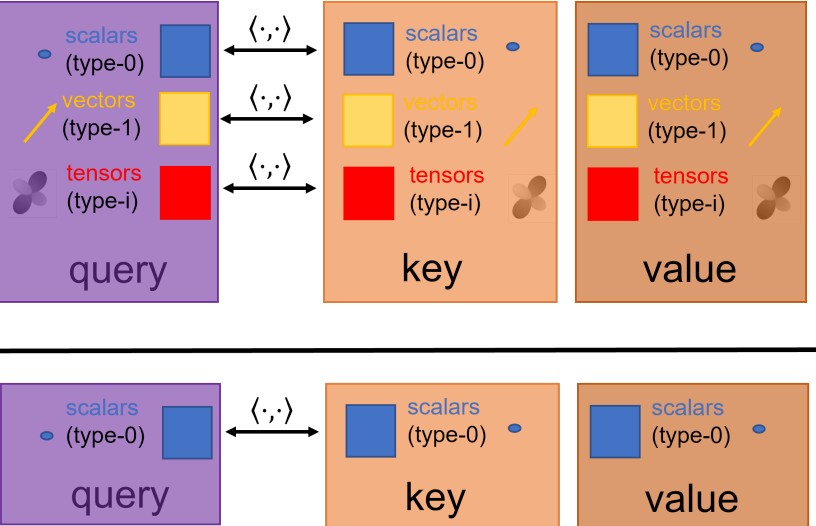

Figure 24: The comparison of multi-head attention modules in the equivariant light field transformer and in the conventional transformer. The figure above is the multi-head attention module in an equivariant light transformer, and the figure below is the conventional transformer. In the light field transformer, the query, key, and value features are composed of different types of features; they can be scalars, vectors, or higher-order tensors. The inner product should apply to the same type of features, and the type of feature determines the way of applying the inner product. In contrast, the feature in a conventional transformer doesn't contain vectors and tensors, and the inner product is conventional.

As a result, the neighboring rays exclusively encompass the rays on the epipolar line for the target ray in each source view, as depicted in Figure 25.

The scalar field over rays serves as the input to the convolution. The output field type corresponds to the regular representation of translation. This is because this field type serves as the input for the cross-attention module later on. If this field type were not utilized, the transformer would reach the entire neighboring set, leading to inferior performance compared to applying the transformer individually for each point and then applying it over the points along the ray. A similar observation is made in [58], which states that the two-stage transformer outperforms the one-stage transformer. Using the field type corresponding to the regular representation of the translation as the input, the transformer from rays to rays is equivalent to performing a transformer for each point, respectively, as explained in the following section.

In Eq. 18, we already provide the solution of the kernel. We give a detailed explanation in this case and show that it is equivalent to performing convolution from rays to rays with output field types corresponding to irreducible representations, followed by applying Inverse Fourier Transform. Given that the input field is a scalar field, we have $\omega_{in}^1 = 0$ and $\omega_{in}^2 = 0$. When considering an output field type of $(\omega_{out}^1, reg)$, where $reg$ represents the regular representation of translation, the convolution can be expressed as follows:

$$(f_{out}^{(\omega_{out}^1, reg)})_t = \int_{y \in \mathcal{N}(x)} \kappa_1(s(x)^{-1}y)(\kappa_2(s(x)^{-1}y))_t f_{in}(y) dy$$

$$= \int_{y \in \mathcal{N}(x)} \kappa_1(s(x)^{-1}y) f(d(\eta, s(x)^{-1}y), \angle([0,0,1]^T, \boldsymbol{d}_{s(x)^{-1}y})) \delta(t - g(s(x)^{-1}y)) f_{in}(y) dy$$

$$= \int_{g(s(x)^{-1}y)=t} \kappa_1(s(x)^{-1}y) f(d(\eta, s(x)^{-1}y), \angle([0,0,1]^T, \boldsymbol{d}_{s(x)^{-1}y})) f_{in}(y) dy.$$

From the above equation, we can intuitively find that when the output field corresponds to the regular representation of the translation, the convolution happens at every point along the ray, respectively. We can treat $f_{out}^{(\omega_{out}^1, reg)}$ as a function over $\mathbb{R}$, and for any $\omega \in \mathbb{R}$ we apply the Fourier Transform to $f_{out}^{(\omega_{out}^1, reg)}$:

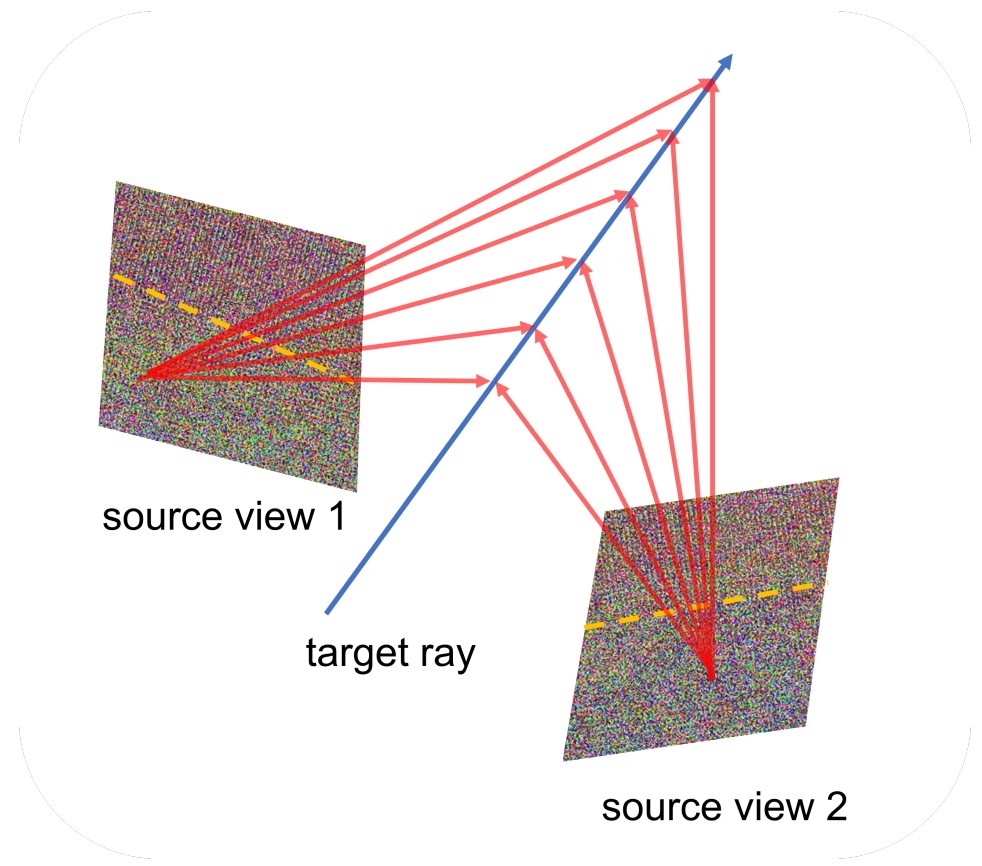

Figure 25: For simplification, we show two source views. For a target query ray $x$, the neighboring rays (denoted by red rays) are on the epipolar lines (denoted as yellow dotted dashes) for the target ray in each source view. For any ray $y \in \mathcal{N}(x)$, $d(x, y) = 0$.

$$
\begin{aligned}
\mathcal{F}(\omega) &= \int_t f_{out}^{(\omega_{out}^1, reg)}(t) e^{-i\omega t} dt \\
&= \int_t \int_{g(s(x)^{-1}y)=t} \kappa_1(s(x)^{-1}y) f(d(\eta, s(x)^{-1}y), \angle([0,0,1]^T, \boldsymbol{d}_{s(x)^{-1}y})) f_{in}(y) dy e^{-i\omega t} dt \\
&= \int_y \kappa_1(s(x)^{-1}y) f(d(\eta, s(x)^{-1}y), \angle([0,0,1]^T, \boldsymbol{d}_{s(x)^{-1}y})) e^{-i\omega g(s(x)^{-1}y)} f_{in}(y) dy \\
&= \int_y \kappa_1(s(x)^{-1}y) \kappa_2'(s(x)^{-1}y) f_{in}(y) dy,
\end{aligned}
$$

where $\kappa_2' = f(d(\eta, s(x)^{-1}y), \angle([0,0,1]^T, \boldsymbol{d}_{s(x)^{-1}y})) e^{-i\omega g(s(x)^{-1}y)}$, which is exactly the kernel corresponding to $\omega_{out}^2 = \omega$ and $\omega_{in}^2 = 0$ as stated in Eq. 14. Therefore, we know that the field corresponding to the irreducible representation of the translation can be treated as the Fourier coefficients of the field corresponding to the regular representation. We can first obtain the features of different irreducible representations attached to the ray and subsequently apply the Inverse Fourier Transform to get the features for points along the ray, as shown in figure 26.

## I.2    Cross-attention over Rays

The feature that generates the query in the transformer is the feature attached to the target ray, whose feature type corresponds to the regular representation of the translation. The feature that generates the key and value in the transformer is attached to the neighboring rays in the source view, whose

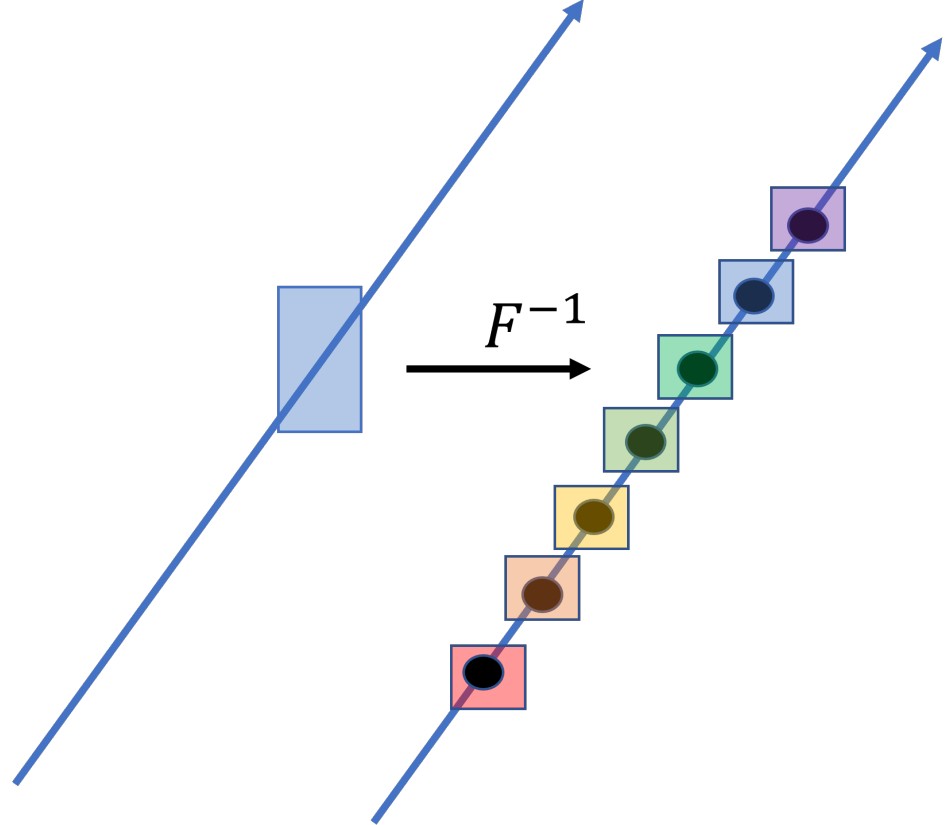

Figure 26: The features for points along the ray (the field type corresponds to the regular representation) can be obtained by the Inverse Fourier Transform of features attached to the ray, where the types of feature fields correspond to the irreducible representation of the translation.

feature type corresponds to the scalar field. The output is the feature attached to the target ray, whose feature type corresponds to the regular representation. Therefore, the transformer becomes:

$$(f_2^{out}(x))_t = \sum_{y \in \mathcal{N}(x)} \frac{exp(\langle (f_q(x, f_2^{in}))_t, (f_k(x, y, f_1^{in}))_t \rangle)}{\sum_{y \in \mathcal{N}(x)} exp(\langle (f_q(x, f_2^{in}))_t (f_k(x, y, f_1^{in}))_t \rangle} )(f_v(x, y, f_1^{in}))_t, \quad (34)$$

where

$$(f_k(x, y, f_1^{in}))_t = (\kappa_k(s_2(x)^{-1}y))_t f_1^{in}(y)$$
$$(f_v(x, y, f_1^{in}))_t = (\kappa_v(s_2(x)^{-1}y))_t f_1^{in}(y)$$
$$(f_q(x, f_2^{in}))_t = C(f_2^{in}(x))_t.$$

In the equations above, $\kappa_k$ and $\kappa_v$ are the kernels derived in Ex. 9 Eq. 16, $C$ is the equivariant weight matrix satisfying Eq. 31.

The expression above indicates that the feature types of both key and value correspond to the regular representation of translation, as well as the feature type of the query. Moreover, the transformer operates on each point along the ray independently. It should be noted that the features $(f_k)_t$, $(f_q)_t$, $(f_v)_t$ and $(f_2^{in})_t$ may have multiple channels and may consist of different types of features corresponding to various representations of $SO(2)$. The inner product $\langle \cdot, \cdot \rangle$ can only happen in the field type of the same representation of $SO(2)$. This allows for the implementation of a multi-head attention module, where each head can attend to a specific type of feature and multiple channels.

### I.3 Self-attention over Points Along the Ray

After the cross-attention over rays, we get the features of the points along the ray, i.e., the feature attached to the ray corresponding to the regular representation of translation. $SE(3)$ acts on the feature $f'$ attached to the point along the ray as mentioned in Eq.15 :

$$(\mathcal{L}_g f')(\boldsymbol{x}, \boldsymbol{d}) = \rho_1(R_Z(R_{g^{-1}}, \boldsymbol{d}))^{-1} f'(g^{-1}\boldsymbol{x}, R_{g^{-1}}\boldsymbol{d}),$$

where $\rho_1$ is the group representation of $SO(2)$.

We will apply the self-attention model to these points along the same ray. For two points $\boldsymbol{x}_1$ and $\boldsymbol{x}_2$ on the same ray $(\boldsymbol{d}, \boldsymbol{x}_1 \times \boldsymbol{d})$, one can observe that for the same type of feature, $\langle(\mathcal{L}_g f')(\boldsymbol{x_1}, \boldsymbol{d}), (\mathcal{L}_g f')(\boldsymbol{x_1}, \boldsymbol{d})\rangle = \langle f'(g^{-1}\boldsymbol{x_1}, R_{g^{-1}}\boldsymbol{d}), f'(g^{-1}\boldsymbol{x_1}, R_{g^{-1}}\boldsymbol{d})\rangle$, which makes attention weight invariant, the transformer could be formulated as:

$$f^{out}(x) = \sum_{\text{y on the same ray as x}} \frac{exp(\langle f_q(f^{in}, x), f_k(f^{in}, x, y)\rangle)}{\sum_{\text{y on the same ray as x}} exp(\langle f_q(f^{in}, x), f_k(f^{in}, x, y)\rangle)} f_v(x, y, f^{in}), \quad (35)$$

where

$$f_k^l(x, y, f^{in}) = c_k(d(x,y))I(f^{in})^l(y)$$
$$f_v^l(x, y, f^{in}) = c_v(d(x,y))I(f^{in})^l(y)$$
$$f_q^l(x, f^{in}) = c_q I(f^{in})^l(x),$$

and $x$ and $y$ are the points along the same ray with direction $\boldsymbol{d}$, we can denote $x$ as $(\boldsymbol{x}, \boldsymbol{d})$ and $y$ as $(\boldsymbol{y}, \boldsymbol{d})$, $d(x,y)$ is the signed distance $\langle \boldsymbol{d}, \boldsymbol{y} - \boldsymbol{x}\rangle$, $c_k, c_v$ are arbitrary functions that take signed distance as the input and output complex values and $c_q$ is an arbitrary constant complex. It should be noted that the features $f_k$, $f_q$, $f_v$, and $f^{in}$ may have multiple channels and consist of different types of features corresponding to various representations of $SO(2)$, the inner product $\langle\cdot, \cdot\rangle$ can only happen in the same type of field. This allows for implementing a multi-head attention module, where each head can attend to a specific feature type and multiple channels. Here, $f_k^l$ denotes the type$-l$ feature in feature $f_k$, $f_v^l$ represents the type$-l$ feature in feature $f_v$, $f_q^l$ denotes the type$-l$ feature in feature $f_l$, and $(f^{in})^l$ represents the type$-l$ feature in feature $f^{in}$.

Note that this transformer architecture also follows the general format of the transformer in Eq. 4. We only simplify the kernel $\kappa_k$, $\kappa_v$ to be trivial equivariant kernels.

To obtain a scalar feature density for each point, the feature output of each point can be fed through an equivariant MLP, which includes equivariant linear layers and gated/norm nonlinear layers. These layers are similar to the ones used in [62] and [63].

## J   $3D$ Reconstruction Experiment

### J.1   Generation of the Dataset

The I dataset is obtained by fixing the orientation of the object as well as the eight camera orientations. With the object orientation fixed, we can independently rotate each camera around its optical axis by a random angle in a uniform distribution of $(-\pi, \pi]$ to obtain the Z dataset. For the R dataset, we rotate every camera randomly by any rotation in $SO(3)$ while fixing the object. The equivariance stands with the content unchanged. Therefore, in practice, we require that the object projection after the rotation does not have new parts of the object. We satisfy this assumption by forcing the camera to fixate on a new random point inside a small neighborhood and subsequently rotate each camera around its optical axis with the uniformly random angle in $(-\pi, \pi]$. We generate the $Y$ dataset by rotating the object only with azimuthal rotations while keeping the camera orientations the same. The $SO(3)$ dataset is generated by rotating the object with random rotation in $SO(3)$ with the orientations of cameras unchanged, which will potentially result in new image content. Equivariance is not theoretically guaranteed in this setup, but we still want to test the performance of our method.

```
------------------------- DeepSpeed Flops Profiler -------------------------
Profile Summary at step 10:
Notations:
data parallel size (dp_size), model parallel size(mp_size),
number of parameters (params), number of multiply-accumulate operations(MACs),
number of floating-point operations (flops), floating-point operations per second (FLOPS),
fwd latency (forward propagation latency), bwd latency (backward propagation latency),
step (weights update latency), iter latency (sum of fwd, bwd and step latency)

params per gpu:                                            10.02 M
params of model = params per GPU * mp_size:                10.02 M
fwd MACs per GPU:                                          0 MACs
fwd flops per GPU:                                         105.09 M
fwd flops of model = fwd flops per GPU * mp_size:          105.09 M
fwd latency:                                               42.56 ms
fwd FLOPS per GPU = fwd flops per GPU / fwd latency:       2.47 GFLOPS

--------------------------- Aggregated Profile per GPU ---------------------------
Top 1 modules in terms of params, MACs or fwd latency at different model depths:
depth 0:
    params      - {'ReNet_inpaint_Nobn': '10.02 M'}
    MACs        - {'ReNet_inpaint_Nobn': '0 MACs'}
    fwd latency - {'ReNet_inpaint_Nobn': '42.56 ms'}
depth 1:
    params      - {'ModuleList': '10.02 M'}
    MACs        - {'ModuleList': '0 MACs'}
    fwd latency - {'ModuleList': '42.35 ms'}
depth 2:
    params      - {'ResLayer': '9.89 M'}
    MACs        - {'Sequential': '0 MACs'}
    fwd latency - {'ResLayer': '39.21 ms'}
depth 3:
    params      - {'BasicBlock': '9.89 M'}
    MACs        - {'R2Conv': '0 MACs'}
    fwd latency - {'BasicBlock': '39.07 ms'}
depth 4:
    params      - {'R2Conv': '9.72 M'}
    MACs        - {'BlocksBasisExpansion': '0 MACs'}
    fwd latency - {'R2Conv': '34.7 ms'}
depth 5:
    params      - {'R2Conv': '170.64 k'}
    MACs        - {'SingleBlockBasisExpansion': '0 MACs'}
    fwd latency - {'R2Conv': '1.06 ms'}
depth 6:
    params      - {'BatchNorm3d': '632'}
    MACs        - {'SingleBlockBasisExpansion': '0 MACs'}
    fwd latency - {'BatchNorm3d': '272.51 us'}
```

Figure 27: The number of parameters and FLOPs of $SE(2)$ equivariant CNNs. We set batch size as one to calculate number of FLOPs.

## J.2 Implementation Details

We use $SE(2)$ equivariant CNNs to approximate the equivariant convolution over the rays. We use the same ResNet backbone as implemented in [30] that is equivariant to the finite group $C_8$, which we find achieves the best result compared with other $SE(2)$ equivariant CNNs. We use a similar pyramid structure as [69] that concatenates the output feature of every block. Since every hidden feature is the regular representation, in the final layer we use $1 \times 1$ $SE(2)$-equivariant convolutional layers to transfer the hidden representation to scalar type.

For the fusion from the ray space to the point space model, we use one layer of convolution and three combined blocks of updating ray features and $SE(3)$ transformers. For the equivariant $SE(3)$ multi-head-attention, we only use the scalar feature and the vector (type-1) feature in the hidden layer. The kernel matrix includes the spherical harmonics of degrees 0 and 1. We also concatenate every output point feature of every block as in the $2D$ backbone. Since the output feature of every block includes the vector feature, we transfer it to the scalar feature through one vector neuron layer and the inner vector product. We use the same weighted SDF loss as in [69] during training, which applies both uniform and near-surface sampling. We report the number of parameters and floating-point operations (FLOPs) of our $2D$ backbone and light fusion networks in Fig. 27 and Fig. 28 respectively.

```
----------------------- DeepSpeed Flops Profiler -----------------------
Profile Summary at step 10:
Notations:
data parallel size (dp_size), model parallel size(mp_size),
number of parameters (params), number of multiply-accumulate operations(MACs),
number of floating-point operations (flops), floating-point operations per second (FLOPS),
fwd latency (forward propagation latency), bwd latency (backward propagation latency),
step (weights update latency), iter latency (sum of fwd, bwd and step latency)

params per gpu:                                      1.53 M
params of model = params per GPU * mp_size:          1.53 M
fwd MACs per GPU:                                    19.57 GMACs
fwd flops per GPU:                                   39.15 G
fwd flops of model = fwd flops per GPU * mp_size:    39.15 G
fwd latency:                                         39.67 ms
fwd FLOPS per GPU = fwd flops per GPU / fwd latency:  986.86 GFLOPS

-------------------------- Aggregated Profile per GPU --------------------------
Top 1 modules in terms of params, MACs or fwd latency at different model depths:
depth 0:
    params      - {'Fusion_0d': '1.53 M'}
    MACs        - {'Fusion_0d': '19.57 GMACs'}
    fwd latency - {'Fusion_0d': '39.67 ms'}
depth 1:
    params      - {'Sequential': '895.9 k'}
    MACs        - {'ModuleList': '15.54 GMACs'}
    fwd latency - {'ModuleList': '32.23 ms'}
depth 2:
    params      - {'Sequential': '629.18 k'}
    MACs        - {'Sequential': '15.54 GMACs'}
    fwd latency - {'Sequential': '32.23 ms'}
depth 3:
    params      - {'Linear': '752.06 k'}
    MACs        - {'VNLinearLeakyReLU': '8.76 GMACs'}
    fwd latency - {'VNLinearLeakyReLU': '23.61 ms'}
```

Figure 28: The number of parameters and FLOPs of the ray fusion model, which is composed of convolution from rays to points and transformer from rays to points. We set batch size as one to calculate the number of FLOPs.

## J.3   Discussion of Results

There is still a performance gap between $I/I$ and $I/Z$. Although $SE(2)$ equivariant networks are theoretically strictly equivariant, the error in practice is introduced by the finite sampling of the image and the pooling layers. Additionally, we use the ResNet that is equivariant to $C_8$ approximation of $SO(2)$, which causes this gap but increases the whole pipeline performance in the other tasks. There is no significant difference between $I/Z$ and $I/R$, which shows that approximating the spherical field convolution by $SE(2)$ equivariant convolution is reasonable in practice.

## J.4   Qualitative Results

Figure 29 shows a qualitative result for the chair category. There are more qualitative results shown in Fig. 37, Fig. 38, and Fig. 39.

## J.5   Ablation Study

First, we replace the $SE(2)$ CNNs backbone with the conventional CNNs to test the effectiveness of $SE(2)$ CNNs. Secondly, we remove the equivariant convolution/transformer part and use trivial aggregation (max-pooling) combined with MLP. Finally, we run an equivariant convolution and transformer without using the type-1 (vector) feature while keeping the number of parameters similar to our model.

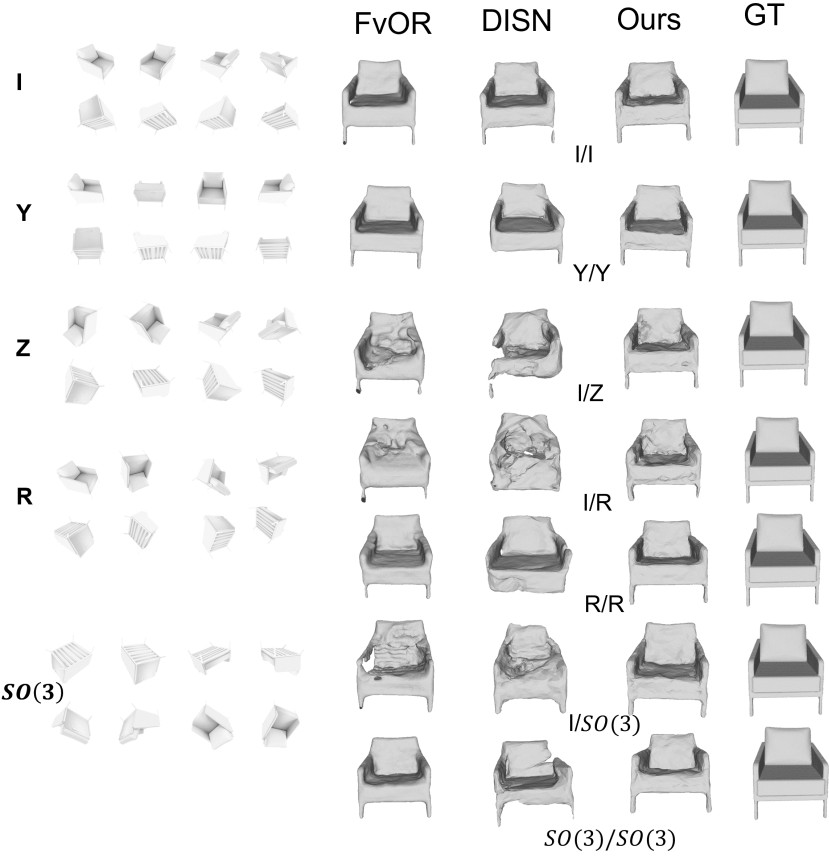

Figure 29: Qualitative results for equivariant reconstruction. Left: input views; Right: reconstruction meshes of different models and ground truth meshes. The captions below the meshes show how the model is trained and tested, explained in the text.

Table 3 summarizes the result on the chair category, which illustrates that in the $I/I$ and $Y/Y$ trials, $SE(2)$ CNN is less expressive than traditional CNN, but it contributes to the equivariance of our model looking at the results of $I/Z$, $I/R$, and $Y/SO(3)$. Equivariant ray convolution and transformer improve both the reconstruction performance and the equivariance outcome. We also compare the ray convolution and transformer with the models operating only on scalar features without vector features, and again, we see a drop in performance in every setting, proving the value of taking ray directions into account.

We also compare to a baseline where the ray difference information is encoded in the feature explicitly. Most models that encode ray directions aim at rendering, like IBRnet. Here, we modified IBRnet (Fig.2 of IBRnet paper) to query 3D points only for their SDF value instead of querying all densities along the ray that would be necessary for rendering. We replaced the ray direction differences with the ray directions themselves because we use a query point and not a query ray. We report in table 4 IoU result for Y/Y and Y/SO(3) (where Y is augmentation only along the vertical axis) for two models – IBRNet with conventional CNNs as 2D backbone and IBRNet with SE(2)-equivariant CNNs as 2D backbone. For the $SO(3)$ setting, we rotate the whole 8 cameras with the same rotation, which is equivalent to rotating the object with the inverse rotation, and we use the object canonical frame to encode the ray information.

The baseline is not equivariant: It explicitly uses the ray directions as inputs to MLPs. Ray directions or their differences change when the coordinate system is transformed, breaking, thus, equivariance. Table 4 demonstrates that our model is more resilient to object rotations. We can enhance equivariance

| Method | w/o SE(2) | w/o conv& trans | w/o type-1 | Full model |
|---|---|---|---|---|
| I/I | **0.767/0.079** | 0.695/0.105 | 0.722/0.093 | 0.731/0.090 |
| I/Z | 0.430/0.234 | 0.533/0.175 | 0.553/0.158 | **0.631/0.130** |
| I/R | 0.417/0.249 | 0.442/0.241 | 0.466/0.203 | **0.592/0.137** |
| R/R | 0.672/0.112 | 0.658/0.122 | 0.682/0.109 | **0.689/0.105** |
| Y/Y | **0.731/0.090** | 0.644/0.124 | 0.677/0.111 | 0.698/0.102 |
| Y/SO(3) | 0.467/0.0.217 | 0.534/0.170 | 0.569/0.163 | **0.589/0.142** |
| SO(3)/SO(3) | 0.655/0.120 | 0.616/0.142 | 0.636/0.130 | **0.674/0.113** |

Table 3: Ablation: w/o $SE(2)$ means replacing $SE(2)$ equivariant network with conventional; w/o ray conv& trans denotes the model where we replace the light field convolution and the light field equivariant transformer with max-pooling; w/o type-1 means using only scalar features in convolution and transformers.

| Method | Y/Y | Y/SO(3) | SO(3)/SO(3) |
|---|---|---|---|
| IBRNet [61] w/o SE(2) | 0.689 | 0.432 | 0.611 |
| IBRNet [61] w/SE(2) | 0.652 | 0.501 | 0.619 |
| Ours | **0.698** | **0.598** | **0.674** |

Table 4: Comparison of our model and a baseline which encodes the ray information explicitly. IBRNet w/o SE(2) is the modified IBRNet with conventional CNN backbone, IBRNet w/SE(2) is the model where we replace the conventional CNN backbone with the SE(2) equivariant CNN.

by using SE(2) equivariant modeling, and our model outperforms the baseline in the Y/Y setting. We believe that the transformer in our model is responsible for the performance improvement.

# K   Neural Rendering Experiment

## K.1   Experiment Settings Discussion

Two experiment settings illustrate our model's equivariance: $I/I$ and $I/SO(3)$. $I/I$ is the canonical setting, where we train and test the model in the same canonical frame defined in the dataset. $I/SO(3)$ is that we test the model trained in the canonical frame under arbitrary rotated coordinate frames, which means that all the camera poses in one scene are transformed by the same rotation without changing their relative camera poses and relative poses between the camera and the scene, which doesn't change the content of the multiple views. The reason we don't apply translation to the cameras is that there exists a depth range for points sampling in the model and the comparing baseline [61], which effectively mitigates the impact of translation.

We should note that the $SO(3)$ setting in this experiment setting differs from $R$ and $SO(3)$ settings in reconstruction. $R$ changes the relative pose of the cameras, and each image is transformed due to the rotation of each camera without altering the content, i.e., the sampling of the light field is nearly unchanged. The $R$ setting aims to demonstrate that replacing the conventional method with ray-based convolution can get rid of the canonical frame for each view.

$SO(3)$ in reconstruction is to rotate the object pose randomly without changing the pose of the camera, which is equivalent to transforming the cameras by the inverse rotation but fixing the object, resulting in changes in the relative poses between the camera and the object, the content of the image and, therefore, the sampling of the light field. This setting shows that even for non-theoretically equivariant cases, our model in reconstruction still demonstrates robustness.

In the rendering experiment using the $SO(3)$ setting, each image itself is not transformed, unlike the $R$ setting in the reconstruction. The content of the images remains unchanged, including the light field sampling, unlike the $SO(3)$ setting in the reconstruction. Since each image is not transformed, even if the conventional $2D$ convolution is applied to the image, the scalar feature attached to the ray is not altered, and the light feature field sampling remains the same up to the transform of the coordinate frame. This setting was used to demonstrate that our model is $SE(3)$-equivariant when the input is the scalar light feature field.

```
------------------------ DeepSpeed Flops Profiler ------------------------
Profile Summary at step 10:
Notations:
data parallel size (dp_size), model parallel size(mp_size),
number of parameters (params), number of multiply-accumulate operations(MACs),
number of floating-point operations (flops), floating-point operations per second (FLOPS),
fwd latency (forward propagation latency), bwd latency (backward propagation latency),
step (weights update latency), iter latency (sum of fwd, bwd and step latency)

params per gpu:                                             21.96 k
params of model = params per GPU * mp_size:                 21.96 k
fwd MACs per GPU:                                           9.1 MMACs
fwd flops per GPU:                                          20.29 M
fwd flops of model = fwd flops per GPU * mp_size:           20.29 M
fwd latency:                                                8.81 ms
fwd FLOPS per GPU = fwd flops per GPU / fwd latency:        2.3 GFLOPS

------------------------- Aggregated Profile per GPU ----------------------------
Top 1 modules in terms of params, MACs or fwd latency at different model depths:
depth 0:
    params      - {'IBRNet_lf': '21.96 k'}
    MACs        - {'IBRNet_lf': '9.1 MMACs'}
    fwd latency - {'IBRNet_lf': '8.81 ms'}
depth 1:
    params      - {'Sequential': '13.75 k'}
    MACs        - {'Sequential': '8.47 MMACs'}
    fwd latency - {'Sequential': '2.84 ms'}
depth 2:
    params      - {'Linear': '14.33 k'}
    MACs        - {'Linear': '8.51 MMACs'}
    fwd latency - {'ELU': '2.56 ms'}
```

Figure 30: The number of parameters and FLOPs of the model, which takes the scalar feature attached to rays as input and predicts the color and density for points along the target ray. The calculation of FLOPs is performed for single-pixel rendering with 10 source views.

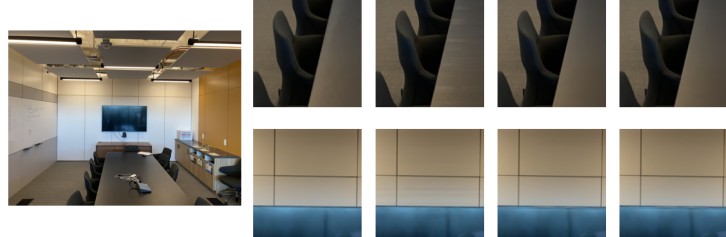

Figure 31: In terms of qualitative results for rendering, we compare the performance of IBRNet and our model in both the given canonical frame (denoted as "IBRNet(I)" and "Ours(I)" respectively) and a rotated frame (denoted as "IBRNet(SO(3))" and "Ours(SO(3))" respectively). Our model performs comparably to IBRNet in the canonical setting. However, IBRNet experiences a performance drop in the rotated frame, while our model remains robust to the rotation.

### K.2  Implementation Details

As described in the paper, we use a similar architecture as [61], where we replace the aggregation of view features by equivariant convolution and equivariant transformer over rays. In equivariant convolution, the input is scalar feature field over rays, which means that $\omega_{in}^1 = 0$ and $\omega_{in}^2 = 0$; for the output field, we use regular representation of translation as described in Sec. 3.3 , and we use $\omega_{out}^1 = 0, 2^1, \cdots, 2^7$ for group representation of $SO(2)$, each field type has 4 channels. In equivariant transformer over rays, we update the key and value before going to the attention module in the experiment; the specific operation is that we concatenate key $f_k$ and query $f_q$, we concatenate

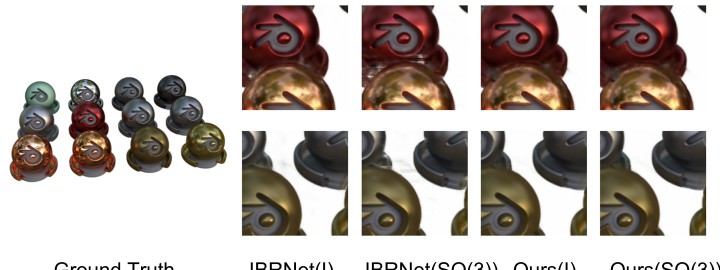

Ground Truth          IBRNet(I)    IBRNet(SO(3))  Ours(I)    Ours(SO(3))

Figure 32: In terms of qualitative results for rendering, we compare the performance of IBRNet and our model in both the given canonical frame (denoted as "IBRNet(I)" and "Ours(I)" respectively) and a rotated frame (denoted as "IBRNet(SO(3))" and "Ours(SO(3))" respectively). Our model performs comparably to IBRNet in the canonical setting. However, IBRNet experiences a performance drop in the rotated frame, while our model remains robust to the rotation.

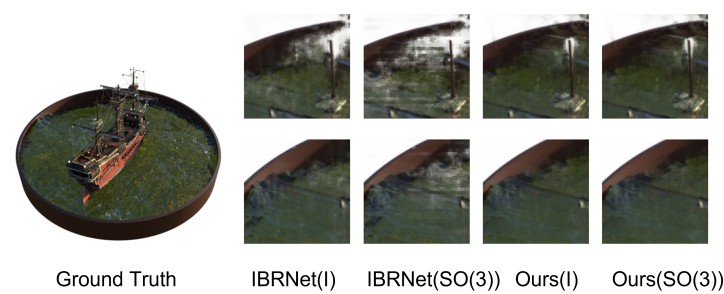

Ground Truth          IBRNet(I)    IBRNet(SO(3))  Ours(I)    Ours(SO(3))

Figure 33: In terms of qualitative results for rendering, we compare the performance of IBRNet and our model in both the given canonical frame (denoted as "IBRNet(I)" and "Ours(I)" respectively) and a rotated frame (denoted as "IBRNet(SO(3))" and "Ours(SO(3))" respectively). Our model performs comparably to IBRNet in the canonical setting. However, IBRNet experiences a performance drop in the rotated frame, while our model remains robust to the rotation.

$f_v$ and query $f_q$, and then we feed the concatenated key and value into two equivariant MLPs (equivariant linear layers and gated/norm nonlinear layers, similar to the ones used in [62]) to get the newly updated key and updated value, which will be fed into attention module. In line with [61], our approach does not involve generating features for the color of every point. In our implementation, we directly multiply the attention weights obtained from the softmax operator in the transformer with the corresponding colors in each view to perform color regression.

We replace the ray transformer with the equivariant transformer over the points along the ray; the input features comprise the feature types corresponding to the group representations $\omega_{in} = 0, 2^1, \cdots, 2^7$ for $SO(2)$. Each feature type has $4$ channels; the output comprises the same feature type, and each type has $2$ channels. We will first convert the feature into a scalar feature by an equivariant MLP (equivariant linear layers and gated/norm nonlinear layers, similar to the ones used in [62].) and then feed it into a conventional MLP to get the density. We report in Fig. 30 the number of parameters and floating-point operations (FLOPs) of the model composed of the convolution and transformers.

### K.3   Qualitative Results

Fig. 32, Fig. 33, Fig. 34, Fig. 31, Fig. 35 and Fig. 36 show the qualitative results on Real-Forward-Facing [41] and Realistic Synthetic $360°$ [49] data. Our model performs comparably to IBRNet in the canonical setting. However, IBRNet experiences a performance drop in the rotated frame, while our model remains robust to the rotation.

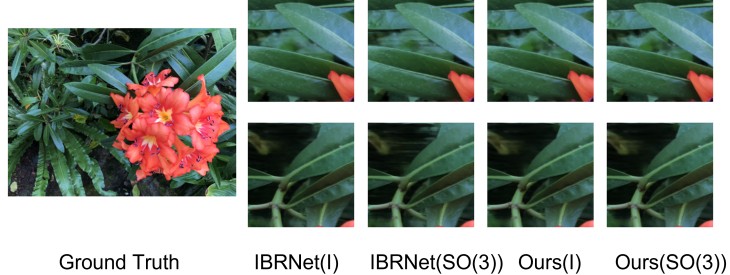

Ground Truth      IBRNet(I)    IBRNet(SO(3))  Ours(I)    Ours(SO(3))

Figure 34: In terms of qualitative results for rendering, we compare the performance of IBRNet and our model in both the given canonical frame (denoted as "IBRNet(I)" and "Ours(I)" respectively) and a rotated frame (denoted as "IBRNet(SO(3))" and "Ours(SO(3))" respectively). Our model performs comparably to IBRNet in the canonical setting. However, IBRNet experiences a performance drop in the rotated frame, while our model remains robust to the rotation.

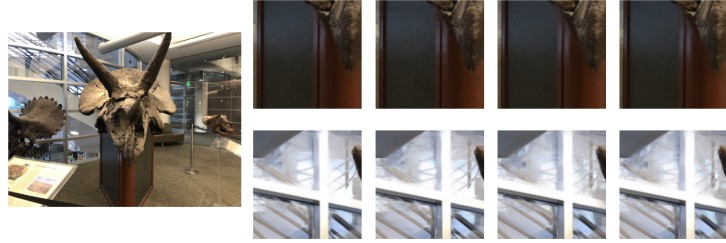

Figure 35: In terms of qualitative results for rendering, we compare the performance of IBRNet and our model in both the given canonical frame (denoted as "IBRNet(I)" and "Ours(I)" respectively) and a rotated frame (denoted as "IBRNet(SO(3))" and "Ours(SO(3))" respectively). Our model performs comparably to IBRNet in the canonical setting. However, IBRNet experiences a performance drop in the rotated frame, while our model remains robust to the rotation.

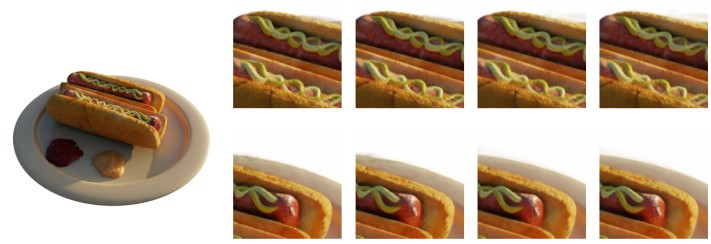

Figure 36: In terms of qualitative results for rendering, we compare the performance of IBRNet and our model in both the given canonical frame (denoted as "IBRNet(I)" and "Ours(I)" respectively) and a rotated frame (denoted as "IBRNet(SO(3))" and "Ours(SO(3))" respectively). Our model performs comparably to IBRNet in the canonical setting. However, IBRNet experiences a performance drop in the rotated frame, while our model remains robust to the rotation.

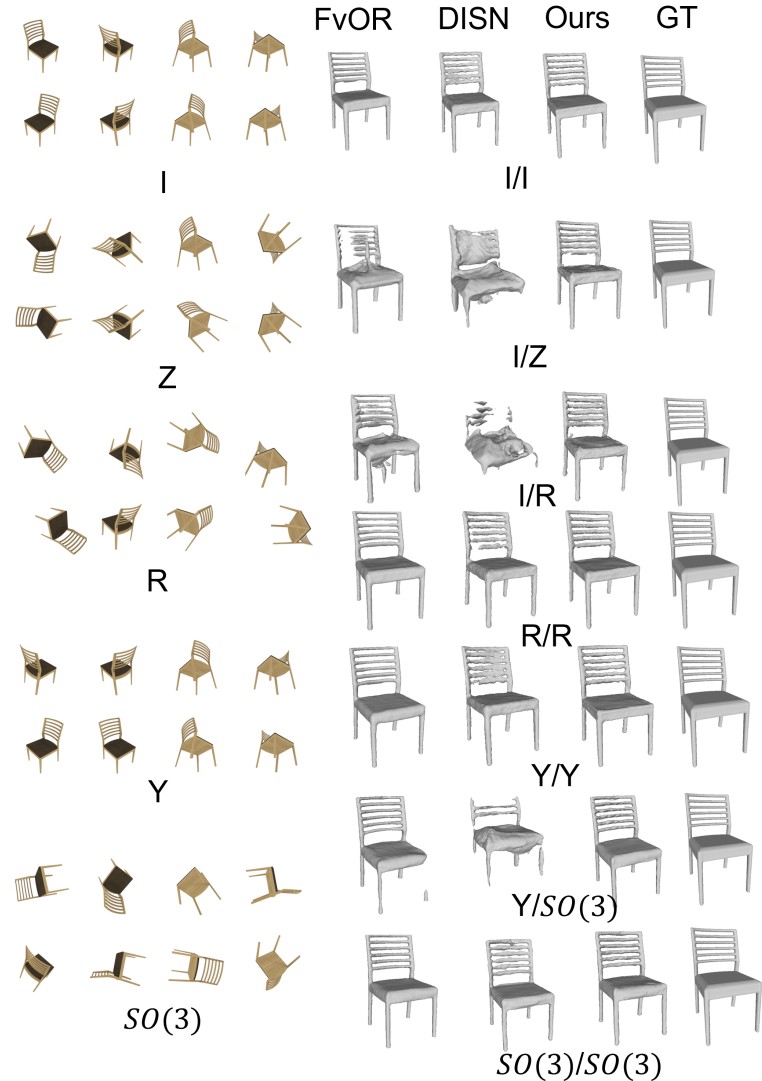

Figure 37: Qualitative Result for the chair. Left: input views; Right: reconstruction meshes of different models. The captions below the meshes show how the model is trained and tested.

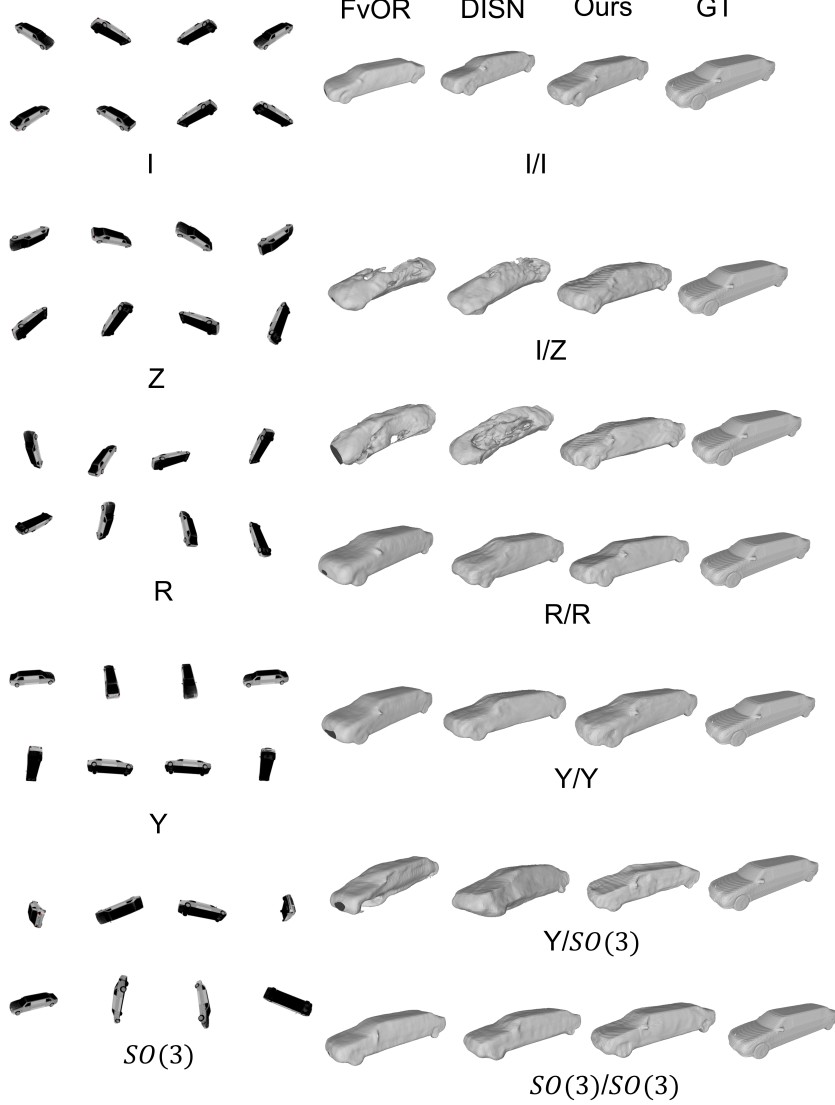

Figure 38: Qualitative Result for the car. Left: input views; Right: reconstruction meshes of different models. The captions below the meshes show how the model is trained and tested.

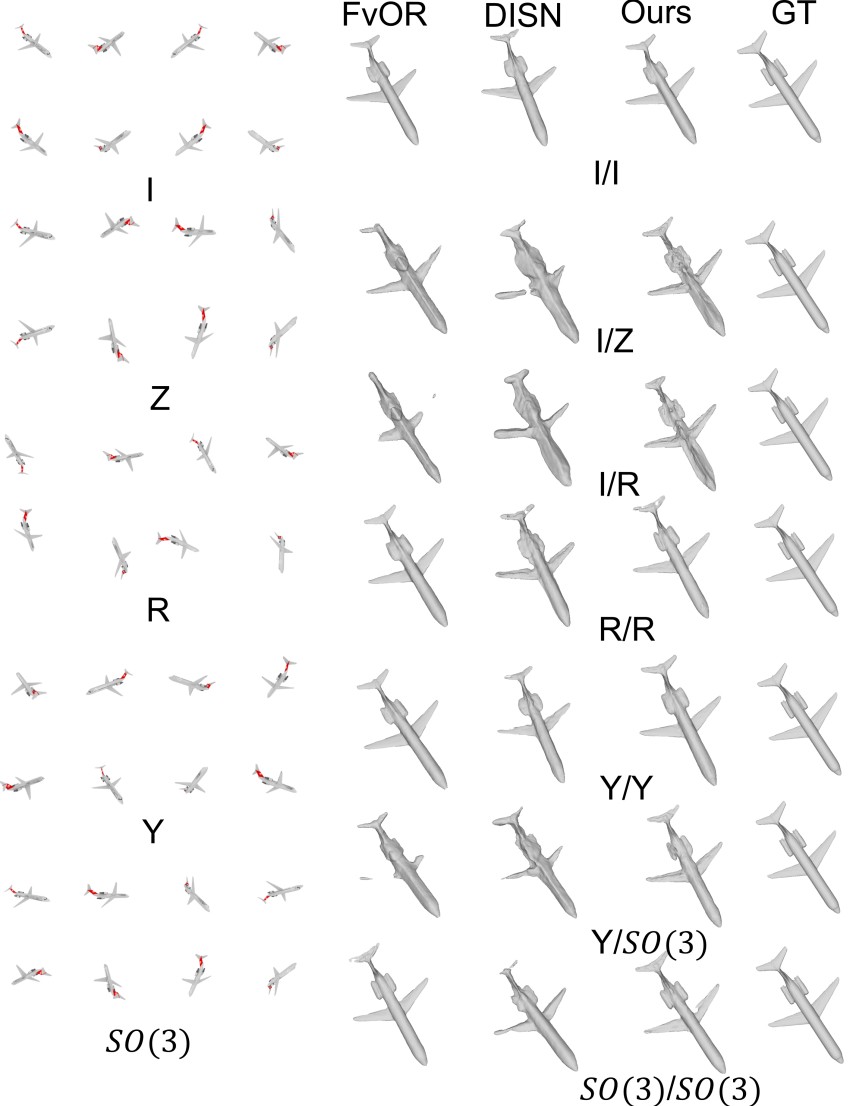

Figure 39: Qualitative Result for the car. Left: input views; Right: reconstruction meshes of different models. The captions below the meshes show how the model is trained and tested.

