# OpenReview forum: "$SE(3)$  Equivariant Convolution and Transformer in Ray Space"
_NeurIPS.cc/2023/Conference — NeurIPS 2023 spotlight_

### Official Review · Reviewer_mek5 · 2023-07-05

**Soundness:** 3 good
**Presentation:** 3 good
**Contribution:** 3 good
**Rating:** 6
**Confidence:** 1

**Summary:**

This paper presents a series of methods on SE(3) equivariant convolution and transformer, which can operate on ray space. The experimental results show that the proposed method can help establish SE(3) reconstruction of signed distance functions and neural radiance fields. I'm not an expert in equivariant networks, I feel sorry but I may only be able to provide a general review.


**Strengths:**

- The paper looks solid in theory.
- Adequate transformation settings in the experiments demonstrate its equivariance property of the proposed network modules.

**Weaknesses:**

- The entire method description section is obscure to me.
- It is a bit hard to capture the overall idea when reading the sentences from Line 44 to Line 70.
- The qualitative results (visual comparisons) are limited.

**Questions:**

- What are the model sizes and training times compared to DISN (for SDF reconstruction) and IBRNet (for novel view synthesis)?

**Limitations:**

It looks the limitations are properly discussed.

---

> ### Author Rebuttal · Authors · 2023-08-10
>
> **1. “The entire method description section is obscure to me.”**
>
> As mentioned in the response to Reviewer VaKN, we will change the order of presentation in the method section. We will first define the two problems of generalized rendering and reconstruction from multiple views and then elaborate on how to make their main ingredients (convolution and transformer) equivariant. The definition of equivariance convolution requires several concepts from representation theory. We will introduce first the definition and then elaborate on the concepts so that even if the concepts themselves are quite complex, the motivation for using them is clear. We employed visualization techniques to vividly illustrate the key concepts outlined in the paper, with additional visual aids provided in the appendix. We will try to make them more comprehensible.
>
> **2. “It is a bit hard to capture the overall idea when reading the sentences from Line 44 to Line 70.”**
>
> We recognize that lines 53-61 describing the definition of equivariance might be confusing and we will rewrite them drawing parallels from other applications of equivariance, as follows.
> Everybody understands what equivariance is in 2D image operations. The output should be transformed the same way as the input image. Exact equivariance holds only when rotations are in multiples of 90 degrees because of the orthogonal sampling grid of the image.
> Things become a bit more complicated in 3D tasks when the input is a point cloud. There, equivariance means that if all input points are transformed, then the output (segmentation, reconstruction) should be transformed as well. Exact equivariance in this case means that all points will be transformed rather than a new sampling of the environment from a rotated sensor configuration. In our case, the input consists of features on rays in 3D. Exact equivariance, here, means that when all rays undergo the same transformation because of a change of reference frame the output features on rays or points will be transformed the same way. This differs from rotating the object because the rays (and content) captured after the rotation will be different from the content before the rotation.
>
> In lines 44 to 51, we extend the equivariant convolution in the ray space to the equivariant transformer in the ray space, which leverages the kernel in equivariant convolution.  We provide two cases of cross-attention: one is from rays to points, where the query feature is attached to points;  and key, value features are attached to the rays; another one is from rays to rays, where the query feature is attached to the target ray and the key, value features are attached to the source rays.
>
> Proceeding from lines 52 to 70, we delve into the practical applications of our model in 3D reconstruction and novel view synthesis, particularly within the realm of multiple views. We initially elucidate the specifics of equivariance within the context of multi-view settings. Subsequently, we detail how we seamlessly integrate the equivariant convolution and transformation modules to cater to the distinct requirements of these two tasks.
>
> **3. “The qualitative results (visual comparisons) are limited.”**
>
> We have introduced an additional visual comparison in Figure 3, which is illustrated in the rebuttal PDF. Due to space constraints within the PDF, the number of visualizations presented is limited. Nevertheless, we are dedicated to enriching the visual aspect of the paper by including more comparative visualizations to enhance clarity and understanding in the revised paper.
>
> **4. “What are the model sizes and training times compared to DISN (for SDF reconstruction) and IBRNet (for novel view synthesis)?”**
>
> For a fair comparison, our model size (num of parameters) is similar to that of DISN ( about 16.2M parameters).  Training duration extends to around 1.2 times that of DISN. For novel view synthesis, our model size (num of parameters) is similar to that of IBRnet (about 9.04M parameters). Training time experiences an approximately 1.8-fold increase compared to IBRnet due to the heightened complexity of calculations within each layer.

---

### Official Review · Reviewer_VaKN · 2023-07-06

**Soundness:** 3 good
**Presentation:** 1 poor
**Contribution:** 4 excellent
**Rating:** 5
**Confidence:** 2

**Summary:**

The aim of this paper is to build equivariance constraints into neural rendering and 3D reconstruction networks. As far as I can tell from the paper, the representation used is a mapping from a ray space to a vector space. The paper describes equivariant convolution and transformer layers that operate over the ray space. Finally, these layers are inserted into existing neural rendering and 3D reconstruction networks to impose equivariance over re-parameterization of the coordinate space in which we represent camera extrinsics.

**Strengths:**

1. The paper establishes a theory of equivariance in the ray space. This is an important contribution that could guide the design of future equivariant 3D scene representation approaches.
2. Adding equivariance leads to robustness to rotations and translations in 3D reconstructions. While it does not improve the quality of neural rendering, it makes renders consistent under coordinate frame transformations.

**Weaknesses:**

Overall, the paper lacks clarity and is not self-contained *at all*. I would suggest a major rewrite that adds a background section, re-structures the methods section and adds needed details to the experiments section.

1. The paper needs a background section. The background should cover at least the definition of rays (using your parameterization), ray space and the IBRNet. A background section is much more important than a 2-page introduction.
2. The methods section is missing a clear overview at the start about what neural components we need and what equivariance constraints we want to build into these components. The methods section directly jumps into dense exposition with frequent references to the 34-page appendix.
3. The proposed model that is actually used in the experiments section is delegated to the appendix.
4. It is not clear why IBRNet is the only baseline for novel view reconstruction. What about NeRF [1], Equivariant NeRF [2] or SRT [3]. The proposed method should be compared with other equivariant novel view reconstruction methods.
5. It is unclear if Section 4.2 involves novel view reconstruction or only rendering the same images in different coordinate frames. If the latter is the case, novel view reconstruction seems like a vital experiment.
6. There are very few qualitative examples of the rendered frames, both in the main paper and in the appendix. There should be a much more extensive qualitative comparison of image rendering across several SOTA methods.

**References:**

[1] https://arxiv.org/abs/2003.08934

[2] https://arxiv.org/abs/2006.07630

[3] https://arxiv.org/abs/2111.13152

**Questions:**

1. How well does the proposed method perform on novel view reconstruction?

2. How does the proposed method compare to Equivariant NeRF or the Scene Representation Transformer?

**Limitations:**

The limitations of the proposed method are not sufficiently addressed except for a brief mention in the conclusion.

---

> ### Author Rebuttal · Authors · 2023-08-10
>
> **W1:** We agree with the reviewer, and we will present the ray space definition and the context of IBRnet in the introduction. The mathematical definition and background of rays were initially placed in the appendix, a decision we acknowledge as not ideal.
>
> **W2:** We will rearrange and rewrite the methods section in the following way: Present first the generalized rendering and reconstruction from multiple views with their neural components of their architectures (convolutional and attentional) and their inputs and outputs. Then, move on with exposing the convolution and attention formula and then explain why we need all the math tools from representation theory necessary in the definition of convolution in homogeneous spaces which is the master-tool of our approach.
>
>  All this material exists, In the introduction lines 26-27, we give the clear purpose of the paper, that is we address the problem of learning geometric priors that are SE(3)-equivariant with respect to transformations of the reference coordinate frame. Further in the introduction, between lines 28 and 30, we clarify that our input entails a light field—a function characterized by its orientation within 3D space—whose values encompass radiance or a range of features derived from pixel data. Transitioning to lines 60-70 of the introduction, we provide a detailed exposition of the neural components or features that play a pivotal role in each stage of the 3D reconstruction and neural rendering tasks. We acknowledge the reviewer's input regarding the order of presentation, recognizing its potential to enhance the paper's overall comprehensibility.
>
> **W3:** The main contribution of the paper is the novel introduction of the equivariant convolution and transformer in a light field. We believe that NeurIPS is a conference appreciating novel methodological contributions and this is why we emphasized the description of them instead of the models of the task we applied them on. This is also the case in all novel equivariance papers published at NeurIPS. We still presented the architectures of the rendering and reconstruction models in Figures 6 and 7. We would prefer to devote any available space to explaining the background as requested in your previous point. Of course, if we have space we will elaborate more on the models in the main paper.
>
> **W4 (Q2):** We chose IBRNet as a baseline because it is the classic generalized Nerf method, which makes use of a conventional transformer of rays, similar to approaches in the literature mentioned in your question. Our architecture is based on the IBRNet, and we only replace the aggregation method with equivariant convolution and replace the conventional transformer with the equivariant transformer. Therefore it is fair to take IBRNet as the baseline to show the robustness of our proposed basic operation (convolution, transformer)).
>
> NeRF is not relevant to our approach because it performs a single scene optimization without using any prior. Any time we change the reference frame we have to rerun NeRF because NeRF is not equivariant to the choice of coordinate system where we compute density and color.
>
> The equivariance in the paper Equivariant Neural Rendering is not the equivariance we mean in our paper: Equivariant Neural Rendering enforces only approximate equivariance via a loss function rather than by design as we do. We will definitely add it to the citations.
>
> In the rebuttal PDF, we added a comparison to SRT (Scene Representation Transformer), as outlined in Table 3. We present the quantitative outcomes obtained from the MultiShapeNet dataset. Notably, both IBRnet and our model's performances lag behind that of SRT.
>
> We want to emphasize that our central contribution is to propose an equivariant convolution and transformer on ray space, which can be generally embedded into broad models in 3D learning. SRT is not strictly equivariant, it has the assumption that the views are upright, is inconsistent with respect to the permutation of the cameras, and depends on the camera ID embedding. By randomly choosing the first camera frame as the canonical frame, it approximately achieves data-driven equivariance.
>
> Our framework is sufficiently generic to allow us to take architectures like the SRT and convert them to be equivariant as long as they consist of transformer (or convolutional) modules.
>
> **W5 (Q1):**  This is a misunderstanding. We perform novel view rendering (=reconstruction) from completely different novel 3D poses (section 3.4).
>
> **W6:** We will augment the paper with additional qualitative results to enhance the overall presentation, and we have included extra visualizations in the rebuttal PDF. Although the page constraints of the rebuttal PDF limited the number of visualizations we could include, rest assured that we will incorporate an expanded set of results in the forthcoming revised version.
>
> **Limitation:**  The primary constraint of this approach stems from the finite sampling of the light field. Sparse view-based sampling inadequately addresses substantial object displacements accompanied by significant changes in perspective, resulting in the breakdown of equivariance, which also explains its suboptimal performance on the MultiShapeNet dataset due to noncomprehensive light field sampling. We will expand upon this aspect in a more detailed section within the forthcoming revised paper.

---

> > ### Comment · Reviewer_VaKN · 2023-08-20
> > **Response**
> >
> > Thank you for the rebuttal and the additional comparisons! I raised my score.
> >
> > You are right that NeRF does not fit into your experimental setup, my mistake. It is interesting that SRT does so much better; e.g. https://arxiv.org/abs/2304.00947 took a step towards making SRT invariant to the reference frame.

---

> > > ### Author Response · Authors · 2023-08-21
> > >
> > > Thank you for appreciating our rebuttal and providing valuable feedback.
> > >
> > > The significant performance disparity between SRT and our approach, as well as IBRNet, can be attributed to i) the order of magnitude in the model size (74M parameters vs 9.04M ours), ii) the global nature of the SRT set-latent representations, and iii) in the larger encoder, encompassing a CNN and a state of the art Vision Transformer applied cross image patches.
> > >
> > > We appreciate the valuable reference you have provided, and we will ensure to cite it. While both our method and RePAST address the frame problem through relative pose, there are differences in the approach. Our model maintains theoretical equivariance even in cases of individual camera rotations around their axes or minor individual rotations accompanied by small content changes. In RePAST, patch tokens are reliant on the camera frame, while the relative pose is tied to the query camera frame. Although we have not conducted experiments with cameras rotated around their axes for neural rendering, we have done so for 3D reconstruction, showcasing the robustness of our model.

---

### Official Review · Reviewer_gW8w · 2023-07-06

**Soundness:** 3 good
**Presentation:** 3 good
**Contribution:** 3 good
**Rating:** 5
**Confidence:** 3

**Summary:**

This paper introduces a method for leveraging geometric priors in 3D reconstruction and novel view rendering when the input views are insufficient. The authors propose learning priors from 2D images using a 2D canonical frame and a 3D canonical frame. They achieve coordinate frame equivariance by introducing an SE(3)-equivariant convolution and transformer in the 3D ray space. The paper demonstrates the efficacy of their approach in tasks such as 3D reconstruction and neural rendering using multiple views, showcasing robust results without transformation augmentation.

**Strengths:**

The proposed method in this paper offers several strengths:

1. Geometric Priors: The approach leverages geometric priors to enhance 3D reconstruction and novel view rendering. By incorporating prior knowledge, it improves the quality of results, particularly when the input views have limited coverage and inter-view baselines.

2. Equivariance to Coordinate Frame Transformations: The method ensures equivariance to coordinate frame transformations, allowing it to handle different orientations and positions of the cameras. This enables more accurate and consistent reconstruction and rendering across various coordinate frames.

3. SE(3)-Equivariant Convolution and Transformer: The introduction of SE(3)-equivariant convolution and transformer in the 3D ray space provides a powerful tool for learning and exploiting geometric priors. It allows for effective feature extraction and representation, leading to improved reconstruction and rendering results.

4. Adaptability to Different Tasks: The method demonstrates adaptability to different tasks, including equivariant 3D reconstruction and equivariant neural rendering. It can be tailored to specific applications, making it versatile and applicable in various scenarios.

5. Robustness without Transformation Augmentation: The approach achieves robust results even in datasets with roto-translated transformations, without the need for additional transformation augmentation techniques. This reduces the complexity and computational requirements while maintaining high-quality outputs.

Overall, these strengths make the proposed method valuable for improving 3D reconstruction and novel view rendering by incorporating geometric priors and addressing challenges related to coordinate frame transformations.

**Weaknesses:**

1. I appreciate the solid math and notation the author adopted in this paper, which makes the writing theoretically sound. However, this may make the reading very difficult to follow when many mathematical explanations irrelevant to the method itself appears in the paper (like the whole paragraph from line 131-138, which in my understanding could be all moved to the appendix). I believe you could lint your writing and leave more space for the qualitative/quantitative experiments section.
2. I appreciate the notation explanations in the appendix. However, when I first read the terms like "type-1 features", it confuses me and I need to look back to see what I have missed before. A more intuitive name or a brief explanation of these important terminology is required, I think. This is just a recommendation on the writing and won't change my final rating.
3. More experiments (ablations, baselines) are well needed to demonstrate the soundness of this paper as a new 3D representation learning framework.

**Questions:**

1. I appreciate the soundness of your math, while in Sec 3.1, could you briefly explain, what should the convolution from rays to point/rays to rays handle and why are they necessary? Any ablation on these designs, like using any of the convolution alone in the setting of overfitting/generalized NeRF?

2. This paper spends a large paragraph explaining how to achieve equivariance in the light field space using convolution and transformer. However, I still don't quite see the advantages behind it, .e.g., why in the airplane dataset, w/o the equivariance design, Fvor is still superior compared to the proposed method under R/Y/SO(3) transformations? Also, IBR-Net also surpasses the proposed method over many datasets and metrics.

3. I wonder in Sec 4.2, should the original NeRF & PixelNeRF and other strong baselines that uses canonical pose be included in the comparisons? Besides, I wonder whether this method could benefit the noisy pose reconstruction setting, e.g., comparing with "SPARF Neural Radiance Fields from Sparse and Noisy Poses".

**Limitations:**

Already discussed in the weakness.

---

> ### Author Rebuttal · Authors · 2023-08-09
>
> **W1:** Indeed, lines 131-138 introduce material that is beyond the standard toolbox even for a mathematically avid NeurIPS reader.  Nevertheless, this material is essential for establishing the definition of the generalized convolution as depicted in Equation 3 of the paper. We agree with the reviewer that we can, though, introduce the convolution and still refer to the appendix to explain the terms in the convolution.
>
> **W2:** We really appreciate these comments and we could easily replace type-0 with scalar and type-1 with vectors avoiding, thus, the cryptic representation theory terms.
>
> **W3:** We conducted ablation studies during the initial submission, and the results are presented in Tables 1 and 2 on page 28 of the appendix. We could move them to the paper upon saving space after moving theoretical segments to the appendix. In Table 1 we showed ablations about the importance of convolution and transformer as well as the vector (type-1) features.
>
> In addition, we provide an ablation in the rebuttal PDF. We replaced the equivariant transformer for points along the ray with the canonical transformer; Our rationale for not exclusively altering the equivariant convolution and transformer from rays to rays lies in the fact that such a change would eliminate the presence of a feature specifically tailored for equivariance in subsequent modules. Consequently, we would be left with the option of applying a conventional transformer to this feature, resulting in a setup identical to IBRNet, which we have already compared against.
>
> In order to apply a conventional transformer to the equivariant feature, we first convert the equivariant feature to the invariant (scalar) feature, which makes the model also equivariant, but the performance is inferior to the full model and it shows that the equivariant feature contains more information and the equivariant transformer therefore will be more expressive.
>
> **Q1:** We need ray-to-ray convolution where both the input and output consist of fields distributed across rays. Conversely, when the input constitutes a field distributed across rays, like a radiance (feature) field, but the output takes the form of a field across 3D Euclidean space, such as an occupancy field or Signed Distance Function (SDF), ray-to-point convolution is necessary.
>
> The selection between ray-to-ray and ray-to-point convolutions is contingent upon the specific task at hand. The pivotal characteristic of the proposed convolution lies in its equivariance to SE(3).  Equivariant ray convolution is essential, providing consistency under SE(3) transformations and overcoming challenges in 3D vision with unknown poses of objects or scenes.
>
> As stated in W3, we did the ablation by replacing the equivariant transformer over points along the ray, in addition, IBRNet is our baseline by replacing equivariant convolution and transformer from rays to rays with the conventional view aggregation in IBRNet and replacing the equivariant transformer over points along the ray by the ray transformer in IBRNet.
>
> Through this analysis, we ascertain that equivariant convolution and transformer not only exhibit robustness towards transformations but also highlight the increased potency of the equivariant transformer when maintaining equivariant features within the model.
>
> **Q2:** It is important to highlight that, across all settings, the object pose is provided in Fvor, which significantly aids the 3D reconstruction process by incorporating point positions into the method. In the Y/Y setting, akin to I/I, Fvor outperforms our method due to the provision of more informative data.
>
> Regarding the SO(3)/SO(3) and R/R settings within the airplane category, we attribute the observed trend to the unique characteristics of airplanes. Being slender objects with distinct features, airplanes are relatively straightforward for non-equivariant networks to memorize transformations through training augmentation or data-driven methods. This can explain the comparable performance between our method and Fvor in such scenarios.
>
> For the neural rendering experiment, IBRNet surpasses our method in the I/I setting, which is understandable, since it uses the absolute positioning encoding while ours uses the relative pose and constrained kernels.
>
> IBRNet is comparable to our method in I/SO(3) settings in the Diffuse Synthetic 360 dataset. We think it is because the synthetic data is less sensitive to the ray direction and has sparser source views, which makes it less dependent on ray directions. The only difference between IBRnet and ours is how to deal with the geometric relation of the rays, when the task is much more dependent on the image feature rather than the geometric information, the priority of the method would be weakened.
>
> **Q3:**   NeRF does not aim to learn a 3D prior by using the radiance field as input, but rather to fit a scene by regressing its radiance field. NeRF's application is primarily geared towards individual scenes, requiring network retraining or optimization reruns whenever the reference frame for densities and colors changes. In this sense, a comparison with the NeRF-based SPARF is not adequate. We will cite it nevertheless.
>
> Our rebuttal PDF includes a robust baseline, NeuRay, as illustrated in Table 2 and Figure 2 of the PDF. NeuRay, like our method, builds upon the IBRNet baseline but goes further by incorporating a depth map or cost volume to estimate point visibility.  We conduct a comparative analysis with NeuRay (Neural Rays for Occlusion-aware Image-based Rendering) on the Real Forward-facing LLFF dataset.
> Our observations reveal that our method demonstrates comparable performance to NeuRay in the I/I setting. In the I/SO(3) configuration, NeuRay experiences a performance decline and inconsistency, whereas our method remains robust against rotations.

---

### Official Review · Reviewer_sFoS · 2023-07-08

**Soundness:** 4 excellent
**Presentation:** 3 good
**Contribution:** 4 excellent
**Rating:** 7
**Confidence:** 3

**Summary:**

This paper proposes to study the geometric priors for 3D reconstruction and neural rendering with a novel perspective of multi-view equivariance. A theoretically-sound definition of the ray neighborhoods with SE(3) is obtained with the theoretical deduction of characterizing ray space with group theory. Then, the SE(3)-equivariant convolution and cross-attention operators are presented. In the experiments, the authors evaluate the proposed mathematical framework on two tasks, multi-view 3D object reconstruction and neural rendering. For the multi-view 3D object reconstruction, the presented SE(3)-equivariant network outperforms Fvor and DISN in most settings for chars, airplanes, and cars on the ShapeNet dataset. For neural rendering, the proposed method obtains better results on the I/SO(3) setting to justify its design. Overall, I think this paper brings us some new messages to understand the ray space by using group theory.

**Strengths:**

1. This paper presents a theoretical perspective to understand the ray spaces for learning geometric priors for multi-view 3D reconstruction. The thorough theoretical analysis presented in this paper is new to me.

2. Based on the theoretical findings, the authors presented equivariant convolution and transformer in ray space and justified their design on two tasks with positive results obtained.

3. This paper is well written with a detailed appendix to understand their theory.

**Weaknesses:**

1. The core of this paper is studying the geometric relationship between rays to define the ray neighborhoods, and then, convolution and attention can be induced in the ray space. If I understand correctly, what's the relationship between the point correspondences and the neighbors in the ray space? For example, are the corresponding rays near or the same in the ray space if we have a pair of keypoint matches $x$ and $x'$ for a pair of input images?

2. As for the generalized neural rendering task, the authors stated that their model queries a target ray and obtains neighboring rays
from source views in the first. To my knowledge, such operations can be done by following the epipolar geometry. Thus I am curious about the difference between the operation used in this paper and the alternative solutions using epipolar geometry.


**Questions:**

I am confused about the sentence "given only the relative poses of the cameras, we show how to learn priors...". Why do authors need to highlight the relative poses in the paper? It is strange for me.

**Limitations:**

The authors have discussed their limitations and broader impacts.

---

> ### Author Rebuttal · Authors · 2023-08-09
>
> **1. “The core of this paper is studying the geometric relationship between rays to define the ray neighborhoods, and then, convolution and attention can be induced in the ray space. If I understand correctly, what's the relationship between the point correspondences and the neighbors in the ray space? For example, are the corresponding rays near or the same in the ray space if we have a pair of keypoint matches x and x’  for a pair of input images?”**
>
> We really appreciate this question. Ultimately, it can be argued that every IBR or reconstruction method inherently boils down to addressing a correspondence challenge.
>
> When a pair of input images share matching keypoints, denoted as x and x', these keypoints represent neighboring rays. Figure 3 of the paper illustrates that when considering a specific ray x, its neighboring rays traverse within a confined cylinder. This cylinder utilizes x as its central axis and the maximum neighboring distance d as its radius. In the context of a two-view setup, as depicted in Figure 9 in the appendix, for a given ray x within view A, the neighboring rays can be categorized into two distinct groups. The first group encompasses rays originating from view A, where the angle between these rays and x is smaller than the specified threshold for the neighboring maximum angle. The second group consists of rays emanating from view B, located in close proximity to the epipolar line associated with x in view B. This proximity is attributed to the fact that the neighboring region corresponds to the projection of the confined cylinder onto view B.
>
> In this configuration, keypoint correspondence entails that subsequent to convolution and attention, querying the density of the keypoint would yield a significantly elevated value.
>
> **2. “As for the generalized neural rendering task, the authors stated that their model queries a target ray and obtains neighboring rays from source views in the first. To my knowledge, such operations can be done by following the epipolar geometry. Thus I am curious about the difference between the operation used in this paper and the alternative solutions using epipolar geometry.”**
>
> This question is also spot on!
> As outlined in the first question and depicted in Figure 9 of the appendix, the neighboring rays within the source view of the target ray are situated within the vicinity of the epipolar line associated with the target rays in that source view. When we establish the maximum neighboring distance as 0 and set the angle threshold to $-\pi$, the resulting neighboring rays precisely align with the rays located on the epipolar line.
>
> **3.  “I am confused about the sentence "given only the relative poses of the cameras, we show how to learn priors...". Why do authors need to highlight the relative poses in the paper? It is strange for me.”**
>
> Certainly, some clarification is needed to contextualize this concept within classical multi-view geometry. In approaches such as classical space carving or plane-sweep reconstruction, a crucial step involves the selection of a reference frame for voxels or sweeping planes, followed by an optimization process. In those cases, there is no question of equivariance, and many times we choose as reference frame one of the camera frames.
>
> However in learning methods that use a prior (encoded in the weights of the network), this prior depends on the choice of the reference frame, and by default generalized reconstruction or rendering methods are not equivariant to the choice of this frame. In calibration cases, where only relative poses are given by an SfM method (instead of a world coordinate system by standard calibration) choosing one of the camera frames as a reference frame will alter the inference of the network if the network is not equivariant. Our method guarantees that this will not happen.

---

> > ### Comment · Reviewer_sFoS · 2023-08-19
> > **Reply to authors' rebuttal**
> >
> > I wish to convey my gratitude to the authors for their comprehensive rebuttal. Following a thorough review of the rebuttal materials and additional comments, I find the technical contributions of the paper to be solid. I am pleased to note that the authors have acknowledged the need for revisions to enhance clarity. Their commitment to addressing these concerns is commendable. I am satisfied with the provided rebuttal and look forward to the revised version.

---

> > > ### Author Response · Authors · 2023-08-20
> > >
> > > We really appreciate you taking the time to review our rebuttal materials and additional comments. Your feedback is highly valued.  We are dedicated to enhancing the paper's clarity as per the concerns highlighted in the reviews, going beyond the revisions presented in the attached PDF.

---

### Official Review · Reviewer_MuuL · 2023-07-30

**Soundness:** 3 good
**Presentation:** 2 fair
**Contribution:** 3 good
**Rating:** 5
**Confidence:** 1

**Summary:**

The paper proposes a novel approach for reconstruction from multiple views using equivariant shape priors. The paper proposes the $SE(3)$-equivariant generalized convolution as the fundamental operation on a light field whose values may be radiance or features. Input ray features and producing output ray features and point features using two different $SE(3)$-equivariant convolutions. The paper demonstrates $SE(3)$-equivariance by obtaining robust results in roto-translated datasets without performing transformation augmentation.

To say it upfront, I do not have background in Equivariant Networks and consider the paper outside my area of expertise. I find it difficult to understand the technical details of the paper. As such, I can only provide some high-level feedback to the paper and hope the AC and other reviewers can provide more detailed evaluations.

**Strengths:**

1. The paper is well written and easy to follow.
2. The experiment results show that the provided models can outperform previous methods on various tasks.
3. This novel general equivariant representation framework for light fields can inspire further work on 3D vision and graphics tasks.


**Weaknesses:**

1. The figures in this paper could be redesigned; some of them (Figure 4, 5) are so small that it is difficult to read the text and symbols they contain. In addition, the effect shown in Figure 9 is not intuitive.
2. The paper demonstrated good reconstruction results on Shapenet. Can the method proposed in this paper reconstruct surfaces on real datasets such as SparseNeuS.
3. About neural rendering experiments, the paper only shows results comparing with IBRNet, which doesn't seem to be the latest method. Can the authors provide the results of more experiments? (a. Local Implicit Ray Function for Generalizable Radiance Field Representation  b. Learning to Render Novel Views from Wide-Baseline Stereo Pairs c. Neural Rays for Occlusion-aware Image-based Rendering)

**Questions:**

Please see Weaknesses.

**Limitations:**

Please see Weaknesses.

---

> ### Author Rebuttal · Authors · 2023-08-09
>
> **1. “The figures in this paper could be redesigned; some of them (Figure 4, 5) are so small that it is difficult to read the text and symbols they contain. In addition, the effect shown in Figure 9 is not intuitive.”**
>
> We will improve the design of Figures 4 and 5, and make them larger while saving space by relocating theoretical content from the paper to appendices. We hope that this will make the figures legible and increase the understanding of the reader.
>
>  In reference to Figure 9, in the top row of images, a noticeable distinction arises between the images generated by IBRNet(I) and IBRNet(SO(3)) as they contain a non-existent black area within the pillar. Contrarily, our results exhibit no such black area. Furthermore, when comparing IBRNet(I) and IBRNet(SO(3)), the latter presents several blurred transverse lines, whereas our method remains robust against rotations.
>
> For the second row, our performance closely matches that of IBRNet in nonrotated scenarios. However, when the reference frame is rotated, our method maintains its quality, while IBRNet's output becomes increasingly blurred, particularly in the bottom right corner. We are committed to enlarging the figures to enhance their visual clarity and intuitiveness.
>
> Because the rebuttal pdf is limited to one page the figures are still small there, but they will be shown in legible size in the revised paper.
>
> **2.“The paper demonstrated good reconstruction results on Shapenet. Can the method proposed in this paper reconstruct surfaces on real datasets such as SparseNeuS.”**
>
> We invested considerable effort in the theoretical formulation of equivariant convolution and attention in a light field, and we were planning to conduct experiments on real datasets in a subsequent paper. We believe that the community will benefit from our exposition of a novel convolution and attention instead of squeezing it into a few paragraphs so that we make the real experiments fit. While we recognize the significance of real-world reconstruction, ShapeNet has been an established dataset for surface reconstruction in hundreds of papers in computer vision.
>
> **3. “Regarding neural rendering experiments, the paper only shows results compared with IBRNet, which doesn't seem to be the latest method. Can the authors provide the results of more experiments? (a. Local Implicit Ray Function for Generalizable Radiance Field Representation b. Learning to Render Novel Views from Wide-Baseline Stereo Pairs c. Neural Rays for Occlusion-aware Image-based Rendering)”**
>
> We have selected IBRNet as our baseline due to its status as a classic generalized NeRF method. This choice is rooted in the fact that IBRNet employs the conventional ray transformer, a technique also referenced in the pertinent literature. Our architecture builds upon IBRNet's foundation, with modifications limited to the replacement of the aggregation mechanism using equivariant convolution and the conventional transformer substituted by an equivariant transformer. Therefore, utilizing IBRNet as a baseline is justifiable to showcase the resilience of our proposed fundamental operations (convolution, transformer).
>
> We have added a comparative analysis with NeuRay (Neural Rays for Occlusion-aware Image-based Rendering) on the Real Forward-facing LLFF dataset. Referencing Table 2 and Figure 2 in the provided PDF, our observations reveal that our method demonstrates comparable performance to NeuRay in the I/I setting. In the I/SO(3) configuration, NeuRay experiences a performance decline and inconsistency, whereas our method remains robust against rotations.
>
> Unfortunately, we were unable to locate the publicly available code for the "Local Implicit Ray Function for Generalizable Radiance Field Representation." This absence of code renders a direct comparison unfair. Furthermore, "Learning to Render Novel Views from Wide-Baseline Stereo Pairs," a method centered around rendering from two views, is not optimal for our approach, as our method assumes ample information extraction through light field sampling. Nevertheless, it would be worth considering whether expanding the ray neighborhood in our technique could offset the sparse light field sampling. We will cite these valuable references.

---

### Author Rebuttal · Authors · 2023-08-10

We extend our gratitude for the invaluable feedback and suggestions provided by the reviewers. We are pleased to note that all reviewers appreciated the contribution (2 excellent, 3 good) and the soundness (1 excellent, 4 good). All reviewers listed multiple strengths of the approach and none of them doubted its novelty.

The most negative reviewer (VaKN) says: “The paper establishes a theory of equivariance in the ray space. This is an important contribution that could guide the design of future equivariant 3D scene representation approaches.” Reviewer MuuL writes: “The paper proposes a novel approach for reconstruction from multiple views using equivariant shape priors….The paper is well written and easy to follow.”
Reviewer sFoS says: “This paper presents a theoretical perspective to understand the ray spaces for learning geometric priors for multi-view 3D reconstruction. The thorough theoretical analysis presented in this paper is new to me…This paper is well written with a detailed appendix to understand their theory.” Reviewer gW8w lists 5 strengths and says: “Overall, these strengths make the proposed method valuable for improving 3D reconstruction and novel view rendering by incorporating geometric priors and addressing challenges related to coordinate frame transformations.”

Two are the main concerns raised: the lack of clarity in the presentation and the lack of comparisons to approaches beyond IBRNet.

We take your feedback about the presentation into serious consideration, we are committed to addressing this concern in the revised version as described in the individual responses to the reviewers. We will convert the introduction section into a background section including necessary definitions that we had in the appendix. We will clarify the definition of equivariance in light fields.
We will change the order of the presentation so that the example tasks are described first in order to motivate the use of equivariant modules like convolution and transformer. Last, we will add more qualitative results to show the performance of our approach.

The second primary concern raised pertains to the rationale behind comparing our approach with IBRNet, along with suggestions for incorporating comparisons with other state-of-the-art methods and conducting further ablation studies.

We have selected IBRNet as our baseline due to its status as a classic and widely recognized generalized NeRF method. IBRNet employs the conventional ray transformer, a technique also prevalent in the literature referred to in the question. Our architecture builds upon the foundations of IBRNet, with a key modification involving the replacement of the combining method with equivariant convolution and the substitution of the conventional transformer with an equivariant counterpart. This reasoned approach justifies IBRNet's selection as the baseline, effectively showcasing the robustness of our fundamental operations, namely convolution, and transformer.

We have added a comparative analysis with NeuRay (Neural Rays for Occlusion-aware Image-based Rendering) on the Real Forward-facing LLFF dataset and with SRT (Scene Representation Transformer) on the MultiShapeNet. We further added more ablations, all included in the rebuttal PDF.

We are committed to acknowledging and citing all the valuable references they have contributed to the paper.

---

### Decision · Program_Chairs · 2023-09-21

**Decision:**

Accept (spotlight)

**Comment:**

Existing methods extract features from multi-view images and aggregate them as geometric priors, which are not SE(3)-equivariant concerning the transformation of the reference frame. This paper introduces the SE(3)-equivariant generalized convolution in ray space, resulting in consistent features that could facilitate 3D reconstruction and novel view synthesis. However, reviewers have highlighted significant concerns, noting that the paper is mainly theoretical and lacks clarity. Additionally, its effectiveness in scenarios with sparse views has not been assessed. The AC acknowledges the potential impact of SE(3)-equivariant feature extraction and supports this paper for a spotlight presentation.  Additionally, the AC encourages the author(s) to refine the presentation's clarity and address the unexplored scenarios.